# Comparative analysis of the Mexico City Prospective Study and the UK Biobank identifies ancestry-specific effects on clonal hematopoiesis

Sean Wen[1,2,9], Pablo Kuri-Morales [3,9], Fengyuan Hu[1], Abhishek Nag[1], Ioanna Tachmazidou[1], Sri V. V. Deevi [1], Haeyam Taiy[1], Katherine R. Smith [1], Douglas P. Loesch [1], Oliver S. Burren[1], Ryan S. Dhindsa [4], Sebastian Wasilewski[1], Jesus Alegre-Díaz [5], Jaime Berumen [5], Jonathan Emberson [6], Jason M. Torres [6], Rory Collins[6], Keren Carss [1], Quanli Wang[4], Slavé Petrovski [1], Roberto Tapia-Conyer [5,10], Margarete A. Fabre[1,7,10], Andrew R. Harper [1,8,10], George S. Vassiliou [2,7,10 ✉] & Jonathan Mitchell [1,10 ✉]

The impact of genetic ancestry on the development of clonal hematopoiesis (CH) remains largely unexplored. Here, we compared CH in 136,401 participants from the Mexico City Prospective Study (MCPS) to 416,118 individuals from the UK Biobank (UKB) and observed CH to be significantly less common in MCPS compared to UKB (adjusted odds ratio = 0.59, 95% confidence interval (CI) = [0.57, 0.61], $P = 7.31 \times 10^{-185}$). Among MCPS participants, CH frequency was positively correlated with the percentage of European ancestry (adjusted beta = 0.84, 95% CI = [0.66, 1.03], $P = 7.35 \times 10^{-19}$). Genome-wide and exome-wide association analyses in MCPS identified ancestry-specific variants in the *TCL1B* locus with opposing effects on *DNMT3A*-CH versus non-DNMT3A-CH. Meta-analysis of MCPS and UKB identified five novel loci associated with CH, including polymorphisms at *PARP11/CCND2*, *MEIS1* and *MYCN*. Our CH study, the largest in a non-European population to date, demonstrates the power of cross-ancestry comparisons to derive novel insights into CH pathogenesis.

The overwhelming majority of genetic association studies to date have been performed on participants of European descent, most of whom were recruited from the USA and UK[1]. The study of non-European cohorts provides opportunities for novel discovery and orthogonal validation of risk variants and loci and for improving our understanding of disease etiology, while ensuring that the benefits of genomic research are broadly applicable[2]. This is especially true for the discovery of rare variants, as their origins are more recent and they tend to be more geographically clustered and population-specific[1,2].

As with all human cells, hematopoietic stem cells (HSCs) accumulate somatic mutations with advancing age. Some of these mutations confer a fitness advantage to the affected HSCs, promoting clonal expansion and engendering the phenomenon of CH[3–8]. CH is associated with an increased risk of hematological and other cancers[9] as well as non-oncological pathologies such as cardiovascular, renal, pulmonary and liver diseases[10–14]. Risk factors that contribute to increased CH include non-modifiable characteristics such as age and sex, exposures such as smoking or treatment with genotoxic drugs and heritable genetic variation[15–17]. However, the impact

of ancestry on CH development remains incompletely understood given that, to date, studies of CH have focused largely on cohorts of European or European-dominant ancestry[15–17].

To address this knowledge gap, we analyzed whole-exome sequencing (WES) data to identify CH in 136,401 participants recruited to the MCPS. We compared the CH status in MCPS participants with 416,118 individuals from the UKB and performed an intra-population analysis of MCPS individuals with varying proportions of European ancestry to delineate the contribution of ancestry to CH. We then performed single-variant genetic association analysis of common and rare germline variants and gene-level genetic association analysis of rare germline variants with CH. The genetic association analyses were performed in MCPS participants alone and in a cross-ancestry meta-analysis combining MCPS and UKB participants. Our study, to the best of our knowledge the largest investigation of CH in a non-European population to date, gives new insights into the contribution of ancestry to CH, discovers ancestry-specific risk variants and identifies novel risk loci from combined analysis of UKB and MCPS participants, among other findings.

## Results

### Frequency of CH

Genetic ancestry at the continental level was determined for MCPS and UKB participants from WES data using *peddy*[18], a machine-learning classifier trained on 2,504 individuals of known ancestry from the 1000 Genomes Project[19]. Latin American populations are genetically diverse, with an ancestral composition resulting from recent admixture of continental populations. We therefore restricted further analyses to 136,401 MCPS individuals with admixed Indigenous American, European and African ancestry (≥95% *peddy*-predicted probability Admixed American) and 416,118 UKB individuals (≥95% *peddy*-predicted probability European) who passed all quality control filters (see Methods and Supplementary Tables 1–3). Somatic variants were called from the WES data using MuTect2 (refs. [20,21]) and filtered against a catalog of predefined mutations in 15 previously validated CH driver genes[10,22] (see Methods and Supplementary Table 4). Importantly, the WES protocol and the variant calling workflow were the same for the UKB and MCPS cohorts (see Methods), resulting in similar variant allele frequency and sequencing coverage profiles across CH driver genes (Supplementary Figs. 1a–j and 2a–o). In total, we detected 4,678 somatic variants in 4,249 individuals in MCPS, and 22,161 somatic variants in 20,488 individuals in UKB (Supplementary Tables 5 and 6). The most recurrently mutated CH driver genes identified among MCPS participants were *DNMT3A, TET2, ASXL1, PPM1D* and *TP53* (Fig. 1a), consistent with previous WES studies of European ancestry[15,16]. The frequency of CH correlated strongly with age in both UKB and MCPS ($P_{UKB}$ and $P_{MCPS} < 2.2 \times 10^{-16}$; Fig. 1b). The characteristics of CH variants identified displayed patterns in line with previous reports (Extended Data Figs. 1a–i and 2a–d), such as for mutation types (Extended Data Fig. 1f,g) and gene-specific CH associations with age (Extended Data Fig. 2a–d and Supplementary Table 7).

Overall CH frequency was significantly higher among UKB (4.92%) than MCPS participants (3.12%, $P_{\chi2} < 2.2 \times 10^{-16}$). Although our panel of 15 genes contains those genes most commonly mutated in CH, as an additional validation, we repeated the analysis using a 58-CH gene panel from a previous publication[22] and found a similar difference in CH frequency between UKB (5.66%) and MCPS (3.62%, $P_{\chi2} < 2.2 \times 10^{-16}$). To account for the different age distributions (MCPS range, 35–112 years; UKB range, 40–70 years; Extended Data Fig. 2e), we compared CH frequency after age-matching and sex-matching and found that it remained significantly higher among UKB than MCPS participants across all ages (overall CH frequency, 4.55% and 2.87%, respectively, $P_{\chi2} < 2.2 \times 10^{-16}$; Extended Data Fig. 2f).

Consistent with the increased frequency of CH in UKB, we observed an increased risk of overall CH in UKB relative to MCPS participants in an inter-population logistic regression with CH as the outcome and population as the main predictor adjusted for age, sex and smoking

(odds ratio (OR) = 1.69, 95% CI = [1.63, 1.75], $P = 7.31 \times 10^{-185}$; Fig. 1c). We generally observed the same pattern when analyzing CH genes individually, with significantly increased risk in UKB versus MCPS of CH driven by mutations in each of the five most common genes (*DNMT3A, TET2, ASXL1, PPM1D* and *TP53*) as well as *JAK2* and *SRSF2*. Focusing on the two most common CH genes, *DNMT3A* and *TET2*, the difference in magnitude of increased risk in UKB was particularly marked (OR = 1.99, 95% CI = [1.90, 2.09], $P = 1.34 \times 10^{-171}$ and OR = 1.48, 95% CI = [1.38, 1.58], $P = 1.22 \times 10^{-30}$ for *DNMT3A* and *TET2*, respectively). Sensitivity analysis including sequencing coverage in addition to age, sex and smoking as covariates in our logistic regression model led to similar results (Extended Data Fig. 3a). A similarly increased risk of CH among UKB participants was observed when we matched participants for age, sex and smoking status as an additional approach to account for demographics differences between the two cohorts (Extended Data Fig. 3b–h and Supplementary Table 8).

### Association between ancestry and CH frequency

To further investigate the relationship between ancestry and CH, we leveraged the admixed ancestry of the MCPS participants[23]. The average proportion of Indigenous American, European and African genome, as inferred by RFMix2.0 (ref. [24]) across MCPS participants included in our study was 66%, 31% and 3%, respectively (Extended Data Fig. 4a). Their mosaic haplotype structure, incorporating multiple intercontinental genetic ancestries, provides an opportunity to robustly assess the relationship between genetic ancestry and CH frequency. We found that the frequency of CH was significantly higher in MCPS participants whose genomes were >50% ancestrally European than in MCPS participants whose genomes were >50% ancestrally Indigenous American across all age groups (Fig. 2a), with an overall CH frequency of 4.65% versus 2.83%, respectively ($P_{\chi2} < 2.2 \times 10^{-16}$). This observation of higher overall CH frequency in MCPS participants who were genetically more European compared to those who were genetically more American held true when we extended the list of CH driver genes to the 58-gene panel[22], with overall CH frequency of 5.11% versus 3.33%, respectively ($P_{\chi2} < 2.2 \times 10^{-16}$). Assessing genetic ancestry within MCPS as an ordinal variable further demonstrated a positive correlation between the fraction of individuals' genomes derived from European ancestry and overall CH frequency (Fig. 2b and Extended Data Fig. 4b–e).

Smoking frequency has been reported to be higher in individuals of European descent than in those of American descent among self-reported Hispanics and Latinos[25]. In MCPS, a logistic regression model including age and sex as covariates showed that the proportion of European ancestry was indeed associated with smoking frequency (ever-smoker, beta = 1.64, 95% CI = [1.56, 1.71], $P < 2.2 \times 10^{-16}$; previous smoker, beta = 1.05, 95% CI = [0.96, 1.14], $P < 2.2 \times 10^{-16}$; current smoker, beta = 2.01, 95% CI = [1.91, 2.10], $P < 2.2 \times 10^{-16}$). As an example, the frequency of ever-smokers among individuals with ≥70% European genome was 66.6% but only 39.7% among individuals with <10% European genome (Fig. 2c). Smoking, in turn, was associated with increased risk of CH as previously reported[15–17] (ever-smoker, OR = 1.13, 95% CI = [1.05, 1.21], $P = 8.21 \times 10^{-4}$; previous smoker, OR = 1.10, 95% CI = [1.02, 1.20], $P = 0.017$; current smoker, OR = 1.17, 95% CI = [1.07, 1.28], $P = 6.13 \times 10^{-4}$). Stratified by smoking status, the trend of increasing CH frequency with increasing European genome percentage was observed among individuals across all smoking strata (Extended Data Fig. 4f–i).

Finally, in a logistic regression model with CH as the outcome and proportion of European genome (in bins of 10%) as the main predictor adjusted for age, sex and smoking status, we observed increasing CH risk with increasing proportion of European genome (Extended Data Fig. 4j,k). Notably, the group of individuals with the highest percentage of European genome (≥70%) had the highest risk of CH relative to the group of individuals with the lowest percentage of European genome (<10%; OR = 1.87, 95% CI = [1.46, 2.39], $P = 4.24 \times 10^{-7}$). Consistent with this finding, we observed an increased risk of overall

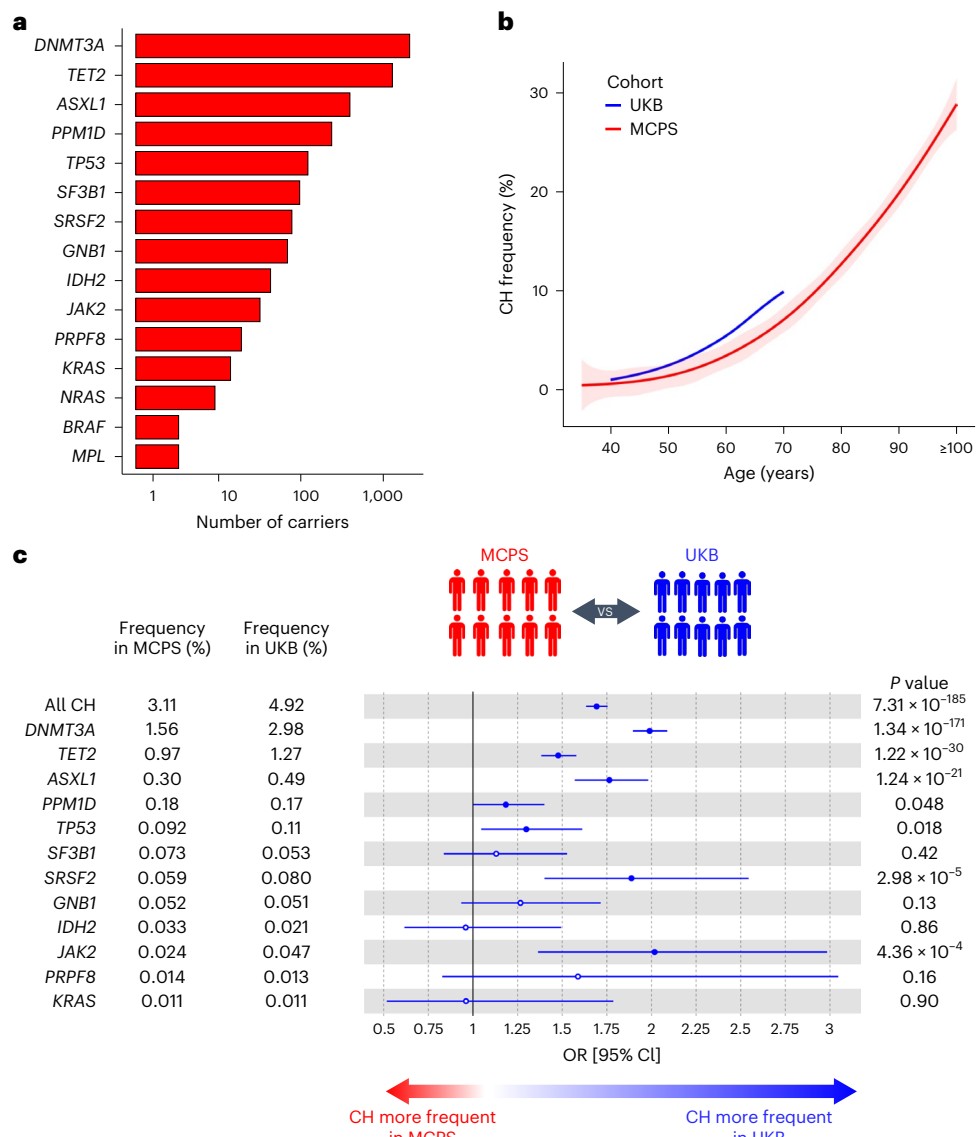

**Fig. 1 | Frequency of CH in the MCPS and UKB. a**, Number of individuals for each CH driver gene identified in MCPS. Driver CH genes are ranked from highest to lowest number of individuals. **b**, Frequency of overall CH by age. The center line represents the fitted values from the general additive model with P-spline smooth class; shaded regions, lower and upper bounds of the 95% CIs of the fitted values. MCPS participants aged 100 years or older were included as a single age group. **c**, Inter-population comparison of the frequency of overall CH and gene-specific CH in UKB compared to MCPS. Only CH driver genes identified in at least ten individuals are shown. Odds ratios and unadjusted two-sided *P* values were derived from a logistic regression model with all CH or gene-specific CH as the outcome and study cohort as the predictor, adjusted for age, sex and smoking status. In total, 414,030 UKB and 136,359 MCPS participants for whom smoking status was available were included for analysis. Measures of center represent the odds ratios; error bars, lower and upper bound of the 95% CI of the odds ratios. Solid circles represent significant associations (*P* < 0.05); hollow circles represent non-significant associations (*P* ≥ 0.05).

CH in an intra-population logistic regression with CH as the outcome and proportion of European genome (continuous variable) as the main predictor adjusted for age, sex and smoking (beta = 0.84, 95% CI = [0.66, 1.03], $P = 7.35 \times 10^{-19}$) and also with increased risk of *DNMT3A*-CH (beta = 1.15, 95% CI = [0.89, 1.41], $P = 4.52 \times 10^{-18}$), *ASXL1*-CH (beta = 1.55, 95% CI = [0.97, 2.14], $P = 1.76 \times 10^{-7}$) and *SRSF2*-CH (beta = 2.17, 95% CI = [0.88, 3.45], $P = 9.64 \times 10^{-4}$; Fig. 2d and Supplementary Table 9). Furthermore, European ancestry of the haplotype block and gene locus in which the CH gene resides also modestly correlated with the frequency of *DNMT3A*-CH, *SF3B1*-CH and *SRSF2*-CH (Supplementary Fig. 3a,b and Supplementary Table 10).

**Association between telomere length and CH frequency**

Emerging evidence from European populations suggests that CH risk is influenced by telomere length[15,26–28]. We therefore sought to assess whether the higher frequency of CH among MCPS participants with a higher proportion of European genome could be, in part, explained by differences in telomere length. First, we computed the leukocyte telomere length (LTL) for 9,598 MCPS participants with whole-genome sequencing (WGS) data available using coverage-normalized TelSeq[29] measurements as previously described[28] (see Methods and Extended Data Fig. 5a). The WGS-inferred LTL was observed to be negatively associated with age, as expected, with individuals in older age groups having shorter LTLs than individuals in younger age groups (Extended Data Fig. 5b). We performed LTL genome-wide association studies (GWAS) with these 9,598 MCPS participants with WGS-inferred LTL data available. In total, 19 loci met a suggestive significance threshold ($P < 5 \times 10^{-6}$), of which six loci were genome-wide significant ($P < 5 \times 10^{-8}$; Fig. 3a and Supplementary Table 11). All genome-wide significant loci (*TERC*, *NAF1*, *TERT*, *TERF1*, *STN1* (*OBFC1*) and *TINF2*) were previously reported[30,31].

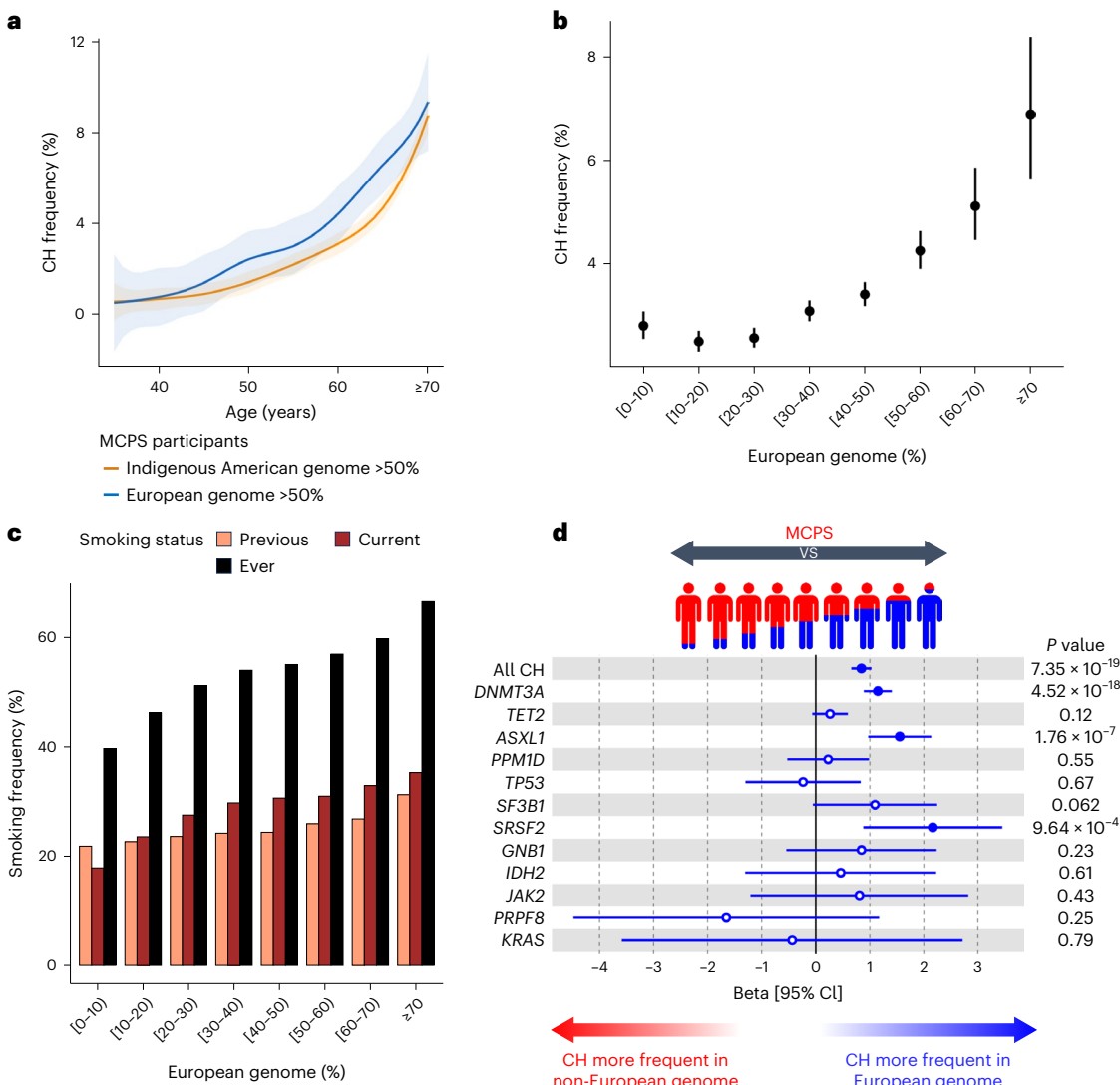

**Fig. 2 | Ancestry association with CH frequency in the MCPS. a**, Frequency of all CH by age among individuals with >50% Indigenous American ancestry and individuals with >50% European ancestry. Ancestry genome proportion was inferred with RFMix2.0 software. The center line represents the fitted values from the general additive model with P-spline smooth class and the shaded region represents the lower and upper bound of the 95% CI of the fitted values. Individuals aged 70 years or above were included as a single age group. **b**, Frequency of all CH by binned proportion of European genome. Measures of center represent the observed CH frequencies; error bars, lower and upper bound of the 95% CI of CH frequencies. In total, 134,297 individuals with RFMix-inferred ancestry available were included for analysis. **c**, Frequency of smoking (previous, current or ever) by binned proportion of European genome. **d**, Intra-population comparison of the frequency of all CH and gene-specific CH among individuals with varying degrees of European and non-European (Indigenous American and African) genome. Only CH driver genes identified in at least ten individuals are shown here. Beta coefficients and unadjusted two-sided $P$ values were derived from a logistic regression model with all CH or gene-specific CH as the outcome and with the proportion of European genome as the predictor, adjusted for age, sex and smoking status. In total, 134,255 individuals with RFMix-inferred ancestry and smoking status available were included for analysis. Measures of center represent the beta coefficients; error bars, lower and upper bound of the 95% CI of the beta coefficients. Full circles represent significant associations ($P < 0.05$); hollow circles represent non-significant associations ($P \geq 0.05$).

We subsequently computed LTL polygenic risk scores (PRSs) on the remaining 126,803 MCPS participants using a genome-wide approach implemented with the PRS-cs software[32] (see Methods). We observed that MCPS participants who had a higher proportion of European ancestry had longer genetically predicted LTLs based on our MCPS-derived PRS (Fig. 3b). For example, the LTL PRS of MCPS participants with ≥70% European genome was significantly greater than that of those with <10% European genome, based on PRS derived from MCPS (median, 3.84 vs −0.24, $P_{Wilcox} < 2.2 \times 10^{-16}$). This finding was also true when using LTL PRSs derived from an ancestry-diverse population (TOPMed[30]) and a European-majority population (UKB[31]; Extended Data Fig. 5c,d). Validation of the MCPS-derived LTL PRSs across the different ancestry groups in UKB demonstrated the highest

correlation between LTL PRSs and WGS-inferred LTL among Admixed Americans relative to other ancestry groups (Extended Data Fig. 5e,f).

Modeling LTL as a risk factor of CH revealed that longer genetically predicted LTL was associated with overall CH, as well as *DNMT3A*-CH and *TET2*-CH as previously reported[15] (overall CH, beta = 0.029, 95% CI = [0.017, 0.041], $P = 4.19 \times 10^{-6}$; *DNMT3A*-CH, beta = 0.040, 95% CI = [0.023, 0.057], $P = 5.10 \times 10^{-6}$; *TET2*-CH, beta = 0.036, 95% CI = [0.014, 0.058], $P = 1.43 \times 10^{-3}$; Fig. 3c and Supplementary Table 12). Additionally, longer genetically predicted LTL was found to be associated with *SRSF2*-CH in this study (beta = 0.11, 95% CI = [0.026, 0.20], $P = 0.011$). Conversely, shorter genetically predicted LTL was associated with *PPM1D*-CH among MCPS participants (beta = −0.055, 95% CI = [−0.106, −0.004], $P = 0.036$). We observed that genetically

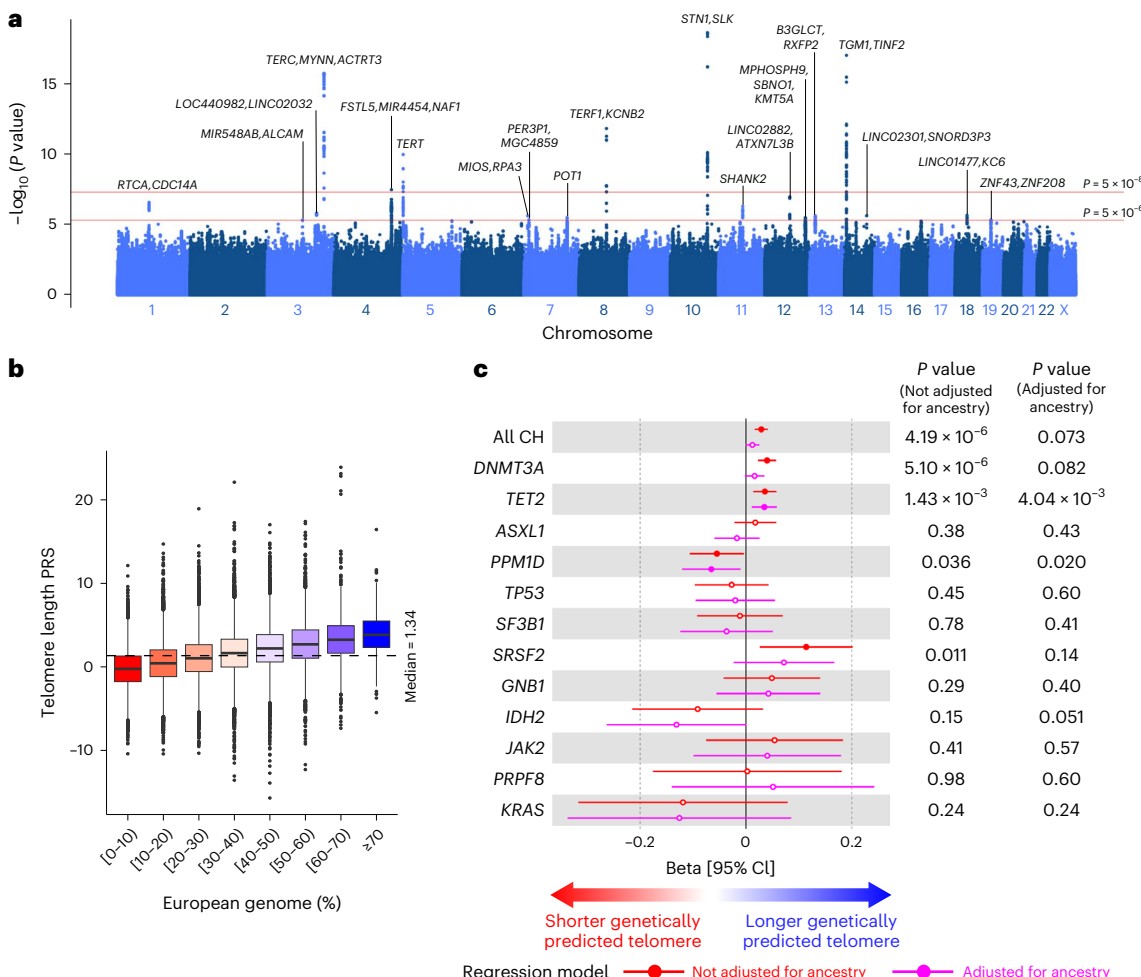

**Fig. 3 | Telomere length association with ancestry and CH in the MCPS. a**, Manhattan plot representing the common germline variants with MAF ≥ 1% included for LTL GWAS in 9,598 MCPS participants with WGS data available. Unadjusted two-sided *P* values on the *y* axis were derived from linear regression implemented using REGENIE software. The most significant variant (smallest *P* value) is annotated for each suggestive locus (*P* < 5 × 10⁻⁶). **b**, Distribution of LTL PRSs across individuals with varying degrees of European genome. PRSs were built using LTL GWAS summary statistics from 9,598 participants for whom WGS data were available and subsequently computed for the remaining 126,803 participants. Boxplots represent the median, first and

third quartiles; whiskers represent 1.5 times the interquartile range. **c**, Association between CH or gene-specific CH (outcome) with LTL PRS (predictor) adjusted for age, sex and smoking status. In total, 124,659 individuals who were not included in the LTL GWAS in **a** and with proportion of European genome and smoking status data available were included for analysis here. Only CH genes identified in at least ten individuals are shown. Beta coefficients and *P* values were derived from a logistic regression model. Measures of center represent the beta coefficients; error bars, lower and upper bound of the 95% CI of the beta coefficients. Full circles represent significant associations (*P* < 0.05); hollow circles represent non-significant associations (*P* ≥ 0.05).

---

predicted LTL remained significant for *TET2*-CH and *PPM1D*-CH after including ancestry as a covariate in our model.

To test whether the higher frequency of CH we had observed in individuals with a higher proportion of European ancestry was related to LTL, we included the LTL PRS as an additional covariate in our logistic regression model. Only a mild attenuation in the association between European ancestry and CH risk was observed (overall CH, beta = 0.80, 95% CI = [0.60, 0.99] vs 0.73 [0.52, 0.94]; *DNMT3A*-CH, beta = 1.12, 95% CI = [0.84, 1.39] vs 1.02 [0.73, 1.32]; *SRSF2*-CH, beta = 2.17, 95% CI = [0.81, 3.52] vs 1.77 [0.31, 3.22] for the model without vs with LTL PRS as a covariate, respectively; Supplementary Table 13).

## Genome-wide common variant associations with CH

To further explore the contribution of common genetic variants (minor allele frequency (MAF) ≥ 1%) to CH risk, we performed GWAS in MCPS participants to evaluate both overall CH and driver-gene-specific CH in addition to splicing factor CH (*SF3B1* and *SRSF2* analyzed in combination). MCPS germline variants included for GWAS were genotyped with

SNP arrays and subsequently imputed with the TOPMed reference panel (see Methods). Association analysis evaluating overall CH, including 4,249 cases and 132,152 controls, identified two genome-wide significant loci (*P* < 5 × 10⁻⁸ for lead variant) among MCPS participants, namely the *TERT* and *TCL1B* loci (Fig. 4a and Table 1). The *TERT* locus has shown the strongest association with CH in all European studies to date[15–17]. The lead *TERT* variant in our MCPS study was rs2853677, with the G allele (MAF, 23%) conferring an increased risk of overall CH (OR = 1.31, 95% CI = [1.24, 1.37], *P* = 1.62 × 10⁻²⁴), similar to what was reported in Europeans[15]. Conditional analyses at the *TERT* locus based on the lead variant in this study or 22 previously reported significant variants[16] at this locus did not identify additional novel independent variants (Supplementary Table 14).

In addition to replicating the well-established *TERT* locus association, we also discovered two novel low-frequency risk variants located 82.9 kb (rs968294563, MAF = 1.24%) and 115 kb (rs187319135, MAF = 1.06%) upstream of *TCL1B* (T cell leukemia/lymphoma protein 1B) on chromosome 14. These were in strong linkage

disequilibrium (D′ > 0.99, $r^2$ = 0.86), and conditional analysis did not reveal independent signals. In addition, rs187319135 (T allele) also displayed genome-wide significant associations with *TET2*-CH and *SF3B1+SRSF2*-CH (OR = 3.59, 95% CI = [2.81, 4.60], $P$ = 3.70 × 10⁻¹⁸ and OR = 6.92, 95% CI = [4.24, 11.3], $P$ = 7.34 × 10⁻¹⁰, respectively, Extended Data Fig. 6a,b). At the nominally significant threshold ($P$ < 0.05), it was also associated with an increased risk of six other gene-specific CH subtypes but a decreased risk of *DNMT3A*-CH (OR = 0.31, 95% CI = [0.18, 0.55], $P$ = 1.44 × 10⁻⁶; Fig. 4b and Supplementary Table 15).

Notably, *TCL1B* is located adjacent to *TCL1A* and, although CH-associated risk variants (rs10131341 and rs2887399) have been previously identified at the *TCL1B-TCL1A* locus in European population analyses[15,16,33], rs187319135 is highly enriched in Indigenous American (Mexican) versus European ancestry (MCPS Variant Browser, https://rgc-mcps.regeneron.com/home (2023); MAF = 1.62% and MAF = 0.06%, respectively). In MCPS, the minor allele of rs187319135 was observed to be frequently co-inherited alongside the major alleles of rs10131341 (D′ = 0.95) and rs2887399 (D′ = 0.97; Supplementary Figs. 4a and 5a); however, the overall allele correlation was low ($r^2$ = 0.0024 and $r^2$ = 0.0016, respectively) because rs187319135 is rarer than rs10131341 and rs2887399. Correspondingly, we observed independence of these signals (Supplementary Figs. 4b–e and 5b–e).

In addition to the *TCL1B-TCL1A* locus, an additional novel locus was identified through GWAS of *ASXL1*-CH in MCPS (Table 1). The lead variant associated with *ASXL1*-CH was rs2958593 at the *CSGALNACT1* locus (Extended Data Fig. 6c), and the T allele (MAF = 16%) conferred an increased risk exclusively of *ASXL1*-CH (OR = 1.63, 95% CI = [1.39, 1.92], $P$ = 2.37 × 10⁻⁸; Extended Data Fig. 7). It is noteworthy that the *CSGALNACT1* locus was marginally genome-wide significant and was identified in a subgroup (gene-specific CH) analysis and therefore may not withstand full Bonferroni adjustment.

Many of the reported lead variant associations with CH in European populations were replicated among MCPS participants. Among previously reported lead CH variants[15–17,34], 8 out of 24 (33%) for overall CH, 12 out of 23 (52%) for *DNMT3A*-CH, 4 out of 7 (57%) for *TET2*-CH, 1 out of 2 (50%) for *ASXL1*-CH and 2 out of 4 (50%) for *JAK2*-CH were nominally significant ($P$ < 0.05) among MCPS participants (Supplementary Figs. 6 and 7a–d and Supplementary Table 16).

**Exome-wide rare variant associations with CH**

We next sought to identify rare germline variants associated with CH in the MCPS cohort using exome-wide association analysis[35]. Common risk variants identified from GWAS were largely recapitulated in our whole-exome association analysis at exome-wide significance ($P$ < 1 × 10⁻⁸; Supplementary Table 17). Focusing on rare variants (MAF < 1%), we identified a rare variant in the *TCL1B* promoter (rs774615666, MAF = 0.33%) that increased risk for *TET2*-CH (OR = 4.23,

95% CI = [2.92, 6.13], $P$ = 2.08 × 10⁻¹⁰) and *SF3B1+SRSF2*-CH (OR = 16.8, 95% CI = [9.72, 29.2], $P$ = 1.71 × 10⁻¹³; Fig. 4c, Table 2 and Extended Data Fig. 8). Notably, rs774615666 was associated, at nominal or suggestive significance thresholds ($P$ < 0.05 and $P$ < 1 × 10⁻⁶, respectively), with increased risk of *PPM1D*-CH, *IDH2*-CH, *JAK2*-CH, *PRPF8*-CH, *KRAS*, *SF3B1*-CH and *SRSF2*-CH but a decreased risk of *DNMT3A*-CH (Fig. 4d and Supplementary Table 18). Within the MCPS cohort, rs774615666 was unique to Indigenous American (Mexican) ancestry (MCPS Variant Browser, https://rgc-mcps.regeneron.com/home (2023); MAF = 0.50% vs MAF = 0% for European ancestry) and was still significant after conditioning on the previously identified *TCL1A* upstream (rs10131341) and promoter (rs2887399) variants for *TET2*-CH (OR = 4.10, 95% CI = [2.83, 5.93], $P$ = 4.56 × 10⁻¹⁰) and *SF3B1+SRSF2*-CH (OR = 17.1, 95% CI = [9.86, 29.6], $P$ = 1.47 × 10⁻¹³). Notably, rs774615666 was in high linkage disequilibrium with the GWAS identified rs187319135 (D′ = 0.95, $r^2$ = 0.32) in MCPS, yet we observed residual association with overall CH and gene-specific CH after conditioning these two risk variants on each other (Extended Data Fig. 9a–d) and after further stratifying individuals into their respective rs187319135 and rs774615666 genotypes (Supplementary Fig. 8). Taken together, the ancestry-specific variants rs187319135 and rs774615666 may be partly independent of one another, and functional assays, such as Hi-C, may help elucidate the causal variant(s).

**Cross-ancestry meta-analysis of CH**

Cross-ancestry GWAS meta-analysis across UKB (imputed with the Haplotype Reference Consortium (HRC) and UK10K + 1000 Genomes panel[36]; see Methods) and MCPS participants yielded five novel CH-associated loci: one for overall CH and four for driver-gene-specific CH (Fig. 5a and Supplementary Table 19). In total, 16 loci were detected for overall CH, including a novel association in the *MYCN* locus (rs12471506, C allele, OR = 1.06, 95% CI = [1.04, 1.08], $P$ = 5.32 × 10⁻⁹). Summary statistics from the multi-ancestry TOPMed cohort consisting of 4,141 CH individuals and 61,263 controls were available for overall CH for replication analysis[17,37]. The inclusion of TOPMed in our cross-ancestry meta-analysis strengthened the association with CH for 10 out of the 14 leading variants (Supplementary Table 20).

Driver-gene-specific CH GWAS for *DNMT3A* revealed 19 loci, of which two were novel: *MEIS1* (rs2280334, C allele; OR = 0.93, 95% CI = [0.91, 0.95], $P$ = 2.21 × 10⁻⁸) and *PARP11-CCND2* (rs582975, T allele; OR = 0.93, 95% CI = [0.91, 0.95], $P$ = 1.71 × 10⁻⁸; Fig. 5b). *MEIS1* is a gene with critical roles in both normal and malignant hematopoiesis[38], whereas *CCND2* is recurrently mutated in acute myeloid leukemia[39]. PARP11 has a role in inflammatory responses by regulating IFN-I signaling[40]. In *TET2*-CH, five loci were detected including a novel association within the *UBE2G1-SPNS3* locus (rs73332852, T allele; OR = 1.50, 95% CI = [1.33, 1.69], $P$ = 6.11 × 10⁻¹¹; Fig. 5c). For *ASXL1*-CH, two loci were

**Fig. 4 | GWAS and ExWAS of all CH and gene-specific CH in the MCPS.**
**a**, Manhattan plot representing the common germline variants with MAF ≥ 1% included for CH GWAS. Unadjusted two-sided $P$ values on the $y$ axis were derived from Firth logistic regression implemented using REGENIE software. One new association at the *TCL1A-TCL1B* locus from MCPS (red) was identified as genome-wide significant ($P$ < 5 × 10⁻⁸), with the nearest gene of the leading genetic polymorphism annotated. One previously reported association from the European population at the *TERT* locus is indicated in blue. **b**, rs187319135 identified as genome-wide significant from overall CH, and *TET2*-CH and *SF3B1+SRSF2*-CH. Overall and gene-specific CH risk estimates conferred by the minor allele (T) are shown here for 136,401 MCPS participants. ORs and unadjusted two-sided $P$ values were derived from Firth logistic regression implemented using REGENIE software. Risk estimates are shown when both CH or gene-specific CH cases and controls have minor allele count (MAC) ≥ 1. Risk estimates for UKB were not included owing to the absence of risk allele (MAC = 0) in CH individuals. Measures of center represent the ORs; error bars, lower and upper bounds of the 95% confidence intervals of the ORs. Full circles represent

significant associations ($P$ < 0.05); hollow circles represent non-significant associations ($P$ ≥ 0.05). **c**, Manhattan plot representing the rare (MAF < 1%) and common germline variants included for *TET2*-CH ExWAS. $P$ values on the $y$ axis were derived from Firth logistic regression implemented using REGENIE software. Rare *TCL1B* promoter variant rs774615666 (red) was identified as exome-wide significant ($P$ < 1 × 10⁻⁸). Common *TERT* variant indicated in blue. **d**, rs774615666 identified as exome-wide significant from *TET2*-CH and *SF3B1+SRSF2*-CH. Overall and gene-specific CH risk estimates conferred by the minor allele (T) are shown here for 136,149 MCPS and 416,118 UKB participants. Risk estimates are shown when both CH or gene-specific CH cases and controls have MAC ≥ 1. ORs and unadjusted two-sided $P$ values were derived from Firth logistic regression implemented using REGENIE software for MCPS and Fisher's exact test for UKB. Measures of center represent the ORs; error bars, lower and upper bound of the 95% confidence interval of the ORs. Full circles represent significant associations ($P$ < 0.05); hollow circles represent non-significant associations ($P$ ≥ 0.05).

identified including the novel *PPIA-H2AZ2-PURB* locus (rs13245012, T allele; OR = 1.19, 95% CI = [1.13, 1.27], $P = 1.92 \times 10^{-9}$; Fig. 5d). One additional novel locus (*PDXDC1*) was discovered from our UKB-only GWAS for *DNMT3A*-CH (rs28545123, C allele; OR = 0.90, 95% CI = [0.87, 0.93], $P = 1.80 \times 10^{-8}$).

As the rare variants found to be associated with CH were ancestry-specific, it was not surprising that cross-ancestry exome-wide association meta-analysis did not identify additional risk variants. Using a gene-level collapsing test, which aggregates all qualifying rare germline variants for a given gene, may increase statistical power by testing

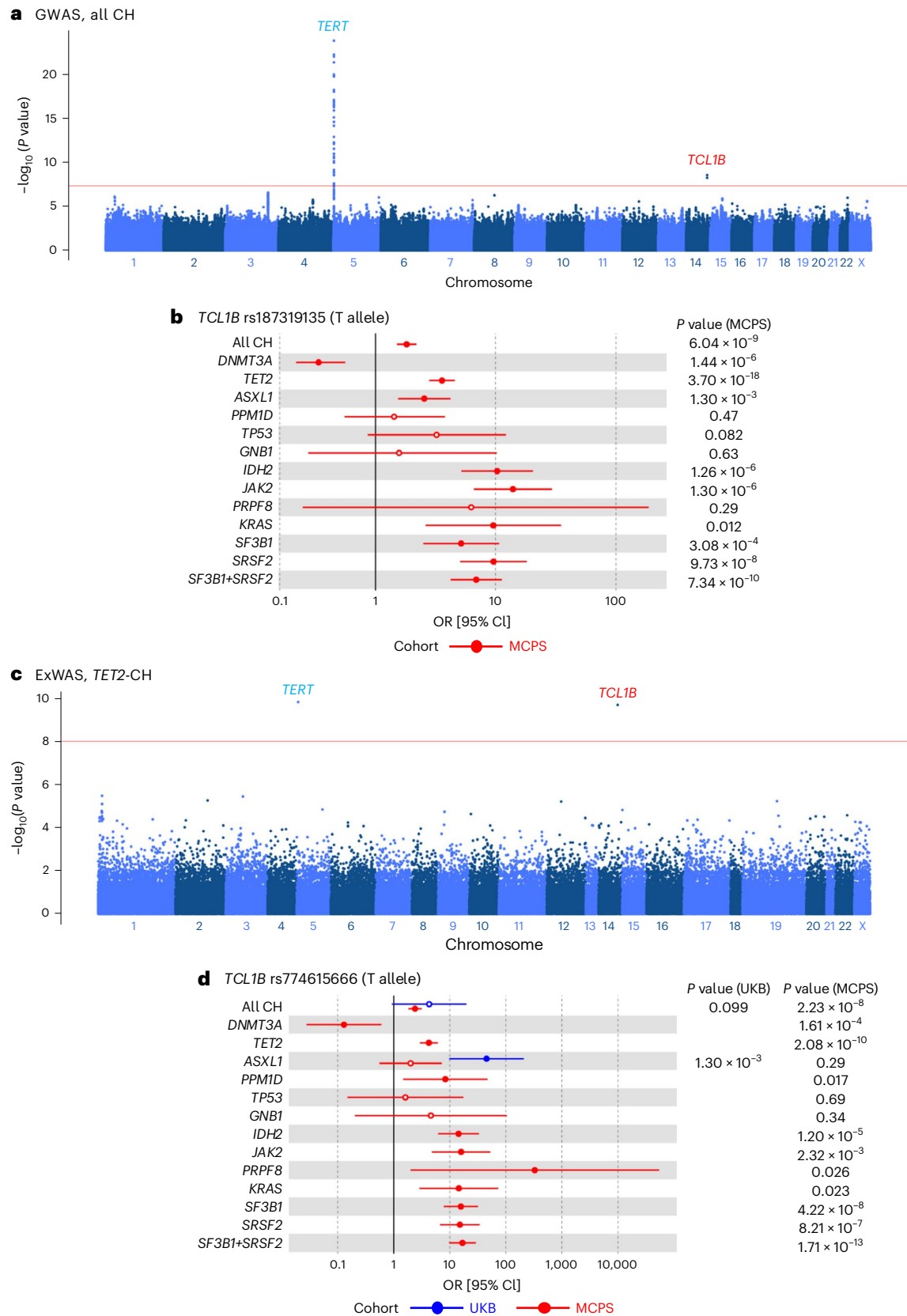

**Table 1 | GWAS summary statistics of leading genetic polymorphisms (smallest P value at each genome-wide significant locus) in the Mexico City Prospective Study**

| CH | rsID | Gene | Reference allele | Effect allele | MAF | OR [95% CI] | P value |
|---|---|---|---|---|---|---|---|
| All CH | rs2853677 | TERT | A | G | 0.225 | 1.31 [1.24, 1.37] | $1.62 \times 10^{-24}$ |
| All CH | rs968294563 | TCL1B | CT | C | 0.012 | 1.78 [1.49, 2.13] | $2.88 \times 10^{-9}$ |
| DNMT3A | rs2853677 | TERT | A | G | 0.225 | 1.40 [1.31, 1.50] | $2.83 \times 10^{-21}$ |
| TET2 | rs2736099 | TERT | G | A | 0.195 | 1.43 [1.30, 1.56] | $2.54 \times 10^{-13}$ |
| TET2 | rs187319135 | TCL1B | C | T | 0.011 | 3.59 [2.81, 4.60] | $3.70 \times 10^{-18}$ |
| ASXL1 | rs2958593 | CSGALNACT1 | C | T | 0.164 | 1.63 [1.39, 1.92] | $2.37 \times 10^{-8}$ |
| SF3B1+SRSF2 | rs187319135 | TCL1B | C | T | 0.011 | 6.92 [4.24, 11.3] | $7.34 \times 10^{-10}$ |

Test statistics were derived from Firth logistic regression implemented using REGENIE software, adjusted for age, sex and first ten genetic principal components. CH driver genes and GWAS loci are in italics.

**Table 2 | ExWAS summary statistics of leading genetic polymorphisms (smallest P value at each exome-wide significant locus) with a MAF of <1% in the Mexico City Prospective Study**

| CH | rsID | Gene | Reference allele | Effect allele | MAF | OR [95% CI] | P value |
|---|---|---|---|---|---|---|---|
| TET2 | rs774615666 | TCL1B | C | T | 0.0032 | 4.23 [2.92, 6.13] | $2.08 \times 10^{-10}$ |
| SF3B1+SRSF2 | rs774615666 | TCL1B | C | T | 0.0032 | 16.8 [9.72, 29.2] | $1.71 \times 10^{-13}$ |

Test statistics were derived from Firth logistic regression implemented using REGENIE software, adjusted for age, sex and first ten genetic principal components. CH driver genes and ExWAS loci are in italics.

the combined effect of rare variants[35,41]. As in previous work[35], we used 11 different qualifying variant models to maximize discovery across potential genetic architectures (see Methods). Although gene-level collapsing analysis in the MCPS cohort did not identify any genes significantly associated with CH, the MCPS–UKB meta-analysis replicated the previously reported CHEK2 association with overall CH (flexible-damaging qualifying variant model; OR = 1.62, 95% CI = [1.43, 1.84], $P = 9.50 \times 10^{-13}$) and with DNMT3A-CH (flexible-damaging qualifying variant model; OR = 1.77, 95% CI = [1.50, 2.07], $P = 3.18 \times 10^{-11}$; Extended Data Fig. 10a,c,e and Supplementary Table 21). Scrutiny of the qualifying rare variants in CHEK2 revealed that the majority were ancestry-specific (Extended Data Fig. 10b,d,f and Supplementary Table 22). Notably, CHEK2 c.1100delC constituted 72% of all CHEK2 qualifying variants in UKB participants, but only 4.7% in MCPS participants. Taken together, cross-ancestry meta-analysis yielded several novel CH susceptibility loci driven by common variants.

## Discussion

We present, to the best of our knowledge the largest analysis of CH in a non-European population to date and leverage genetic ancestry to uncover novel insights into the etiology of this common age-related phenomenon. We discover that the frequency of CH among 136,401 individuals from Latin America (MCPS cohort) is 1.6-fold lower than the frequency among 416,118 individuals in UKB (3.12% vs 4.92%, respectively), a difference that persisted in age-matched and sex-matched participants (2.87% vs 4.55%). Although a previous study reported CH frequency to be similar across different ancestries[16], smaller but more ancestry-diverse studies suggested a lower frequency among self-reported Latinos or Hispanics than among Europeans[4,17,22], consistent with our findings. Notably, none of the CH genes were more common in MCPS. Particularly striking was the substantially lower frequency of JAK2-CH in MCPS than in UKB, and it is tenable that this underlies the lower prevalence of myeloproliferative neoplasms, most of which are JAK2-driven, in Hispanic populations[42,43].

A potential limitation of comparing CH frequency between MCPS and UKB is the technical differences in the sequencing and bioinformatics pipelines[44,45]. This was mitigated as far as possible by WES DNA libraries being prepared by the same exome capture kit (IDT xGen

v.1 capture kit), sequenced on the same platform (NovaSeq 6000) with the same sequencing mode (75 bp paired-end mode) by the same sequencing provider (Regeneron Genetics Centre)[23,46,47] and subjected to the same germline (Illumina DRAGEN Bio-IT Platform Germline Pipeline v.3.0.7) and somatic (MuTect2) variant calling pipeline[20,21,48]. Collectively, this resulted in similar somatic variant allele frequencies and coverage profiles observed between MCPS and UKB samples. Although technical variation between MCPS and UKB affecting CH detection cannot be entirely ruled out, we replicated the difference in CH frequency within the MCPS study itself, in which we observed that increased CH risk was associated with a higher fraction of European than Indigenous American ancestry.

Differences in the prevalence of diseases between populations are the result of variation in both environmental exposures and genetics[49]. Although we have accounted for known risk factors in our analysis (age, sex and smoking), residual confounding may nevertheless still be present; for example, because of unmeasured components of smoking including the duration of smoking and the number of cigarettes smoked daily. Additional environmental factors are also likely to have a role, and metformin has recently been shown to reduce the clonal fitness of DNMT3A-mutant HSPCs[50]. It is noteworthy that for a subset of driver genes (TET2, PPM1D, TP53, JAK2 and MPL), although our inter-population analysis showed increased UKB frequency, our intra-population analysis among MCPS participants did not demonstrate higher frequencies among individuals with a higher proportion of European genome. This may suggest that environmental risk factors, which would be expected to vary more between MCPS and UKB than within MCPS, have a larger role in determining the frequency of clones driven by mutations in these genes[51,52]. As novel CH risk factors are revealed over the coming years, it will be important to assess whether they explain population differences in CH frequency.

Our germline association analyses with CH in MCPS and UKB, and their meta-analysis, identified seven novel loci, increasing the number of germline associations of CH from 51 to 58. Additionally, we identified ancestry-specific, low-frequency genetic variants in the TCL1B locus that were independent of the TCL1A upstream variant (rs2887399) previously identified in European studies[16] and shown to drive clonal expansion in TET2 and ASXL1-mutant HSPCs[33]. Furthermore, similar

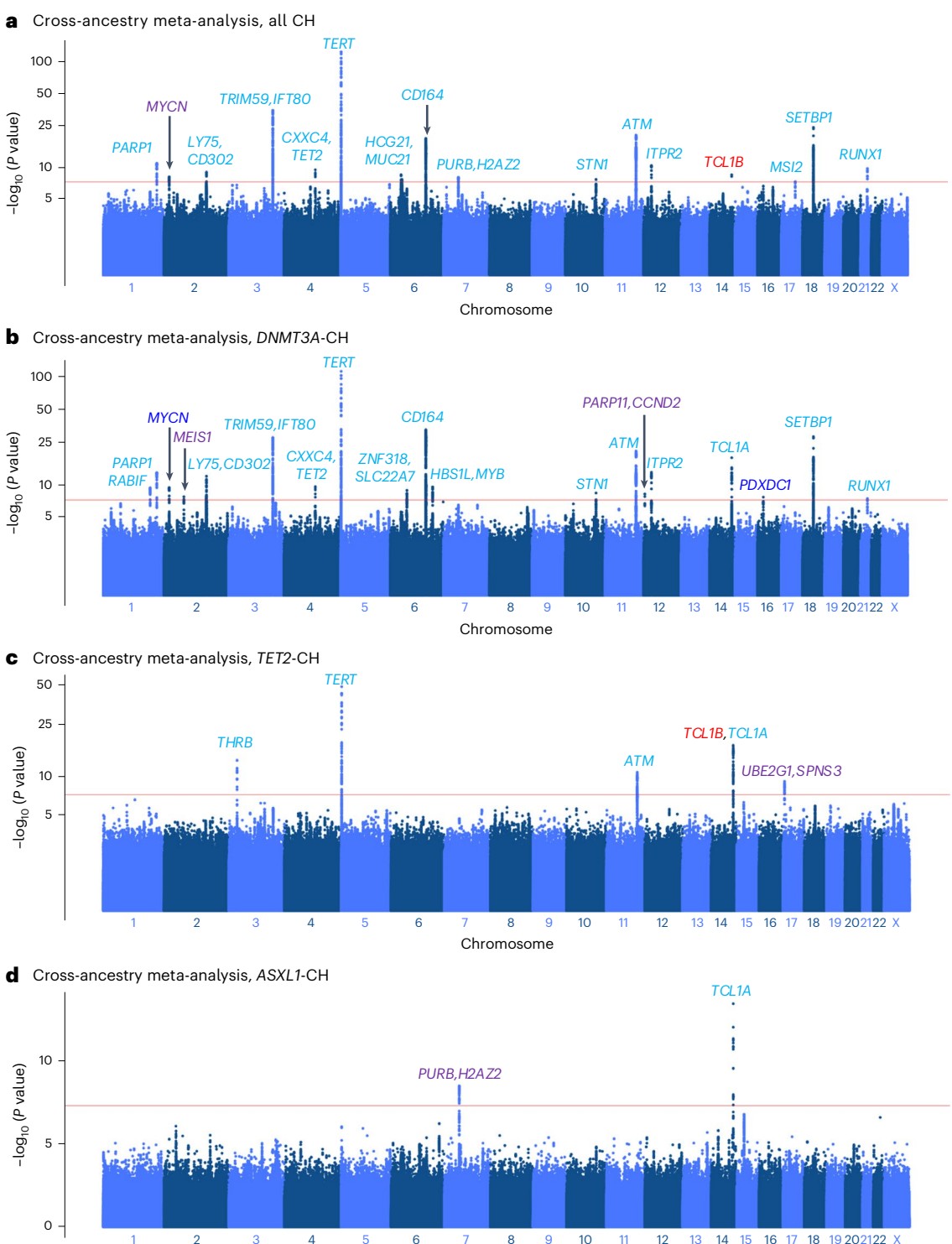

**Fig. 5 | Cross-ancestry GWAS meta-analysis of all CH and gene-specific CH in the MCPS and UKB. a–d,** Manhattan plots representing the common germline variants with MAF ≥ 1% included for GWAS in MCPS and UKB Europeans for all CH (**a**), and *DNMT3A*-CH (**b**), *TET2*-CH (**c**) and *ASXL1*-CH (**d**). Unadjusted two-sided $P$ values on the $y$ axis were derived from the $P$ value-based method implemented using METAL software. Five novel loci from the meta-analysis of MCPS and UKB (purple) were identified as genome-wide significant ($P < 5 \times 10^{-8}$), with the nearest gene of the leading genetic polymorphism annotated for the respective locus. Previously reported associations from European populations are indicated in light blue and novel associations identified in the European population in our study are indicated in dark blue.

to rs2887399, the *TCL1B* upstream (rs187319135) and promoter (rs774615666) risk variants were associated with increased risk to *TET2*-CH and *ASXL1*-CH but decreased risk to *DNMT3A*-CH in MCPS. Both *TCL1B* and *TCL1A* are observed to be aberrantly expressed in T cell leukemia driven by t(14;14)(q11;q32,1), a translocation event that juxtaposes *TCL1B/A* to the α/δT cell receptor locus[53]. Therefore, although it is tempting to implicate TCL1B as a potential driver of clonal expansion—and by extension, CH development—it is worth noting

that *TCL1B* is epigenetically silenced and therefore transcriptionally repressed in HSPCs[33]. Nevertheless, it is plausible that the *TCL1B* risk variants may lead to ectopic expression of the gene. One method of assessing the potential functional impact of GWAS or exome-wide association study (ExWAS) variants is through their linkage to expression quantitative trait loci (eQTL). However, eQTL analysis of the *TCL1B* promoter risk variant in MCPS was hampered by the lack of such non-European-specific variants in publicly available European-majority eQTL datasets. Although there are emerging non-European eQTL resources, these datasets are relatively small[54]. This highlights the urgency of establishing large-scale, non-European population-based resources, including eQTL databases, to allow equitable research in diverse ancestries and communities.

Collectively, we identify substantial differences in the frequency of CH and its subtypes between populations and discover ancestry-specific genetic determinants. The implications of population-level differences in CH frequency include considerations around population-wide or targeted CH screening planning and strategy, resource allocation and risk assessment for CH-related diseases including cardiovascular or other pathologies. Our work demonstrates that the investigation of CH in diverse populations can reveal biological insights, with differential implications for population-level precision medicine and, ultimately, support the advancement of global health equality.

## Online content

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

[1]Centre for Genomics Research, Discovery Sciences, BioPharmaceuticals R&D, AstraZeneca, Cambridge, UK. [2]Department of Haematology, Wellcome-MRC Cambridge Stem Cell Institute, Jeffrey Cheah Biomedical Centre, University of Cambridge, Cambridge, UK. [3]Instituto Tecnológico y de Estudios Superiores de Monterrey, Monterrey, Mexico. [4]Centre for Genomics Research, Discovery Sciences, BioPharmaceuticals R&D, AstraZeneca, Waltham, MA, USA. [5]Faculty of Medicine, National Autonomous University of Mexico, Copilco Universidad, Ciudad de México, Mexico. [6]Clinical Trial Service Unit & Epidemiological Studies Unit, Nuffield Department of Population Health, University of Oxford, Oxford, UK. [7]Department of Haematology, Cambridge University Hospitals NHS Foundation Trust, Cambridge, UK. [8]Clinical Development, Research and Early Development, Respiratory and Immunology (R&I), BioPharmaceuticals R&D, AstraZeneca, Cambridge, UK. [9]These authors contributed equally: Sean Wen, Pablo Kuri-Morales. [10]These authors jointly supervised this work: Roberto Tapia-Conyer, Margarete A. Fabre, Andrew R. Harper, George S. Vassiliou, Jonathan Mitchell. ✉e-mail: gsv20@cam.ac.uk; jonathan.mitchell@astrazeneca.com

## Methods

### Study population

Participants were included from two population-based studies: MCPS and UKB.

MCPS is a prospective cohort of more than 150,000 adults, aged at least 35 years, who were recruited between 1998 and 2004 from the contiguous urban districts of Coyoacán and Iztapalapa in Mexico City[23,55]. Of these participants, WES data are available from 141,046 individuals. UKB is a prospective cohort of approximately 500,000 adults, aged between 40 to 70 years and recruited since 2007[36,46]; WES data are available from 469,809 of those individuals.

All participants provided written informed consent. The MCPS study was approved by the Mexican Ministry of Health, the Mexican National Council for Science and Technology and the University of Oxford, and the UKB study has approval from the North-West Multi-centre Research Ethics Committee (11/NW/0382).

### WES pre-processing and variant calling

WES data for genomic DNA from MCPS and UKB were generated by the Regeneron Genetics Centre as previously described[23,46,47]. Both MCPS and UKB samples were prepared with the same production pipeline. Specifically, sequencing libraries were generated using the IDT xGen v.1 capture kit and subsequently sequenced on the NovaSeq 6000 platform in 75 bp paired-end mode. The average sequencing depth across UKB and MCPS samples was 57.8× and 57.2×, respectively. This metric was computed as the average alignment coverage over the consensus coding sequence located on the autosomes. Both cohorts also had similar mapping quality with 89.21% and 89.28% of reads, with a MAPQ (mapping quality score) of >40, respectively.

The FASTQ files were subsequently processed at AstraZeneca as previously described[48]. Both MCPS and UKB samples were subjected to the same processing pipeline. Specifically, sequencing reads were demultiplexed using the 10 bp index barcodes with bcl2fastq (v.2.19.0) to obtain the sequencing reads for each sample in FASTQ format. Next, sequencing read alignment to the GRCh38 genome reference and germline variant detection, for exome-wide association analysis and gene-level collapsing analysis, was performed using the Illumina DRA-GEN Bio-IT Platform Germline Pipeline (v.3.0.7). Somatic variant calling was performed using GATK MuTect2 (refs. [20],[21]). A panel of normals was created from 200 of the youngest UKB participants without a hematologic malignancy diagnosis to remove potential recurrent artifacts with GATK *FilterMutectCalls*. The *--orientation-bias-artifact-priors* option was also specified to remove read orientation artifacts based on priors generated with *LearnReadOrientationModel*.

### CH detection

To identify CH driver variants, we first retrieved Mutect2 PASS somatic variants occurring in a previously defined list of 74 genes with leukemogenic driver mutations[17]. They were annotated with the transcript ID, exon number, cDNA change, amino acid change and protein consequence with Ensembl Variant Effect Predictor software[56,57], and filtered according to the previously defined criteria for putative CH drivers based on variant consequence[17] (Supplementary Table 4). Only variants supported by at least three alternate allele reads and with a variant allele frequency of at least 3% but not more than 40% were retained. All subsequent analysis was initially restricted to 15 pre-leukemic driver genes that demonstrated an association between the presence of a putative driver mutation and age, namely *DNMT3A*, *TET2, ASXL1, PPM1D*, *TP53*, *SF3B1*, *SRSF2*, *GNB1*, *IDH2*, *JAK2*, *PRPF8*, *KRAS*, *NRAS*, *BRAF* and *MPL* (Extended Data Fig. 2a,b and Supplementary Table 7)[15]. Based on this CH variant classification, the gene panel consisted of genes with any protein-truncating variants (frameshift, nonsense and splice-site) along the gene body and recurrent hotspot variants (*DNMT3A*, *TET2* and *TP53*), genes with protein-truncating variants at specific exons (*ASXL1* and *PPM1D*) and genes with only recurrent hotspot variants

(*SF3B1*, *SRSF2*, *GNB1*, *IDH2*, *JAK2*, *PRPF8*, *KRAS*, *NRAS*, *BRAF* and *MPL*). Additionally, using identical variant filtering criteria, we extended CH detection to the 58-gene panel[22], and reported and compared downstream association results for both the 15-gene and 58-gene panels.

### Sample selection

In both MCPS and UKB, samples were selected based on: (1) contamination <4% computed by VerifyBAMID software[58]; (2) sex concordant between clinically reported and chromosome X:Y consensus coding sequence coverage ratios; (3) ≥94.15% of consensus coding sequence r22 bases[59] covered with ≥10× coverage; (4) within four s.d. of mean genetic principal components one to four as computed by the *peddy* software (v.0.3.2)[18]; and (5) single-nucleotide polymorphism (SNP) array quality control (genotype missingness of ≤10%). Samples from MCPS were additionally selected based on being within 2 s.d. of the mean read-depth distribution, having no pairs with kinship of >0.45 and probability of ≥0.95 of Admixed American ancestry. Samples from UKB were additionally selected based on having no pairs with kinship of >0.1769, probability of ≥0.95 of European ancestry, no diagnosis of hematological neoplasms before blood sample collection and consent not withdrawn as of April 2024. Kinship and ancestry were inferred using the KING and *peddy* software, respectively[18,60].

For each individual in the study, *peddy* was used to predict their genetic ancestry with a machine-learning classifier trained on 2,504 individuals of known ancestry from the 1000 Genomes project[18,19]. The classifier was trained on a set of approximately 25,000 bi-allelic sites in the 1000 Genomes project, which were also present in the MCPS and UKB WES data, by first performing randomized principal component analysis and then training a support vector machine on the first four principal components. Once trained, the support vector machine classifier was applied to the WES germline variant calls of each individual in our study after being projected onto the principal components calculated from the 1000 Genomes project samples. This generated the most likely ancestry (African, Admixed American, East Asian, European or South Asian) for each individual along with a probability defining the certainty of the prediction.

UKB samples with a pre-blood collection hematological neoplasm diagnosis were ascertained from ICD9 codes (admissions and cancer registry), ICD10 codes (admissions, cancer registry and death registry), National Health Service Read Codes versions 2 and 3 (primary care visits; that is, general practitioner appointments) and self-reported conditions and cancer conditions from verbal interviews (Supplementary Table 3).

After quality control, 136,401 individuals from MCPS and 416,118 individuals from UKB were included in our study (Supplementary Tables 1 and 2).

### Local ancestry inference and intra-population analysis

Proportions of European, Indigenous American and African ancestry were inferred at each genomic interval (window) using the RFMix2.0 software as previously described[23,24]. In total, 39,861 genomic intervals were defined by RFMix2.0. The association between European ancestry and CH frequency among MCPS participants (intra-population) was assessed at three levels (global, haplotype and gene) using logistic regression with continental ancestry fraction as the exposure and CH as the outcome. For a given CH driver gene, the haplotype-level and gene-level ancestry were defined using the RFMix-defined genomic intervals overlapping with the haplotype and gene region, respectively, in which the CH driver gene resides. Thus, the gene-level interval(s) is contained within the haplotype-level intervals. The haplotype-level and gene-level ancestry estimates were averaged across the two alleles for each individual and categorized into European homozygous, European heterozygous and non-European (America or Africa) homozygous based on European ancestry thresholds of ≥95%, 45–55% and ≤5%, respectively. The association between European ancestry

at the haplotype or gene level and gene-specific CH frequency was assessed using an additive model whereby non-European homozygous was the reference group. Age, sex and smoking status were included as covariates, and logistic regression was performed using the *glm* function as implemented by the *stats* package in R (v.4.2.2).

## Telomere length PRS

LTL was inferred in 9,602 individuals for whom WGS data were available[23] using coverage-normalized TelSeq measurements[28,29]. Specifically, TelSeq was used to infer telomere length from WGS data and was further normalized with sample-specific sequencing coverage. Of the 9,602 individuals with WGS data, 9,598 individuals with LTL within ±3 s.d. were included for LTL GWAS. REGENIE[61] was used for genetic association analysis, with age and sex and the first ten genetic principal components included as covariates as described above. No evidence of genomic inflation was observed (inflation factor = 1.07).

Variants with MAF ≥ 1% and an imputation score of ≥0.3 were retained and the GWAS summary statistics of retained variants were subsequently used to compute the LTL PRSs in the remaining 126,803 MCPS participants. Specifically, the posterior SNP effect sizes under continuous shrinkage prior were computed using the GWAS summary statistics and external Admixed American linkage disequilibrium reference panel using PRS-cs software[32]. The computed effect sizes and MCPS imputed genotype files in PGEN format were used as input in PLINK2 to compute the PRS for each individual in MCPS with the *cols=scoresums* option to obtain the raw (non-averaged) values[62,63].

An LTL PRS was also computed for UKB individuals, using a previously published framework[64]. The UKB imputed genotype files in BGEN format were first filtered for the variants included for PRS-cs above using bgenix software (https://enkre.net/cgi-bin/code/bgen/doc/trunk/doc/wiki/bgenix.md). Multi-allelic variants were subsequently collapsed using SQLite. Specifically, for each multi-allelic variant identified, the variant with the alternative and reference alleles that match the variant included for PRS-cs above was retained. The filtered and multi-allelic-collapsed BGEN files were subsequently converted to PGEN format with PLINK2. Finally, the effect sizes computed with PRS-cs above and imputed genotype file in PGEN format were used as input in PLINK2 to compute the PRS for each individual in UKB.

The MCPS-derived LTL PRS computed in UKB individuals were subsequently used to validate the PRS across the five main ancestry groups in UKB: Admixed American, African, East Asian, European and South Asian[65]. To this end, the effect size of MCPS-LTL PRS as a predictor of the WGS-inferred LTL was computed in a linear regression model, adjusted for age, sex, smoking status and the first four genetic principal components. Furthermore, a second model in which LTL PRS was excluded was built. The percentage improvement in $R^2$ in the first model compared to the second model in each ancestry group was then used as a metric in addition to the effect size returned by the first model to assess the specificity of the MCPS-derived LTL PRS (Extended Data Fig. 5e,f).

Additional LTL PRS were computed in MCPS participants using independent variants reported from published telomere length GWAS in a multi-ancestry cohort (TOPMed; effect size of Hispanic/Latinos from Table 1 of ref. 30) and European-majority cohort (UKB; https://github.com/siddhartha-kar/clonal-hematopoiesis/blob/main/mendelian_randomization/tl.txt)[15,31].

## Genetic association analyses

For MCPS, GWAS and ExWAS, respectively of germline variants with overall CH and gene-specific CH were performed using REGENIE software[61]. In brief, the individuals were genotyped using Illumina Global Screening Array (v.2) beadchip as previously described[23]. Sample-level quality control was performed to remove samples with genotype missingness of >10% and related samples. Variant-level quality control was performed to remove non-autosomal variants, variants

with missingness of ≥2%, variants with C>G, G>C, A>T or T>A base changes, variants in long-range linkage disequilibrium regions and insertions or deletions, and to retain variants with MAF ≥ 1%, with a Hardy–Weinberg equilibrium *P* value of <10⁻⁶ and variants pruned for linkage disequilibrium $r^2 < 0.1$ (windows of 50 SNPs and a size step of five SNPs). The remaining variants were used in step one of REGENIE. In step one of REGENIE, a whole-genome regression model was fitted using genotyped variants to each CH phenotype, and a set of genomic leave-one-chromosome-out predictions was returned. The model was then fitted separately for each of the 22 autosomes and chromosome X in step two of REGENIE. Specifically, the germline variants (imputed with TOPMed reference panel)[23] and WES-identified variants[41] for GWAS and ExWAS, respectively, were tested for association with each CH phenotype using Firth logistic regression based on the additive model. Age, sex and the first ten genetic principal components were included as covariates in both steps of REGENIE, and the leave-one-chromosome-out predictions were additionally included as covariates in step two of REGENIE. The summary statistics from each chromosome were then combined before downstream analysis and were restricted to variants with an imputation quality score of >0.6 and a minor allele count of ≥5 in cases and controls[30]. A filter of MAF ≥ 1% was additionally applied for variants identified from GWAS[15]. Statistically significant GWAS and ExWAS variants were defined with $P < 5 \times 10^{-8}$ and $P < 1 \times 10^{-8}$, respectively, and the leading SNP for each locus was defined with the SNP with the smallest *P* value[15,30,41]. A locus was considered novel when there were no previously reported CH-associated genome-wide significant SNPs falling within ±1 Mb of the leading SNP.

For UKB, GWAS was performed using REGENIE software as described above for MCPS. In brief, individuals were genotyped using the UK Biobank Axiom Array and UKB BiLEVE Axiom Array. Sample-level quality control was performed to retain individuals of European descent and to remove samples with genotype missingness of >5%, samples with non-XX or non-XY chromosome configurations and samples with high heterozygosity. Variant-level quality control was performed to remove variants with missingness of ≥2%, non-autosomal variants, variants with C>G, G>C, A>T or T>A base changes, variants in long-range linkage disequilibrium regions or in commonly inverted regions, insertions or deletions and variants with different allele frequencies between the UK Biobank Axiom Array and UKB BiLEVE Axiom Array (defined with $P < 10^{-12}$ when Fisher's exact test was applied on genotype counts) and to retain variants with MAF ≥ 1%, variants with Hardy–Weinburg equilibrium *P* values of <10⁻⁶ and variants pruned for linkage disequilibrium $r^2 < 0.1$ (windows of 50 SNPs and a size step of five SNPs). The remaining samples and variants were used in step one of REGENIE. Variants imputed with the HRC panel, and additionally, with the UK10K + 1000 Genomes panel for variants not present in the HRC[36], were used in step two of REGENIE. The variant coordinates were additionally converted from GRCh37 to GRCh38 human reference genome assembly with CrossMap[66]. ExWAS was performed on WES-identified variants using Fisher's exact two-sided test based on the allelic model as previously described[35,41].

Gene-level collapsing analysis for MCPS and UKB was performed using Fisher's exact two-sided test as previously described[35,41]. In total, 11 different sets of qualifying variant models were assessed based on the variant effect on protein-coding sequence, MAF from gnomAD and internal test cohort, Rare Exome Variant Ensemble Learner (REVEL) score[67] and Missense Tolerance Ratio (MTR)[68]. The qualifying variant models were 'syn' (synonymous; MAF ≤ 0.005%), 'flexdmg' (non-synonymous; $MAF_{global} \le 0.05\%$, $MAF_{any\ given\ ancestry} \le 0.1\%$, REVEL score ≥ 0.25), 'flexnonsynmtr' (non-synonymous; $MAF_{global} \le 0.05\%$, $MAF_{any\ given\ ancestry} \le 0.1\%$, MTR < 0.78 or MTR centile < 50), 'UR' (non-synonymous; $MAF_{global} = 0\%$, REVEL score ≥ 0.25), 'URmtr' ('UR' and MTR < 0.78 or MTR centile < 50), 'raredmg' (missense; $MAF_{global} \le 0.005\%$, REVEL score ≥ 0.25), 'raredmgmtr' ('raredmg' and MTR < 0.78 or MTR centile < 50), 'ptv' (protein-truncating;

$MAF_{global} \le 0.1\%$, $MAF_{any\,given\,ancestry} \le 0.1\%$), 'ptv5pcnt' (protein-truncating; $MAF_{global} \le 5\%$, $MAF_{any\,given\,ancestry} \le 5\%$), 'ptvraredmg' ('ptv' and 'raredmg') and 'rec" (non-synonymous, $MAF_{global} \le 0.5\%$, $MAF_{any\,given\,ancestry} \le 0.5\%$). All qualifying variants were assessed in a dominant model except 'rec', which was assessed in a recessive model. Statistically significant genes were defined by a study-wide significance threshold of $P < 1 \times 10^{-8}$ (ref. [41]).

Genetic association analyses were repeated with the case and control groups randomly assigned to each individual to obtain the permuted $P$ values for genomic inflation assessment. MCPS and UKB did not display high levels of genomic inflation with overall CH: the inflation factors were 0.98 and 1.12 for GWAS, 1.02 and 1.20 for ExWAS and 1.01 and 1.02 for gene-level collapsing analysis.

### Genetic association meta-analyses

Meta-analysis across MCPS and UKB was performed using cross-ancestry (stratified) meta-analysis based on the summary statistics returned from GWAS, ExWAS and gene-level collapsing analyses[2]. GWAS meta-analysis was performed using the inverse variance-weighted average (IVW)-based method by summarizing the effect size of each variant from both cohorts. GWAS meta-analysis was additionally performed using the $P$ value-based method implemented in METAL[69], which is more robust than IVW when combining summary statistics with large effect size standard errors. Statistically significant variants were defined with $P < 5 \times 10^{-8}$ from both IVW-based and $P$ value-based methods. ExWAS meta-analysis was performed using the $P$ value-based method, and statistically significant variants were defined with $P < 1 \times 10^{-8}$. Cochran's $Q$-test was additionally performed to assess for variant-level heterogeneity in meta-analysis[70]. Both IVW-based and $P$ value-based methods and Cochran's $Q$-test were implemented by the METAL software[69]. Gene-level collapsing meta-analysis was performed using the Cochran–Mantel–Haenszel test using the *mantelhaen.test* function as implemented by the *stats* package in R (v.4.2.2).

### Statistics and reproducibility

Except when specific software packages are named, all statistical analyses and plotting were performed using R (v.4.2.2). No statistical methods were used to predetermine sample size.

### Reporting summary

Further information on research design is available in the Nature Portfolio Reporting Summary linked to this article.

### Data availability

Full summary statistics for GWAS are available from the NHGRI-EBI GWAS Catalog[71]. The GWAS Catalog accession numbers for MCPS are GCST90435341 (all CH), GCST90435342 (*DNMT3A*-CH), GCST90435343 (*TET2*-CH), GCST90435344 (*ASXL1*-CH), GCST90435345 (*PPM1D*-CH), GCST90435346 (*TP53*-CH), GCST90435347 (*SF3B1*-CH), GCST90435348 (*SRSF2*-CH), GCST90435349 (*GNB1*-CH), GCST90435350 (*IDH2*-CH), GCST90435351 (*JAK2*-CH) and GCST90435352 (*SF3B1*+SRSF2-CH). The GWAS Catalog accession numbers for UKB are GCST90435353 (all CH), GCST90435354 (*DNMT3A*-CH), GCST90435355 (*TET2*-CH), GCST90435356 (*ASXL1*-CH), GCST90435357 (*PPM1D*-CH), GCST90435358 (*TP53*-CH), GCST90435359 (*SF3B1*-CH), GCST90435360 (*SRSF2*-CH), GCST90435361 (*GNB1*-CH), GCST90435362 (*IDH2*-CH), GCST90435363 (*JAK2*-CH) and GCST90435364 (*SF3B1*+*SRSF2*-CH). The GWAS Catalog accession numbers for the cross-ancestry meta-analysis are GCST90435365 (all CH), GCST90435366 (*DNMT3A*-CH), GCST90435367 (*TET2*-CH), GCST90435368 (*ASXL1*-CH), GCST90435369 (*PPM1D*-CH), GCST90435370 (*TP53*-CH), GCST90435371 (*SF3B1*-CH), GCST90435372 (*SRSF2*-CH), GCST90435373 (*GNB1*-CH), GCST90435374 (*IDH2*-CH), GCST90435375 (*JAK2*-CH) and GCST90435376 (*SF3B1*+*SRSF2*-CH). Full summary statistics for ExWAS and gene-

collapsing analysis are available on Zenodo (ExWAS, https://doi.org/10.5281/zenodo.12690507 (ref. [72]); gene-collapsing analysis, https://doi.org/10.5281/zenodo.12691572 (ref. [73])). Individual-level data are under controlled access to protect the sensitive information of the study participants. Individual-level UKB data may be requested by application to the UKB. All WES data described in our study are available to registered researchers through the UKB data access protocol. Exomes can be found in the UKB showcase portal (https://biobank.ndph.ox.ac.uk/showcase/label.cgi?id=170). Additional information about data access registration is available at https://www.ukbiobank.ac.uk/enable-your-research/register. Individual-level MCPS data may be requested from the Data and Sample Access Policy available on the study's Oxford-hosted webpage (http://www.ctsu.ox.ac.uk/research/mcps)[23]. In an effort to increase research equity, MCPS releases data to researchers in Mexico (and their collaborators) 2 years before releasing it worldwide. Accordingly, baseline data, mortality data and resurvey data are currently available and genetic data will be available to all researchers in 2025. All WES and WGS data are available through the DNAnexus platform powered by Amazon Web Services. The TOPMed Data Coordinating Center, supported by NLHBI contract HHSN26800001, established the dbGaP accession (phs001974) for sharing the TOPMed genomic summary results that we accessed in this study (application ID: 135432-3).

### Code availability

All analyses were performed using publicly available software and web-based applications as indicated in the Methods section.

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

## Acknowledgements

We thank the participants and investigators in the MCPS and UKB study who made this work possible; the Mexican Health Ministry, the National Council of Science and Technology for Mexico, the Wellcome Trust (058299/Z/99), Cancer Research UK, British Heart Foundation and the UK Medical Research Council (MC_UU_00017/2) for funding the MCPS; the UKB Exome Sequencing Consortium (UKB-ESC) members AbbVie, Alnylam Pharmaceuticals, AstraZeneca, Biogen, Bristol Myers Squibb, Pfizer, Regeneron and Takeda for funding the generation of the data and the Regeneron Genetics Center for completing the sequencing and initial quality control of the exome sequencing data; the AstraZeneca Centre for Genomics Research Analytics and Informatics team for processing and analysis of sequencing data; and B. Prins for feedback on the manuscript. We further acknowledge the NHLBI Trans-Omics for Precision Medicine (TOPMed) program for supporting the collection of whole-genome sequence data and other omics data used in the genetic studies for which multi-study genomic summary results have been generated. Some contributing studies were co-funded by the NHGRI Centers for Common Disease Genetics (CCDG) program. We also thank the investigators, staff and participants of each contributing study. G.S.V. was supported by a Cancer Research UK Senior Cancer Fellowship (C22324/A23015).

## Author contributions

S. Wen, P.K.-M., R.T.-C., M.A.F., A.R.H., G.S.V. and J.M. designed the study. F.H., S.V.V.D., H.T., S. Wasilewski and Q.W. performed the bioinformatics processing. S. Wen and J.M. performed the analysis and statistical interpretation. S. Wen, M.A.F., A.R.H., G.S.V. and J.M. wrote the manuscript. S. Wen, P.K.-M., F.H., A.N., I.T., S.V.V.D., H.T., K.R.S., D.P.L., O.S.B., R.S.D., S. Wasilewski, J.A.-D., J.B., J.E., J.M.T., R.C., K.C., Q.W., S.P., R.T.-C., M.A.F., A.R.H., G.S.V. and J.M. reviewed the manuscript.

## Competing interests

S. Wen, F.H., A.N., I.T., S.V.V.D., H.T., K.R.S., K.C., D.P.L., O.S.B., R.S.D., S. Wasilewski, Q.W., S.P., M.A.F., A.R.H., J.M. are current employees and/or stockholders of AstraZeneca. R.C. is the chair of the data monitoring committee of the PROMINENT trial and the deputy chair of a not-for-profit clinical trial company (PROTAS) unrelated to this work. The other authors declare no competing interests.

## Additional information

**Extended data** is available for this paper at https://doi.org/10.1038/s41588-025-02085-6.

**Correspondence and requests for materials** should be addressed to George S. Vassiliou or Jonathan Mitchell.

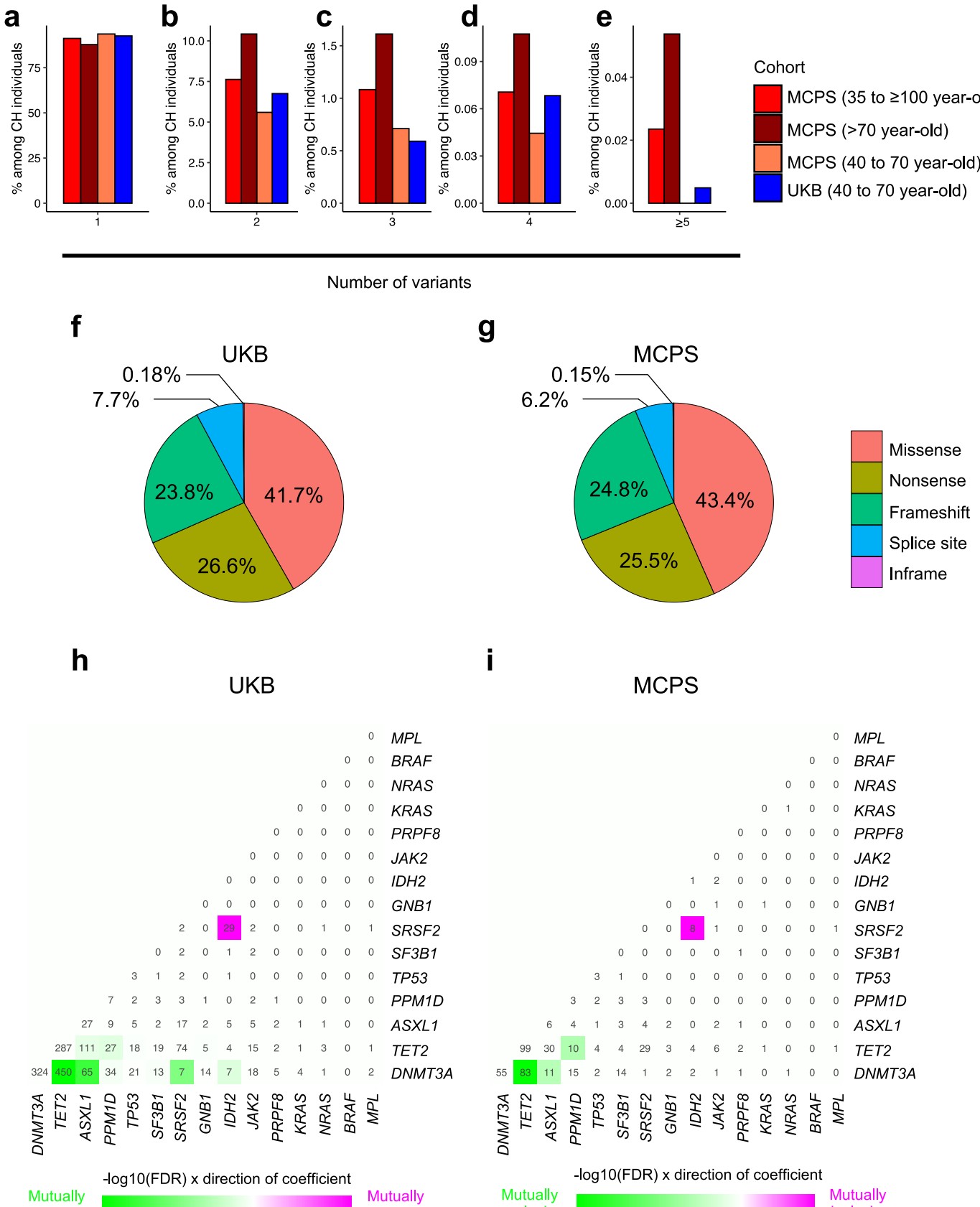

**Extended Data Fig. 1 | See next page for caption.**

**Extended Data Fig. 1 | Frequency and characteristics of clonal haematopoiesis (CH). a**–**e**, Percentage of CH individuals with 1 (**a**), 2 (**b**), 3 (**c**), 4 (**d**), or 5 or more (**e**) CH driver gene variants stratified by different age groups. **f**, **g**, CH driver gene variants stratified by consequence on protein-coding sequence in UK Biobank (UKB; **f**) and Mexico City Prospective Study (MCPS; **g**). **h**, **i**, Assessment of co-occurrence or mutual exclusivity of CH driver genes among participants with at least two mutated CH driver genes in UKB (**h**) and MCPS (**i**). ORs and two-sided *P* values were derived from logistic regression model. *P* values were adjusted for multiple testing using Benjamini-Hochberg procedure. CH, clonal haematopoiesis; FDR, false discovery rate; MCPS, Mexico City Prospective Study; UKB, UK Biobank.

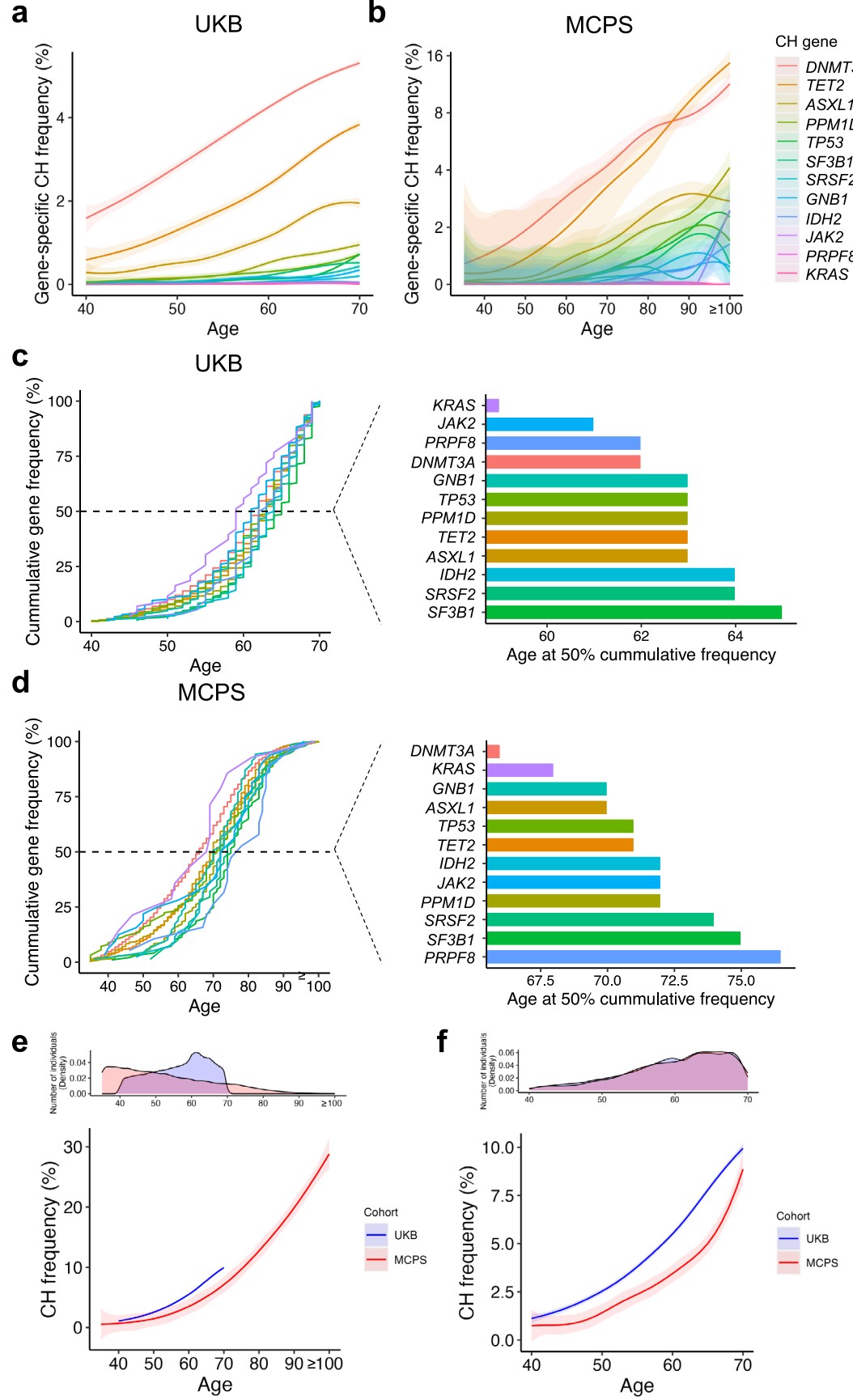

Extended Data Fig. 2 | See next page for caption.

**Extended Data Fig. 2 | Frequency of clonal haematopoiesis (CH) by age.**
Frequency of CH individuals by age stratified by gene-specific CH in UK Biobank
(UKB; **a**) and Mexico City Prospective Study (MCPS; **b**). The centre line represents
the fitted values from the general additive model with P-spline smooth class, and
the shaded region represents the lower and upper bound of the 95% confidence
interval of the fitted values. Y-axis is in log2 scale. Only CH driver genes identified
in at least 10 individuals are shown here (**a**, **b**). **c**, **d**, Cumulative frequency of CH
individuals by age stratified by gene-specific CH in UKB (**c**) and MCPS (**d**). Only
CH driver genes identified in at least 10 individuals are shown here. **e**, Same as

Fig. 1b, but with UKB and MCPS age distribution indicated. **f**, Frequency of CH
individuals by age after age- and sex-matching UKB and MCPS participants.
Overall CH frequency was 4.55% and 2.87% for UKB and MCPS participants,
respectively ($P < 2.2 \times 10^{-16}$). The centre line represents the fitted values from
the general additive model with P-spline smooth class, and the shaded region
represents the lower and upper bound of the 95% confidence interval of the fitted
values (**e**, **f**). CH, clonal haematopoiesis; MCPS, Mexico City Prospective Study;
UKB, UK Biobank.

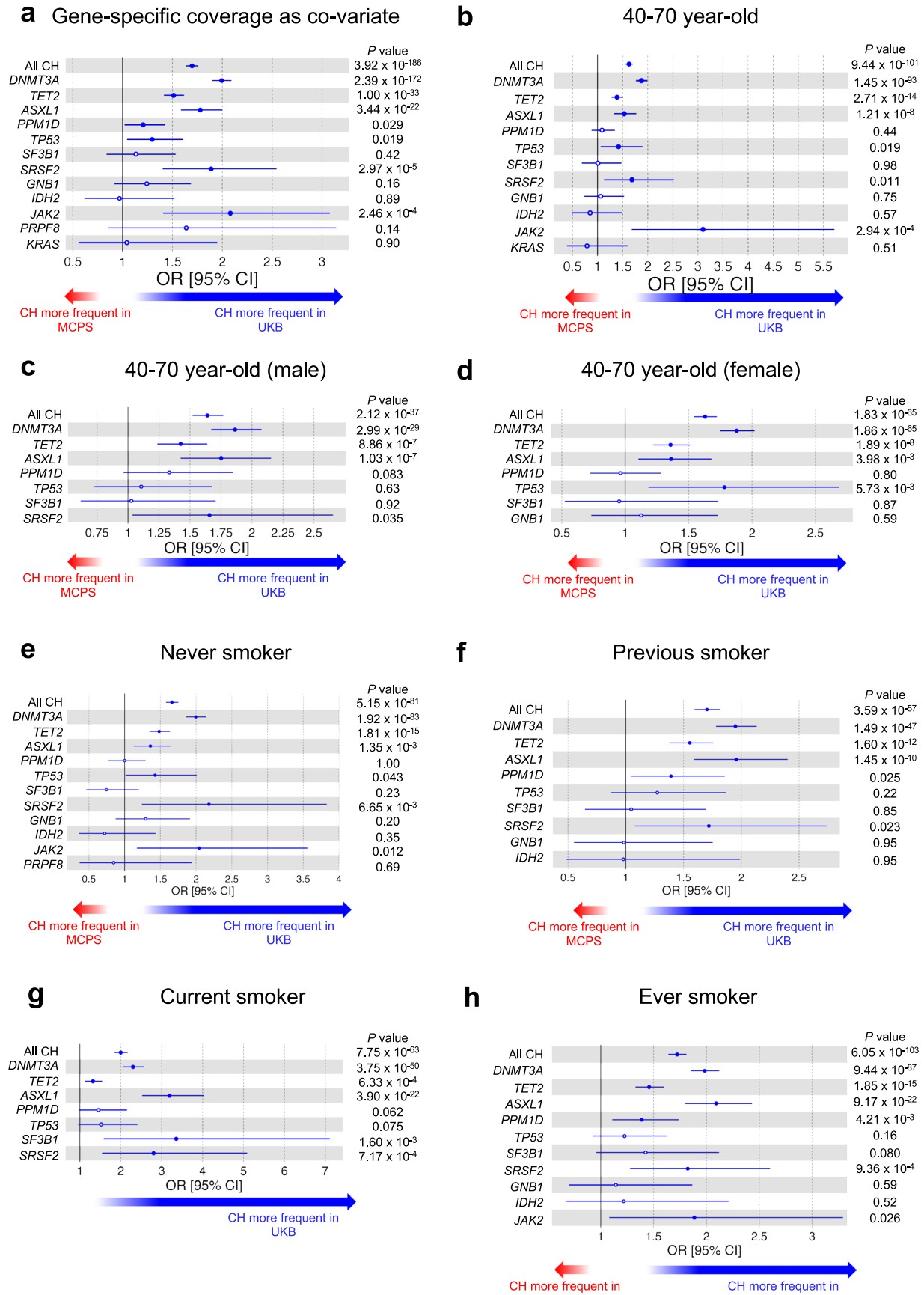

**Extended Data Fig. 3 | See next page for caption.**

**Extended Data Fig. 3 | Inter-population analysis of overall clonal haematopoiesis (CH) and gene-specific CH between UK Biobank (UKB) and Mexico City Prospective Study (MCPS). a**, Comparison of CH frequency between 416,118 UKB and 136,401 MCPS participants. Logistic regression model was adjusted for age, sex, and smoking status, and the percentage of bases with ≥20x coverage for the corresponding CH gene. For overall CH, the average coverage across the 15-gene panel was included as the co-variate. **b**, Comparison of CH frequency between 416,109 UKB and 95,294 MCPS participants aged 40–70 years of age. Logistic regression model was adjusted for age, sex, and smoking status. **c**, Comparison of CH frequency between 191,476 UKB males and 31,074 MCPS males aged 40–70 years of age. Logistic regression model was adjusted for age and smoking status. **d**, Comparison of CH frequency between 224,633 UKB females and 64,220 MCPS females aged 40–70 years of age. Logistic regression model was adjusted for age and smoking status. **e**, Comparison of CH frequency between 223,303 UKB and 66,517 MCPS never smokers. **f**, Comparison of CH frequency between 147,650 UKB and 32,579 MCPS previous smokers. **g**, Comparison of CH frequency between 43,077 UKB and 37,263 MCPS current smokers. **h**, Comparison of CH frequency between 190,727 UKB and 69,842 MCPS ever smokers. Logistic regression model was adjusted for age and sex (**e**–**h**). Only gene-specific CH genes identified in at least 10 individuals are shown here. Odds ratios and unadjusted two-sided *P* values were derived from logistic regression model with all CH or gene-specific CH as outcome and study cohort as the predictor. Measures of centre represent the odds ratios, and the error bars represent the lower and upper bound of the 95% confidence interval of the odds ratios. Full circles represent significant associations (*P* < 0.05) while hollow circles represent non-significant associations (*P* ≥ 0.05). 95% CI, 95% confidence interval; CH, clonal haematopoiesis; MCPS, Mexico City Prospective Study; OR, odds ratio; UKB, UK Biobank.

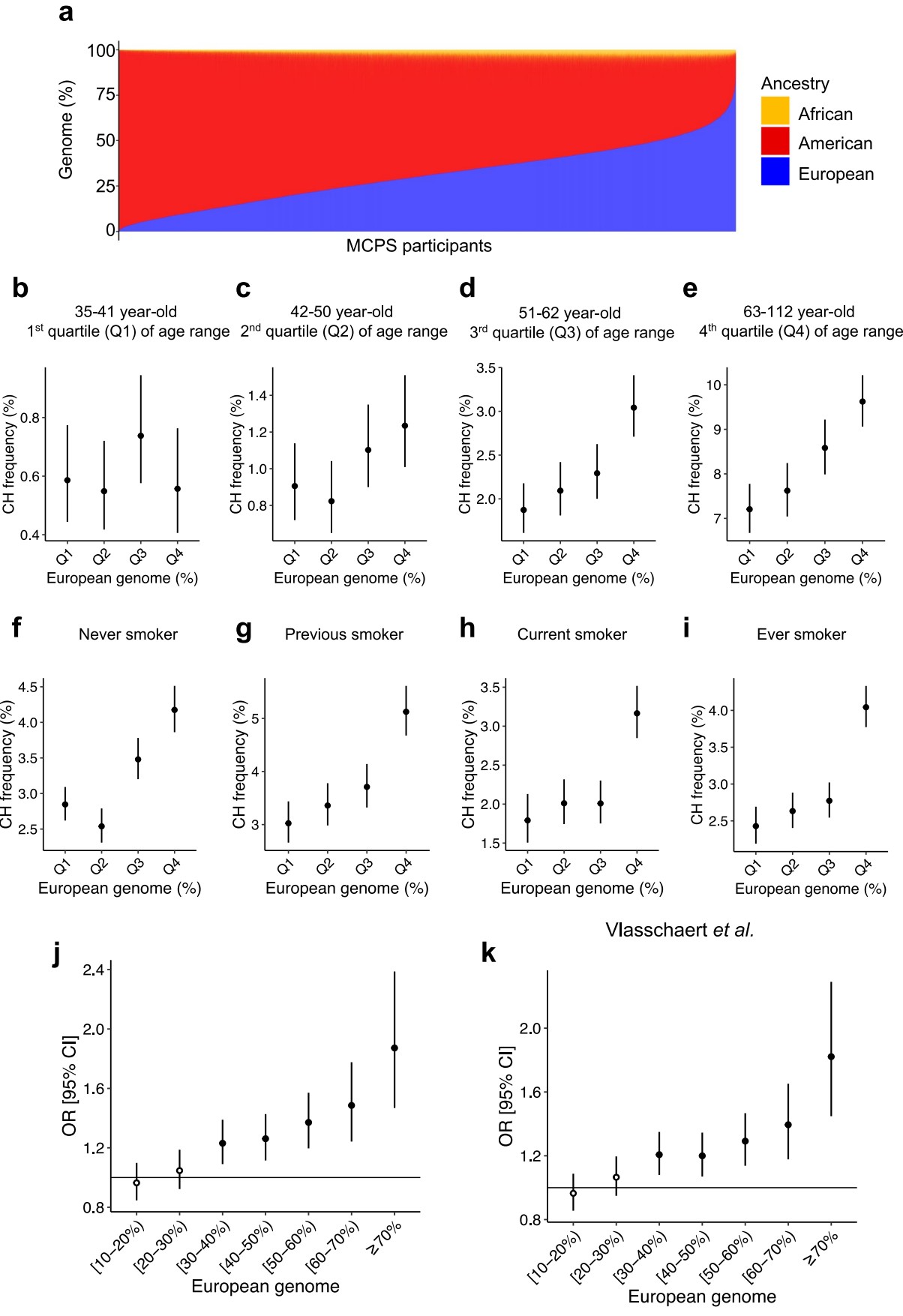

**Extended Data Fig. 4 | See next page for caption.**

**Extended Data Fig. 4 | Frequency of all clonal haematopoiesis (CH) across different strata of European genome proportions in Mexico City Prospective Study (MCPS). a**, On the y-axis is the proportion of European, Indigenous American, and African genome across the MCPS participants on the x-axis. **b**–**e**, Frequency of overall CH across groups of individuals with increasing proportion of European genome, stratified by age quartiles. The proportions of European genome were binned by quartiles in this analysis whereby quartile 1 (Q1), Q2, Q3, and Q4 represent [0–18.2), [18.2–30.8), [30.8–42.5), and [42.5–90.3] proportion of European genome, respectively. In total, 134,297 individuals with proportion of European genome available were included for analysis here. **f**–**i**, Frequency of overall CH across groups of individuals with increasing proportion of European genome, stratified by smoking status (never, previous, current, and ever). In total, 134,255 individuals with proportion of European genome and smoking status available were included for analysis here. Error bars represent the lower and upper bound of 95% confidence interval of the

CH frequencies (**b**–**i**). **j**, **k**, Overall CH risk associated with binned proportion of European genome relative to individuals with <10% European genome when CH individuals were defined using the 15-gene panel in our study (**j**) or defined using the 58-gene panel (Vlasschaert et al.; **k**). Odds ratios and unadjusted two-sided *P* values were derived from logistic regression model with all CH as outcome, and with binned proportion of European genome as predictor, adjusted for age, sex, and smoking status. In total, 134,255 individuals with proportion of European genome and smoking status available were included for analysis here. Measures of centre represent the odds ratios, and the error bars represent the lower and upper bound of the 95% confidence interval of the odds ratios. Full circles represent significant associations (*P* < 0.05) while hollow circles represent non-significant associations (*P* ≥ 0.05) (**j**, **k**). 95% CI, 95% confidence interval; CH, clonal haematopoiesis; MCPS, Mexico City Prospective Study; OR, odds ratio; Q, quartile.

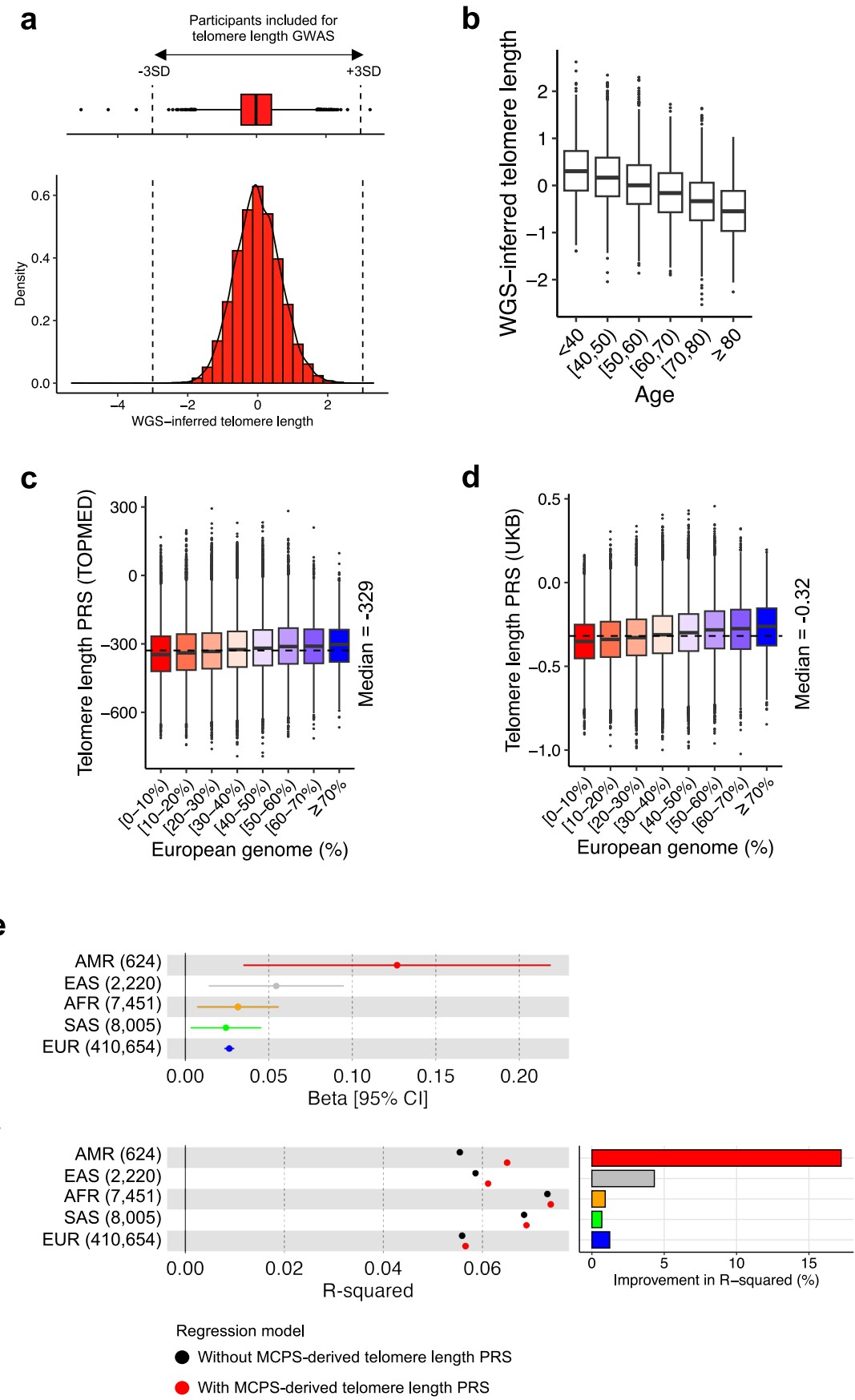

**Extended Data Fig. 5 | See next page for caption.**

**Extended Data Fig. 5 | Establishing, applying, and validating telomere length polygenic risk score (PRS) derived from Mexico City Prospective Study (MCPS) participants with whole-genome sequencing (WGS) data available. a**, In total, WGS-inferred leukocyte telomere length (LTL) measurements were available for 9,602 individuals, of which, 9,598 individuals had LTL measurements within ±3 standard deviations and subsequently included for LTL GWAS. **b**, Distribution of WGS-inferred LTL by increasing age group among 9,598 individuals. **c**, **d**, Distribution of LTL PRS across 126,803 individuals with varying degree of European genome for PRS built using published independent variants from Trans-Omics for Precision Medicine (TOPMed) multi-ancestry cohort (**c**) and from UKB European-majority cohort (**d**). The 9,598 individuals with WGS-inferred LTL which were included for LTL GWAS were excluded from analysis in (**c**, **d**). Boxplots represent the median, first and third quartiles, and whiskers represent 1.5 times the interquartile range (**b**–**d**). **e**, Validation of MCPS-derived LTL PRS among the different ancestry groups in UKB based on correlation between LTL PRS versus WGS-inferred LTL. A linear regression model was fitted using WGS-inferred LTL as the outcome and LTL PRS as the predictor, adjusted

for adjusted for age, sex, smoking status, and the first four genetic principal components. The effect size of the model for each ancestry group subsequently reported here. Measures of centre represent the beta coefficients, and the error bars represent the lower and upper bound of the 95% confidence interval of the beta coefficients. **f**, Validation of MCPS-derived LTL PRS by assessing improvement in R-squared value computed by comparing linear regression model with versus without LTL PRS as co-variate. Specifically, the former model is the same as (**e**). In the latter model, LTL PRS was excluded as a co-variate. The percentage increase in R-squared value was computed for the former model relative to the latter model and subsequently reported here (**f**). In total, 624 AMR, 2,220 EAS, 7,451 AFR, 8,005 SAS, and 410,654 EUR participants with LTL PRS, WGS-inferred LTL and smoking status available were included for analysis here (**e**, **f**). 95% CI, 95% confidence interval; AFR, African; AMR, Admixed American; EAS, East Asian; EUR, European; LTL, leukocyte telomere length; MCPS, Mexico City Prospective Study; PRS, polygenic risk score; SAS, South Asian; TOPMed, Trans-Omics for Precision Medicine; UKB, UK Biobank; WGS, whole-genome sequencing.

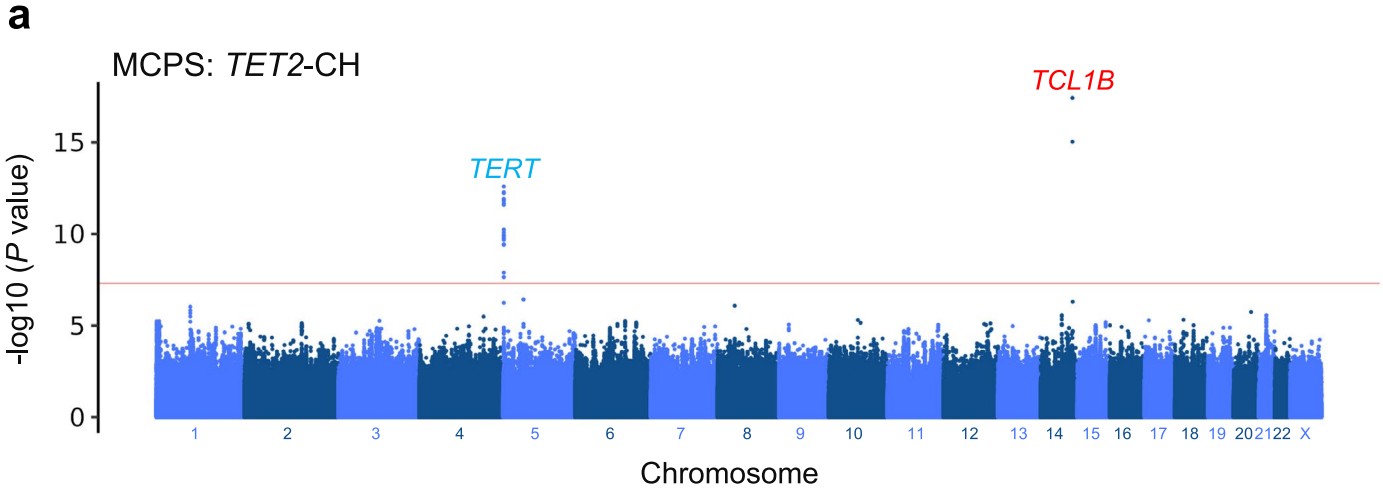

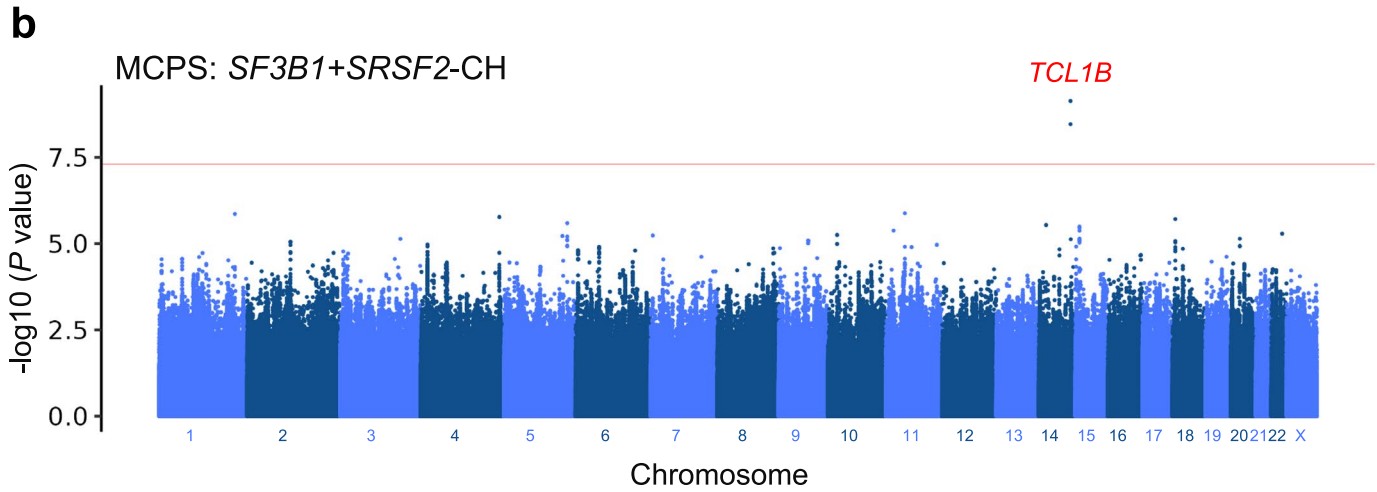

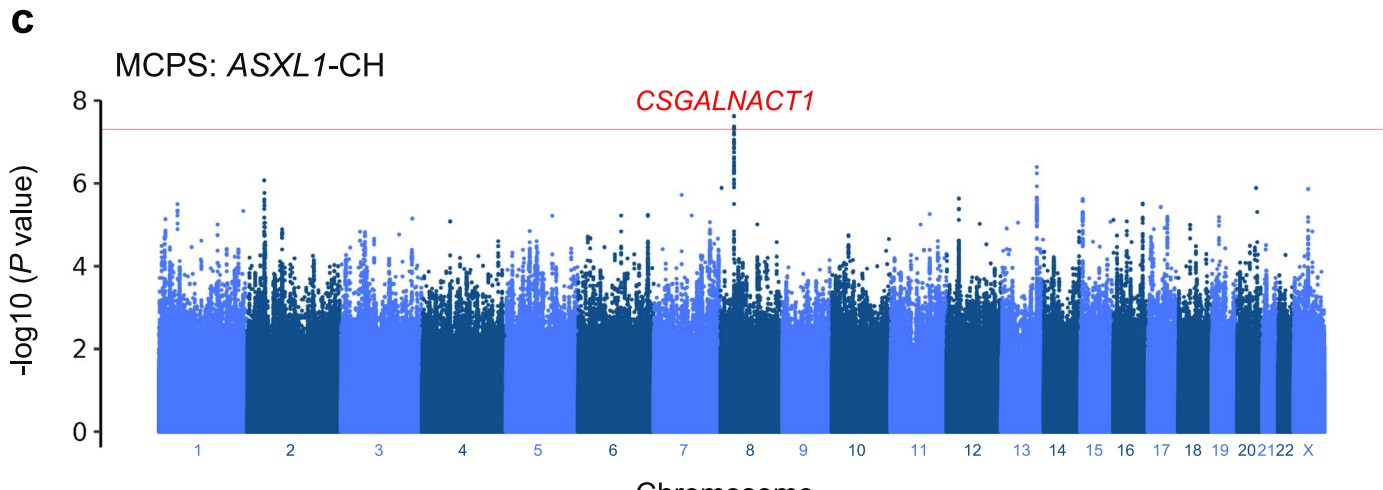

**Extended Data Fig. 6 | Genome-wide association study (GWAS) gene-specific clonal haematopoiesis (CH) in Mexico City Prospective Study (MCPS).** **a–c**, Manhattan plots representing the common germline variants with minor allele frequency (MAF) ≥ 1% included for GWAS in MCPS for *TET2-* (**a**), splicing factor- (**b**), and *ASXL1-* (**c**) CH. Unadjusted two-sided *P* values on y-axis were derived from Firth logistic regression implemented by REGENIE software.

Three novel signals (red) were identified as genome-wide significant (*P* value < 5 × 10⁻⁸, red horizontal line) with the nearest gene of the leading SNP annotated for the respective locus. Previously reported associations from European populations indicated in blue. CH, clonal haematopoiesis; GWAS, genome-wide association study; MAF, minor allele frequency; MCPS, Mexico City Prospective Study.

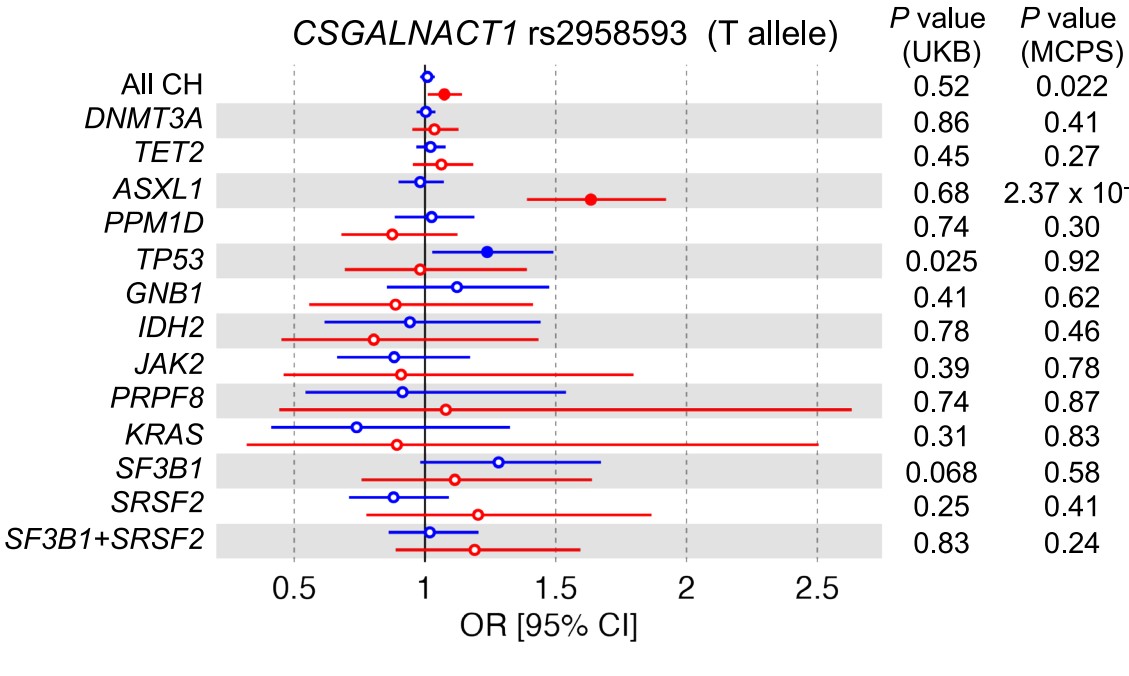

Cohort ●— UKB ●— MCPS

**Extended Data Fig. 7 | Risk estimates of a novel common clonal haematopoiesis (CH) risk variant identified from *ASXL1*-CH genome-wide association study (GWAS) in Mexico City Prospective Study (MCPS).** rs2958593 identified as genome-wide significant. Overall CH and gene-specific CH risk estimates conferred by the minor allele (T) shown here for 406,826 UK Biobank (UKB) and 136,401 MCPS participants. Odds ratios and unadjusted two-sided *P*-values were derived from Firth logistic regression implemented by REGENIE software. Measures of centre represent the odds ratio, and the error bars represent the lower and upper bound of the 95% confidence interval of the odds ratios. Full circles represent significant associations (*P* < 0.05) while hollow circles represent non-significant associations (*P* ≥ 0.05). CH, clonal haematopoiesis; GWAS, genome-wide association study; MCPS, Mexico City Prospective Study; OR, odds ratio; UKB, UK Biobank.

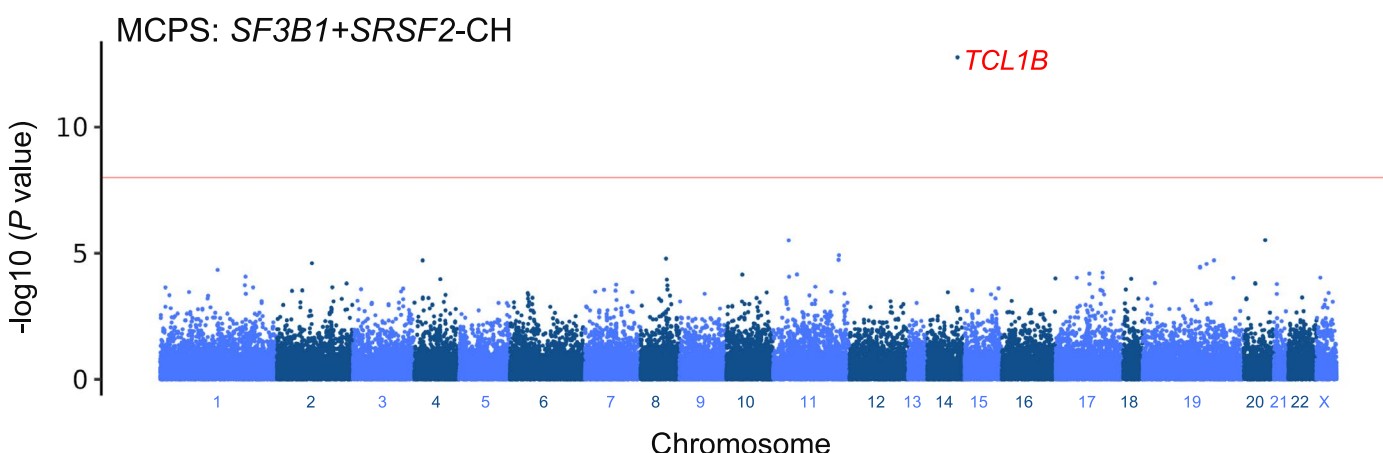

**Extended Data Fig. 8 | Exome-wide association study (ExWAS) of *SF3B1+SRSF2*-CH in Mexico City Prospective Studt (MCPS).** Manhattan plot representing the common (minor allele frequency (MAF) ≥ 1%) and rare (MAF < 1%) germline variants included for ExWAS. Unadjusted two-sided *P*-values on y-axis were derived from Firth logistic regression implemented by REGENIE software. One novel association from ExWAS indicated in red. Nearest gene of the leading SNP annotated. CH, clonal haematopoiesis; ExWAS, exome-wide association study; MAF, minor allele frequency; MCPS, Mexico City Prospective Study.

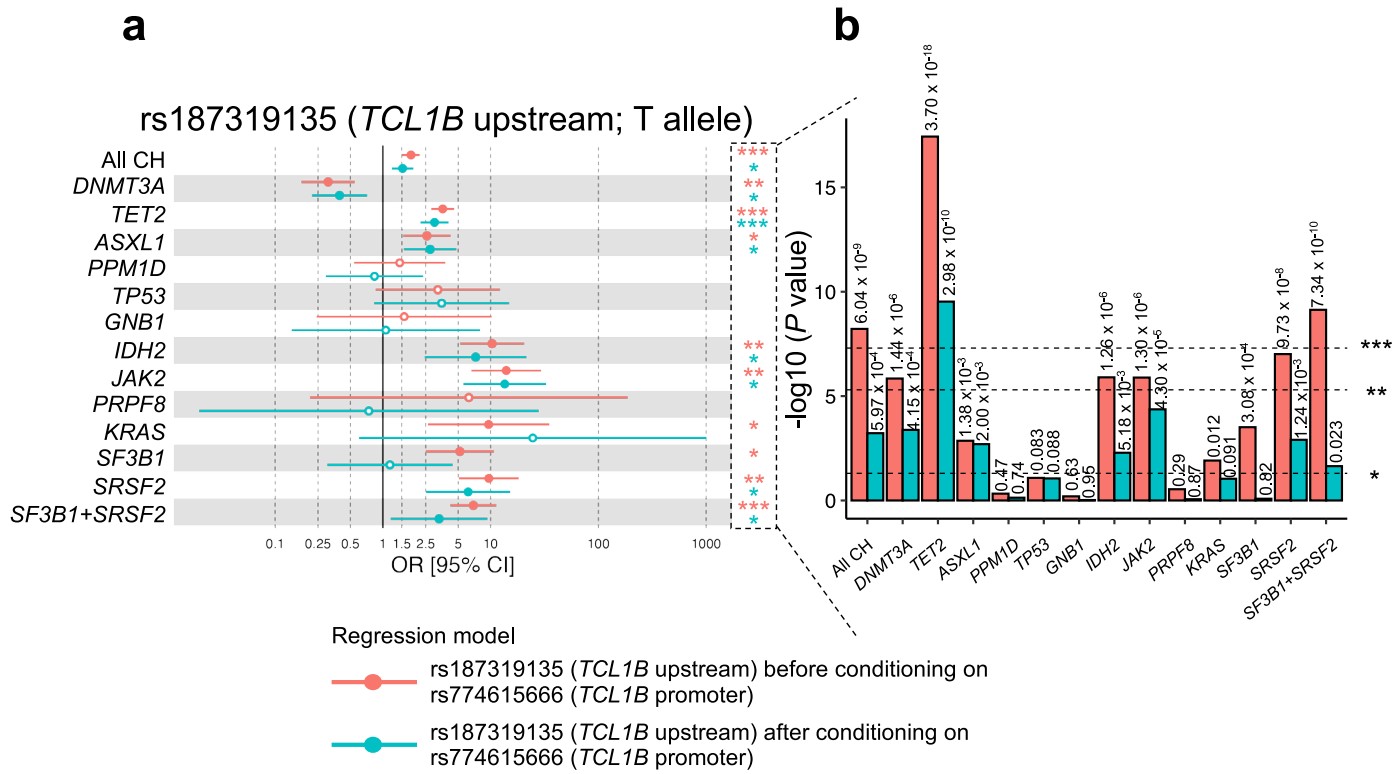

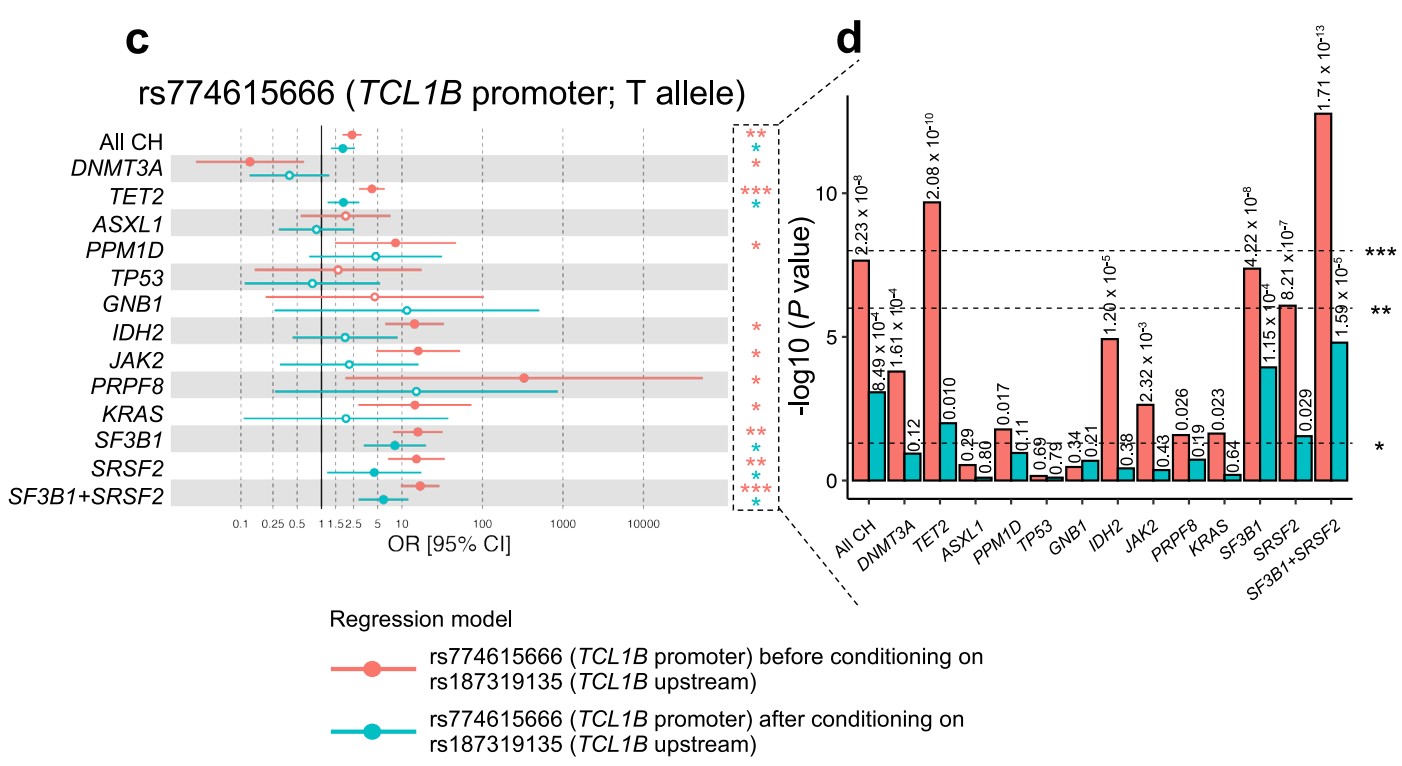

**Extended Data Fig. 9 | See next page for caption.**

**Extended Data Fig. 9 | Conditional analysis of rs187319135 (*TCL1B* upstream) and rs774615666 (*TCL1B* promoter) variants in Mexico City Prospective Study (MCPS). a**, **b**, Risk conferred by rs187319135 (T allele) to overall clonal haematopoiesis (CH) and gene-specific CH before versus after conditioning on rs774615666. In the latter mode, genotype of rs774615666 was determined from whole-exome sequencing (WES) and included as co-variate in the Firth logistic regression model implemented by REGENIE software, adjusted for age, sex, and first ten genetic principal components. In total, 136,149 participants with WES-called rs774615666 genotype and complete co-variate data available were included for analysis here. **c**, **d**, Risk conferred by rs774615666 (T allele) to overall CH and gene-specific CH before versus after conditioning on rs187319135. In the latter mode, genotype of rs187319135 was hard-called from the imputed genetic data, and included as co-variate in the Firth logistic regression model implemented by REGENIE, adjusted for age, sex, and first ten genetic principal components. Thresholds for hard-calling genotypes were $0 \leq x \leq 0.1$, $0.9 \leq x \leq 1.1$, and $1.9 \leq x \leq 2.0$ for homozygous minor allele, heterozygous minor/

major allele, and homozygous major allele, respectively, where $x$ is the allelic dosage (expected number of copies of major allele). Allelic dosages outside the range of thresholds were coded as missing. In total, 134,651 participants with SNP array-based hard-called rs187319135 genotype, WES-called rs774615666 genotype, and complete co-variate data available were included for analysis here. (**a**, **c**) Odds ratio and unadjusted two-sided $P$ values were derived from Firth logistic regression implemented by REGENIE software. Measures of centre represent the odds ratios, and the error bars represent the lower and upper bound of the 95% confidence interval of the odds ratios. Full circles represent significant associations ($P < 0.05$) while hollow circles represent non-significant associations ($P \geq 0.05$). $P$ value for rs187319135 *** $< 5 \times 10^{-8}$ (genome-wide significant), ** $< 5 \times 10^{-6}$ (suggestive), * $< 0.05$ (nominal). $P$ value for rs774615666 *** $< 1 \times 10^{-8}$ (exome-wide significant), ** $< 1 \times 10^{-6}$ (suggestive), * $< 0.05$ (nominal). 95% CI, 95% confidence interval; CH, clonal haematopoiesis; MAF, minor allele frequency; MCPS, Mexico City Prospective Study; OR, odds ratio; UKB, UK Biobank.

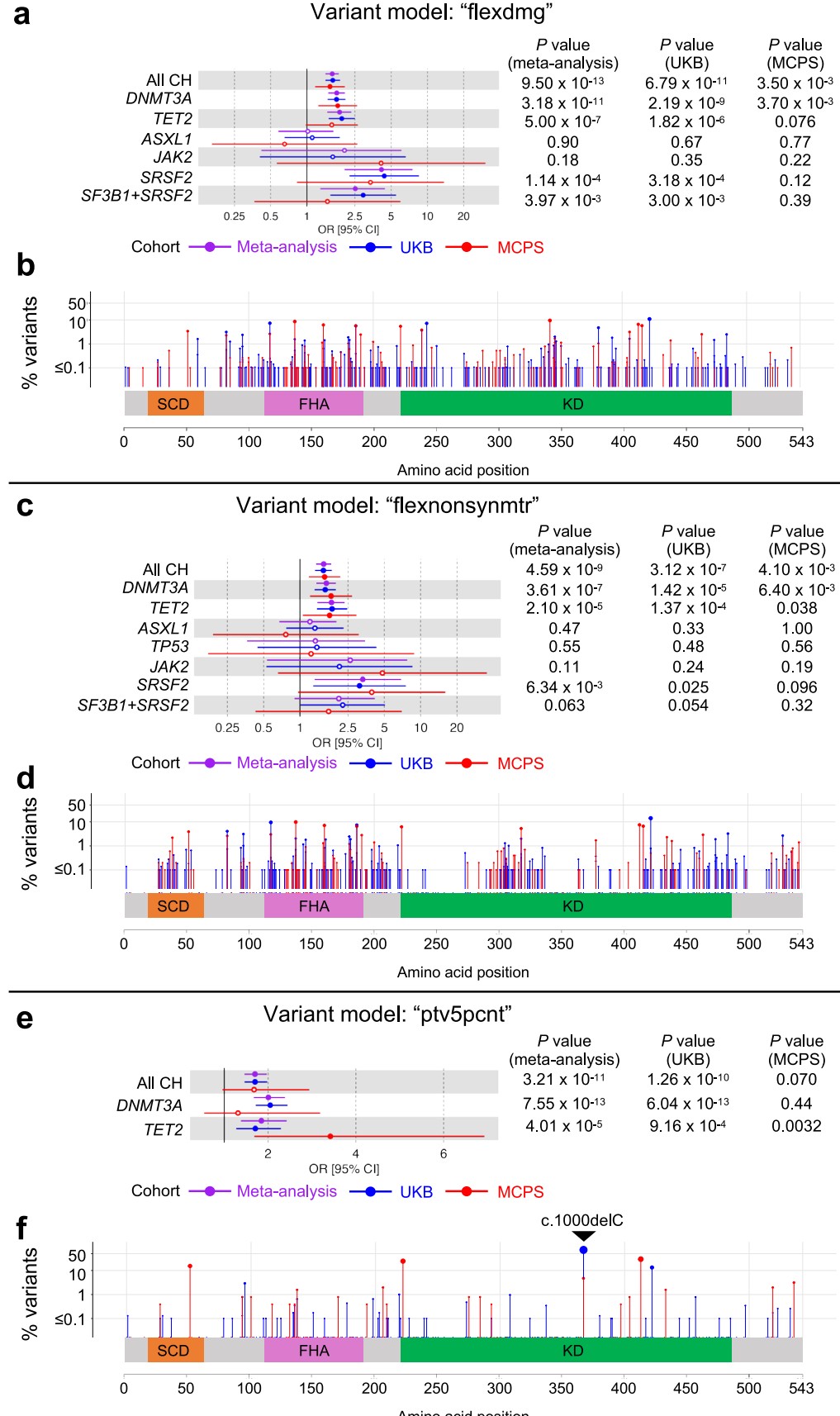

Extended Data Fig. 10 | See next page for caption.

**Extended Data Fig. 10 | Rare *CHEK2* variant burden association meta-analysis across Mexico City Prospective Study (MCPS) and UK Biobank (UKB). a**, **b**, *CHEK2* "flexdmg" qualifying variant model identified as genome-wide significant (*P* value < 1 × 10⁻⁸) in overall clonal haematopoisis (CH) and *DNMT3A*-CH. **c**, **d**, *CHEK2* "flexnonsynmtr" qualifying variant model identified as genome-wide significant in overall CH. **e**, **f**, *CHEK2* "ptv5pcnt" qualifying variant model identified as genome-wide significant in overall CH and *DNMT3A*-CH. For each qualifying variant model, the risk estimates conferred (**a**, **c**, **e**) and individual variant as a percentage all *CHEK2* variants identified in MCPS and UKB (**b**, **d**, **f**) shown. Odds ratio and unadjusted two-sided *P* values were derived from Cochran-Mantel-Haenszel (CMH) test. Measures of centre represent the odds ratios, and the error bars represent the lower and upper bound of the 95% confidence interval of the odds ratios. Full circles represent significant associations (*P* < 0.05) while hollow circles represent non-significant associations (*P* ≥ 0.05) (**a**, **c**, **e**). In total, 136,398 MCPS and 416,115 UKB participants were included for analysis here. 95% CI, 95% confidence interval; CH, clonal haematopoiesis; FHA, forkhead-associated domain; KD, kinase domain; MCPS, Mexico City Prospective Study; OR, odds ratio; SCD, SQ/TQ cluster domain (SCD); UKB, UK Biobank.

Prof. George Vassiliou

# Reporting Summary

## Statistics

For all statistical analyses, confirm that the following items are present in the figure legend, table legend, main text, or Methods section.

| n/a | Confirmed | |
|---|---|---|
| ☐ | ☒ | The exact sample size (*n*) for each experimental group/condition, given as a discrete number and unit of measurement |
| ☐ | ☒ | A statement on whether measurements were taken from distinct samples or whether the same sample was measured repeatedly |
| ☐ | ☒ | The statistical test(s) used AND whether they are one- or two-sided<br>*Only common tests should be described solely by name; describe more complex techniques in the Methods section.* |
| ☐ | ☒ | A description of all covariates tested |
| ☐ | ☒ | A description of any assumptions or corrections, such as tests of normality and adjustment for multiple comparisons |
| ☐ | ☒ | A full description of the statistical parameters including central tendency (e.g. means) or other basic estimates (e.g. regression coefficient) AND variation (e.g. standard deviation) or associated estimates of uncertainty (e.g. confidence intervals) |
| ☐ | ☒ | For null hypothesis testing, the test statistic (e.g. *F*, *t*, *r*) with confidence intervals, effect sizes, degrees of freedom and *P* value noted<br>*Give P values as exact values whenever suitable.* |
| ☒ | ☐ | For Bayesian analysis, information on the choice of priors and Markov chain Monte Carlo settings |
| ☒ | ☐ | For hierarchical and complex designs, identification of the appropriate level for tests and full reporting of outcomes |
| ☐ | ☒ | Estimates of effect sizes (e.g. Cohen's *d*, Pearson's *r*), indicating how they were calculated |

*Our web collection on statistics for biologists contains articles on many of the points above.*

## Software and code

Policy information about availability of computer code

| Data collection | No software was used for data collection. |
|---|---|
| Data analysis | All analyses were performed using publicly available software and web-based applications as indicated in the Methods section.<br>1. Conversion of sequencing data in BCL format to FASTQ format: bcl2fastq v2.19.0<br>2. Sequencing read alignment to the GRCh38 genome reference and germline variant detection: Illumina DRAGEN Bio-IT Platform Germline Pipeline v3.0.7<br>3. Somatic variant calling: GATK MuTect2 v.2.2.0<br>4. DNA sample contamination: VerifyBAMID<br>5. Kinship: KING v2.2.3<br>6. Ancestry probability: peddy v0.4.2<br>7. Genome- and exome-wide association studies (GWAS and ExWAS, respectively): REGENIE v3.5<br>8. Whole-genome inferred telomere length: TelSeq v.0.0.2<br>9. Polygenic risk scoring, linkage disequilibrium analysis: PLINK2.0<br>10. Local ancestry inference: RFMix2.0<br>11. Logistic regression: glm function as implemented in stats package in R v4.2.20<br>12. Cross-ancestry GWAS and ExWAS meta-analysis: METAL (released 2011-03-25)<br>13. Gene-level collapsing analysis: Cochran-Mantel-Haenszel (CMH) test using the mantelhaen.test function as implemented by the stats package in R v4.2.2 |

For manuscripts utilizing custom algorithms or software that are central to the research but not yet described in published literature, software must be made available to editors and reviewers. We strongly encourage code deposition in a community repository (e.g. GitHub). See the Nature Portfolio guidelines for submitting code & software for further information.

## Data

Policy information about availability of data

All manuscripts must include a data availability statement. This statement should provide the following information, where applicable:

- Accession codes, unique identifiers, or web links for publicly available datasets
- A description of any restrictions on data availability
- For clinical datasets or third party data, please ensure that the statement adheres to our policy

Full summary statistics for GWAS are available on NHGRI-EBI GWAS Catalog while the full summary statistics for ExWAS and gene-collapsing analysis are available on Zenodo. The GWAS Catalog accession numbers and Zendo links are indicated in Supplementary Table 23. Individual-level UK Biobank data may be requested via applicable to the UK Biobank. Individual-level MCPS data may be requested via Data and Sample Access Policy available on the study's Oxford-hosted webpage (http://www.ctsu.ox.ac.uk/research/mcps).

## Research involving human participants, their data, or biological material

Policy information about studies with human participants or human data. See also policy information about sex, gender (identity/presentation), and sexual orientation and race, ethnicity and racism.

| | |
|---|---|
| Reporting on sex and gender | Sex concordance was determined by comparing clinically-reported sex against chromosome X:Y consensus coding sequence coverage ratio. Only sex, but not gender, was reported by this study. |
| Reporting on race, ethnicity, or other socially relevant groupings | Ancestry of study participants were determined using the peddy software. Europeans individuals from UK Biobank were defined with peddy-inferred European probability of at least 95%. Similarly, Admixed American individuals from Mexico City Prospective Study (MCPS) were defined with peddy-inferred Admixed American probability of at least 95%. Neither race nor ethnicity was reported by this study. |
| Population characteristics | MCPS is a prospective cohort of more than 150,000 adults with genetic (whole-genome sequencing, whole-exome sequencing, SNP array) and phenotypic, and metabolomic data available. Details have been described in Ziyatdinov et al. (Nature, 2023) and Tapia-Conyer et al. (International Journal of Epidemiology, 2006). UKB is a prospective cohort of approximately 500,000 adults with genetic (whole-genome sequencing, whole-exome sequencing, SNP array) and phenotypic, proteomic, and matobolomic data available. Details for UKB have been described in Szustakowski et al. (Nature Genetics, 2021) and Bycroft et al. (Nature, 2018). <br><br> The median age of MCPS and UKB participants are 58 and 51 years old, respectively. In MCPS, female and male constitute 67% and 33% of participants, respectively. In UKB, female and male constitute 54% and 46% of participants, respectively. |
| Recruitment | MCPS participants were aged at least 35 years, and recruited between 1998 and 2004 from the contiguous urban districts of Coyoacán and Iztapalapa in Mexico City. UKB participants were aged between 40 to 70 years, and recruited since 2007. |
| Ethics oversight | The MCPS study was approved by the Mexican Ministry of Health, the Mexican National Council for Science and Technology, and the University of Oxford, and the UKB study has approval from the North-West Multi-centre Research Ethics Committee (11/NW/0382). |

Note that full information on the approval of the study protocol must also be provided in the manuscript.

# Field-specific reporting

Please select the one below that is the best fit for your research. If you are not sure, read the appropriate sections before making your selection.

☒ Life sciences  ☐ Behavioural & social sciences  ☐ Ecological, evolutionary & environmental sciences

For a reference copy of the document with all sections, see nature.com/documents/nr-reporting-summary-flat.pdf

# Life sciences study design

All studies must disclose on these points even when the disclosure is negative.

| | |
|---|---|
| Sample size | Initial 141,046 individuals from MCPS and 469,809 individuals from UKB were identified on the basis of whole-exome sequencing data available. |
| Data exclusions | In both MCPS and UKB, samples were selected on the basis of (1) contamination <4% computed by VerifyBAMID software, (2) gender concordant between clinically reported and chromosome X:Y consensus coding sequence (CCDS) coverage ratios, (3) ≥94.15% of CCDS r22 bases covered with ≥10x coverage, (4) within 4SDs of mean genetic principal components 1-4 as computed by the peddy software, and (5) SNP array QC (genotype missingness ≤10%). Samples from MCPS were additionally selected based on within 2 standard deviations (SDs) of the mean read-depth distribution, no pairs with kinship >0.45, and probability ≥0.95 of Admixed American ancestry. Samples from UKB were additionally selected based on no pairs with kinship >0.1769 and probability ≥0.95 of European ancestry. Kinship and ancestry were inferred using the KING and peddy softwares, respectively. peddy Admixed American and European ancestry probabilities were computed with the 1000 Genomes Admixed American and European reference panel, respectively. For UKB, individuals with prior diagnosis of |

haematological malignancies were excluded. Post-QC, 136,401 individuals from MCPS and 416,118 individuals from UKB were included in our study.

Replication | Non-applicable. This is a non-experimental, descriptive population-based study.

Randomization | Non-applicable. This is a non-experimental, descriptive population-based study.

Blinding | Non-applicable. This is a non-experimental, descriptive population-based study.

# Reporting for specific materials, systems and methods

We require information from authors about some types of materials, experimental systems and methods used in many studies. Here, indicate whether each material, system or method listed is relevant to your study. If you are not sure if a list item applies to your research, read the appropriate section before selecting a response.

## Materials & experimental systems

| n/a | Involved in the study |
|---|---|
| ☒ | Antibodies |
| ☒ | Eukaryotic cell lines |
| ☒ | Palaeontology and archaeology |
| ☒ | Animals and other organisms |
| ☒ | Clinical data |
| ☒ | Dual use research of concern |
| ☒ | Plants |

## Methods

| n/a | Involved in the study |
|---|---|
| ☒ | ChIP-seq |
| ☒ | Flow cytometry |
| ☒ | MRI-based neuroimaging |

## Plants

Seed stocks | *Report on the source of all seed stocks or other plant material used. If applicable, state the seed stock centre and catalogue number. If plant specimens were collected from the field, describe the collection location, date and sampling procedures.*

Novel plant genotypes | *Describe the methods by which all novel plant genotypes were produced. This includes those generated by transgenic approaches, gene editing, chemical/radiation-based mutagenesis and hybridization. For transgenic lines, describe the transformation method, the number of independent lines analyzed and the generation upon which experiments were performed. For gene-edited lines, describe the editor used, the endogenous sequence targeted for editing, the targeting guide RNA sequence (if applicable) and how the editor was applied.*

Authentication | *Describe any authentication procedures for each seed stock used or novel genotype generated. Describe any experiments used to assess the effect of a mutation and, where applicable, how potential secondary effects (e.g. second site T-DNA insertions, mosiacism, off-target gene editing) were examined.*

