## [Peer Review File · Nature Genetics]

Comparative analysis of 136,401 Admixed Americans and 416,118 Europeans identifies ancestry-specific effects on clonal hematopoiesis

Corresponding Author: Dr Jonathan Mitchell

Version 0:

Decision Letter:

2nd Apr 2024

Dear Dr Mitchell,

First, please accept my apologies for the delay in returning this decision to you. Thank you for your patience.

Your Article, "Comparative analysis of 136,401 Admixed Americans and 419,228 Europeans reveals ancestry-specific genetic determinants of clonal haematopoiesis" has now been seen by 3 referees. You will see from their comments copied below that while they find your work of considerable potential interest, they have raised quite substantial concerns that must be addressed. In light of these comments, we cannot accept the manuscript for publication, but would be very interested in considering a revised version that addresses these serious concerns.

We hope you will find the referees' comments useful as you decide how to proceed. If you wish to submit a substantially revised manuscript, please bear in mind that we will be reluctant to approach the referees again in the absence of major revisions.

You'll see that the reviewers have identified some important technical issues which together significantly undercut confidence in your findings. Of note, two reviewers have flagged the 15-gene panel used to phenotype and we'd suggest that you address these all these technical concerns as a priority, preferably with new analyses. That said, we'd expect all comments to be addressed in full, ideally with experiments or textually where appropriate.

If you choose to revise your manuscript taking into account all reviewer and editor comments, please highlight all changes in the manuscript text file. At this stage we will need you to upload a copy of the manuscript in MS Word .docx or similar editable format.

*2) If you have not done so already please begin to revise your manuscript so that it conforms to our Article format instructions, available here. Refer also to any guidelines provided in this letter.

*3) Include a revised version of any required Reporting Summary: <https://www.nature.com/documents/nr-reporting-summary.pdf>

Please be aware of our [guidelines](https://www.nature.com/nature-research/editorial-policies/image-integrity) on digital image standards.

Link Redacted

If you wish to submit a suitably revised manuscript we would hope to receive it within 6 months. If you cannot send it within this time, please let us know. We will be happy to consider your revision so long as nothing similar has been accepted for publication at Nature Genetics or published elsewhere. Should your manuscript be substantially delayed without notifying us in advance and your article is eventually published, the received date would be that of the revised, not the original, version.

Thank you for the opportunity to review your work.

Sincerely,

Safia Danovi, PhD
Senior Editor, Nature Genetics
ORCID: 0009-0007-7822-5479

Referee expertise:

Referee #1: blood trait GWAS

Referee #2: GWAS (incl. trans-ancestry)

Referee #3: CH genetics

Reviewers' Comments:

Reviewer #1:

Remarks to the Author:

This manuscript by Wen and colleagues presents an investigation of frequencies of clonal hematopoiesis (CH) in admixed American participants from the Mexico City Prospective Cohort (MCPS) and compares frequencies and distribution of CH to the primarily European ancestry UK Biobank (UKBB) study. Lower frequencies of CH are observed as percentage of European ancestry percentage decreases. The study also validates prior CH susceptibility loci in the MCPS as well as identified additional common and low-frequency/rare variants. Overall the study has well-defined aims, carries out analyses using appropriate methods and makes important inferences on CH that are supported by the data. The study is well-written and presents an important advance in the understanding of CH.

Below are a few substantive comments aimed at improving the overall quality and impact of the manuscript.

- More details on the use of peddy to infer genetic ancestry are needed for others to be able to reproduce this or similar analyses in the future, especially since the output is used as a primary variable in analyses.
- I suggest the use of "frequency" of CH over "prevalence" of CH as prevalence refers to the proportion of a defined population found to be affected by a condition at a specific time.
- I appreciate the level of detail that went into ensuring somatic variants were called similarly between MCPS and UKBB. That said, the authors do not discuss potential differences in sequencing chemistry or exome capture that could contribute to some (but likely not all) of the difference in CH frequency. I suggest some mention of this be added to the Discussion to provide a complete and balanced presentation.
- The fact that none of the CH genes examined were more common in the MCPS population is interesting. I suggest further highlighting in the manuscript.
- I found the paragraphs on telomere length and CH to be distinct enough from the current section that it should merit its own section detailing the relationship and highlighting important results/conclusions.
- It was not clear to me the GWAS section was performed using genotyping arrays until I read through the methods. I suggest

mentioning that in the GWAS results section as there is no prior mention of genotype data being available for the MCPS study.

-Figure 2b would benefit by adding strata of age to the CH frequency by European genome percentage to better show this CH-European ancestry result is not being confounded by differences in age distribution.

-Figure 3a and 3c are very hard to see in the current circular Manhattan plots. I suggest something more similar to that used in Figure 4.

Reviewer #2:

Remarks to the Author:

Wen & Kuri-Morales et al present an interesting and well-written paper that explores the genetic basis of clonal hematopoiesis (CH) in large and well-characterized cohort. Mexico City Prospective Study (MCPS) provides a rich dataset for filling a major research gap – the under-representation of admixed populations in this research area. Most of the statistical analyses seem appropriate and clearly described, although a bit more elaboration on variant calling methods and cohort processing would be helpful. The major concern with this work relates to the case definition, specifically the 15 gene panel used to define CH.

Specific comments:

1. What is the sequencing depth? Referenced papers say ~20X, but it would be good to include in this directly in manuscript as well. Extended Data Figure 3 partly addresses this, although there is significant difference in coverage for some genes
2. A panel of normals (PoN) was "created from 200 of the youngest UKB participants without a hematological malignancy", however CH variants, and particularly those specific to the UKB population, could still be present. Could MCPS participants have been included in the PoN? If not, would removal of variants derived only from UKB affect CH detection?
3. There may be a major issue here arising from the fact that the 15 gene catalog used to define CH was defined in UKB. These genes were selected because they demonstrate increased prevalence of driver variants with age. So naturally, we'd expect UKB patients to harbor these variants at a high prevalence relative to other genes (and even more-so with age). It could be problematic to apply these same 15 genes to the MCPS population, particularly given the differences in ancestry. This calls into question one of the main results that CH was less common in MCPS than UKB (OR=0.56, p<10⁻²⁰⁰) and the observation that EUR ancestry is associated with higher risk of CH.
4. Perhaps there are a different set of genes with high prevalence of driver variants in MCPS? On average actually fewer somatic variants per individual in UKB than MCPS (1.08 vs. 1.10): 4,656 somatic variants in 4,234 individuals in MCPS vs. 22,518 somatic variants in 20,786 UKB individuals.
5. It's not clear what the role of the 74-gene panel is. It seems this is the most comprehensive set of CH mutations, so how to does the 15 gene subset relate to this? Why not use information from the rest of the panel? How does the CH case definition compare with the TOPMed study (Bick et al. 2020)
6. Other comments/suggestions:
7. Please clarify whether the 9,649 MCPS individuals with WGS data used to derive the PRS for telomere length (TL) are distinct from the remaining 136,401 MCPS subjects. Assuming the training and testing subsets are non-overlapping, I suggest considering a genome-wide approach (eg: LDpred2, PRS-CS) rather than selecting suggestively associated (p <10⁻⁶) independent variants.
8. I would caution against over-interpreting associations between ancestry and genetically inferred TL: "MCPS individuals with higher proportions of European ancestry had longer genetically predicted telomere length". This could simply reflect the better predictive performance of the TL PRS in individuals with a greater proportion of EUR ancestry, rather than a true genetic difference. Please consider validating each TL PRS in an independent dataset.
9. Supplementary Table 9 is informative for comparing with previously published work, but it seems that conditional analyses should also be performed in the MCPS to characterize independent associations at the TERT locus in this population
10. Cross-ancestry meta-analysis: did the authors consider meta-analyzing or replicating some their findings in TOPMed (Bick et al. 2020), as this would include at least some Hispanic/Latino individuals with Indigenous ancestry?

Minor points:

11. Abstract: please include the number of CH cases, this is more informative for assessing the power of the study than the total sample size
12. I couldn't find any variant-level heterogeneity statistics in the cross-ancestry meta-analysis. Please include this information
13. I suggest toning down the conclusions in the abstract "... demonstrates the profound impact of ancestry on CH development"

Reviewer #3:

Remarks to the Author:

In this paper the investigators used whole exome sequencing (WES) to detect clonal hematopoiesis (CH) in a sample of people with admixed European and indigenous American ancestry from the Mexico City Prospective Study (MCPS). They compare CH in MCPS to CH detected previously in European ancestry individuals from the UK Biobank (UKB). Their most notable finding is that CH is very significantly more common in Europeans from the UKB than it is in MCPS participants. They carry out a GWAS in the MCPS and find ancestry-specific germline variants at TCL1B, close to but with separate effects from previously known variants near the neighbouring TCL1A gene. They detect several other novel GWAS variants, either through the MCPS alone or in a meta-analysis of MCPS and UKB.

Overall the paper is very nicely written, well illustrated and makes for an interesting read. The primary finding of a big difference in CH rates between European and indigenous American ancestries is, if true, an important contribution to our understanding of CH. Moreover, the finding of variants at *TCL1B* that demonstrate similar properties to those at *TCL1A* but nevertheless map to a distinct location raises interesting questions about which gene (or both) is implicated in the genesis of CH. Accordingly, I think the paper would appeal to a broad general audience.

However, there are areas of concern, some quite major. The reported difference in CH prevalence between MCPS and UKB is very dramatic (OR = 0.56, Pval = 1.60E-206). In my sorry experience, results of such magnitude are almost invariably traceable to a technical artefact. The onus is on the investigators to convince the reader of the biological veracity of their results.

1. According to Vlasschaert et al. (Ref 39), small changes in sequence processing can have large effects on the accuracy of CH calling. The primary whole exome sequencing was done outside of the authors' group, at Regeneron. It appears that FASTQ files were transferred to the authors' group at AstraZeneca and analysed further there. The authors need to provide a detailed account of how the sequence data for both MCPS and UKB were produced and processed. Such an account should be self-contained within this manuscript so that the reader does not have to trawl through the references to assemble an account of exactly what was done for each cohort. Any differences in sequence production and processing pipelines between MCPS and UKB should be highlighted and an explanation given as to why this would not affect the conclusions. A more detailed examination of the resulting mutation spectra in the two cohorts might be helpful here. I note also in this context that the description of CH mutation detection given in Dhindsa et al. (Ref 21) is a bit too sketchy for the current purposes.

2. The authors are clearly aware of this issue and they adopt the rather elegant solution of partitioning the MCPS subjects into those with predominantly European and predominantly indigenous American ancestry. They find that the difference in CH prevalence is maintained, albeit with reduced effect due to the admixture. On the face of it, this is pretty compelling evidence, but:

a. Their definition of CH requires the detection of one or more mutations that are on shortlist of 15 candidate driver genes with specified mutations. This shortlist was compiled from a much longer list of 74 candidate driver genes/mutations. The criteria for selecting the shortlist are not adequately described. The genes and mutations that constitute the shortlist are not identified in the manuscript. It appears that the shortlist was constructed using the predominantly European UKB sample. This could introduce a search bias in favour of CH mutations most commonly found in Europeans and in particular in UKB participants. Such a bias would be carried over into the ancestry partition of the MCPS. The result would be a higher estimated CH prevalence in the European ancestry partition of the MCPS and in the entire UKB sample.

b. MuTect2 uses a panel of normals to estimate per-site beta distribution parameters for use in refining somatic likelihood assignment (Kessler et al., Ref 16). In the present MCPS study, the investigators used a panel of normals comprised of 200 young UKB participants (i.e. Europeans) and applied it to both UKB and MCPS samples. This could create a somatic mutation calling bias in favour of Europeans and such a bias would again be carried over to the ancestry partition of the MCPS. Other investigators (e.g. Kessler et al.) generate separate panels of normals for each cohort.

c. Sequence read alignment used GRCh38 as a reference, which is essentially a European genome. MuTect2 uses mapping quality measures to generate confidence in somatic mutation identification. This could create a mapping quality bias in favour of Europeans which again would be carried over to somatic mutation calls in the ancestry partition of the MCPS.

3. Smoking is an important determinant of CH risk (Stacey et al., Ref 8). How smoking was taken into account in the analyses is not sufficiently explained in the manuscript. Moreover, smoking shows a strong association with proportion of European ancestry amongst Hispanic/Latino population samples, with a higher prevalence of smoking behaviour correlated with greater European ancestry (Choquet et al. 2021, PMID: 33633108). If not properly managed, this could cause an increase in apparent CH prevalence in Europeans due to residual confounding.

4. It is not clear precisely how the odds ratios comparing the CH risk in the UKB compared with the MCPS were arrived at. This is of particular concern because of the very different age structures between the two cohorts. The stated odds ratios do not seem to tally with the differences in prevalence as shown in Fig 1b. The authors need to show precisely what model they used and provide particular detail on how age and smoking covariates were handled. The same applies to the odds ratios derived from the ancestry partition of the MCPS.

5. A component with Mexican/Central American ancestry can be identified within the UKB cohort. Do the authors' findings replicate in an ancestry partition of the UKB sample?

6. The Methods describes only the exclusion of samples for technical reasons. Were individuals with prior neoplasia excluded from either of the cohorts? What ICD10 codes were used for exclusion? What other exclusion criteria were employed? Were they the same for both cohorts?

7. Page 8: It is unclear why the investigators choose to compare the LD of rs187319135 with rs10131341 rather than the more reproducibly associated *TCL1A* variant rs2887399. Are the LD relationships similar? It is not clear in the text whether the D-prime and r-squared values are derived from the MCPS sample.

8. It is fine for the authors to document their observations for the *CSGALNACT1* and *DIAPH3* loci. However, these associations arise from what is essentially a subgroup analysis (*ASXL1-CH* and *TP53-CH* subgroups). The levels of

significance would not withstand a full Bonferroni adjustment. The associations do not appear to replicate well in the UKB, even though the variants are common enough there. The authors might want to add a “health warning” concerning the robustness of the observations for CSGALNACT1 and DIAPH3. The statement that the DIAPH3 variant “conferred an increased risk exclusively of TP53-CH” isn’t really true, as Extended Data Figure 9 shows there is some risk also of PPM1D-CH in MCPS.

9. The Discussion consists primarily of a restatement of the study findings. The authors claim that they have made “novel insights into CH pathogenesis” and revealed “fundamental biological insights”. What are they?

How, for example, do the authors think that the difference in prevalence in CH between ancestries might come about? Are there differences in mutation rate, or clonal selection pressure, or what? Do the authors think that some germline variants promote CH through their effects on TCL1B, or TCL1A, or both? While I understand that the authors may not wish to be drawn into unbridled speculation, without some level of interpretation their findings seem to be reduced to the level of mere phenomenology.

Minor Points:

10. Figure 1b and others: the 95% confidence interval shadings are not visible. This may be simply due to how the graphic objects are rendered in MS-Word. Nevertheless, the reviewers need to be able to see them.

11. Extended Data Figure 5b: It is not very clear how the OR on the Y axis were determined here. A better explanation or an alternative presentation is required.

12. Extended Data Figures 8d and 13a: Both red and blue bars are defined as “before conditioning” in the legends.

13. Pg 10, last sentence: “Notably, CHEK2 c.1100del constituted...”. Don’t you mean c.1100delC here?

14. Pg 12, first sentence: “promoter risk variant (rs774615666) were less common in Indigenous American...” Don’t you mean “more common” here?

15. Pg 19, Data availability: “Summary statistics for GWAS, ExWAS, gene burden association analysis, ... are provided in the Supplementary Tables” The full summary data do not appear to be in the Supplementary Tables. Are the authors planning to release them?

Version 1:

Decision Letter:

15th Oct 2024

Dear Dr Mitchell,

Your Article, "Comparative analysis of 136,401 Admixed Americans and 416,118 Europeans reveals ancestry-specific genetic determinants of clonal haematopoiesis" has now been seen by 3 referees. You will see from their comments below that Reviewer #1 has asked for one last analysis. We are interested in the possibility of publishing your study in Nature Genetics, but would like to consider your response to these concerns in the form of a revised manuscript before we make a final decision on publication.

We therefore invite you to revise your manuscript taking into account the feedback by Reviewer #1. Please highlight all changes in the manuscript text file. At this stage we will need you to upload a copy of the manuscript in MS Word .docx or similar editable format.

*1) Include a “Response to referees” document detailing, point-by-point, how you addressed each referee comment. If no action was taken to address a point, you must provide a compelling argument. This response will be sent back to the referees along with the revised manuscript.

*2) If you have not done so already please begin to revise your manuscript so that it conforms to our Article format instructions, available

http://www.nature.com/ng/authors/article_types/index.html>here.

*3) Include a revised version of any required Reporting Summary: <https://www.nature.com/documents/nr-reporting-summary.pdf>

Link Redacted

We hope to receive your revised manuscript within four to eight weeks. If you cannot send it within this time, please let us know.

Sincerely,

Safia Danovi, PhD
Senior Editor, Nature Genetics
ORCID: 0009-0007-7822-5479

Reviewers' Comments:

Reviewer #1 (Remarks to the Author):

I commend the authors on thoughtfully addressing my comments with a substantially improved manuscript. I appreciate the point raised by Reviewer 3 with respect to confounding of the ancestry-CH relationship by smoking. Smoking is difficult to capture in population-based investigations as measures of ever-never or current/former/never do not capture important features such as duration, intensity or type of tobacco used. To provide further support of the robustness of the findings, I suggest a sensitivity analysis performed only among never smokers to demonstrate a similar ancestry effect in this "clean" subset. Likewise, I suggest adding a sentence to the Discussion section to indicate the possibility of residual confounding by unmeasured components of tobacco smoking such as duration of smoking, smoking intensity, etc.

Reviewer #2 (Remarks to the Author):

I appreciate the extensive additional analyses, particularly the analyses based on the 58 gene panel. The authors have done an excellent job addressing my concerns. I have no further comments.

Reviewer #3 (Remarks to the Author):

No further comments.

Version 2:

Decision Letter:

Our ref: NG-A64635R1

1st Nov 2024

Dear Dr Mitchell,

Thank you for submitting your revised manuscript "Comparative analysis of 136,401 Admixed Americans and 416,118 Europeans reveals ancestry-specific genetic determinants of clonal haematopoiesis" (NG-A64635R1). It has now been seen by Reviewer #1 and their comments are below. The reviewers find that the paper has improved in revision, and therefore we'll be happy in principle to publish it in Nature Genetics, pending minor revisions to satisfy our editorial and formatting guidelines.

Sincerely,

Safia Danovi, PhD
Senior Editor, Nature Genetics
ORCID: 0009-0007-7822-5479

Reviewer #1 (Remarks to the Author):

The authors have fully addressed my concerns. I have no further comments.

Reviewer #1:

Remarks to the Author:

This manuscript by Wen and colleagues presents an investigation of frequencies of clonal hematopoiesis (CH) in admixed American participants from the Mexico City Prospective Cohort (MCPS) and compares frequencies and distribution of CH to the primarily European ancestry UK Biobank (UKBB) study. Lower frequencies of CH are observed as percentage of European ancestry percentage decreases. The study also validates prior CH susceptibility loci in the MCPS as well as identified additional common and low-frequency/rare variants. Overall the study has well-defined aims, carries out analyses using appropriate methods and makes important inferences on CH that are supported by the data. The study is well-written and presents an important advance in the understanding of CH.

Below are a few substantive comments aimed at improving the overall quality and impact of the manuscript.

We thank the Reviewer for the positive review of our study and appreciate their highlighting the quality of the analyses and importance of the results. We are also grateful for the comments on how to improve the manuscript. Each comment has been addressed in detail below and we have updated the manuscript accordingly. This has, we believe, strengthened our study and we look forward to the reviewer's further consideration of our work.

1. More details on the use of *peddy* to infer genetic ancestry are needed for others to be able to reproduce this or similar analyses in the future, especially since the output is used as a primary variable in analyses.

To ensure reproducibility as well as to enable similar analyses in the future, we have now provided additional details on how *peddy* was used to infer genetic ancestry in our study. In the **Results section ("Frequency of CH", paragraph 1)** of the revised manuscript this is summarised as

"Genetic ancestry at continent level was determined for MCPS and UKB participants from whole exome sequencing (WES) data using *peddy*¹⁸, a machine

learning classifier trained on 2,504 individuals of known ancestry from the 1000 Genomes Project¹⁹. Further analyses were then restricted to 136,401 MCPS individuals with admixed Indigenous American, European, and African ancestry ($\geq 95\%$ *peddy*-predicted probability Admixed American) and 416,118 UKB individuals ($\geq 95\%$ *peddy*-predicted probability European) who passed all quality control filters (see Methods and Supplementary Tables 1-3)."

Additionally, details are provided in the **Methods section ("Sample selection", paragraph 2)**:

"For each individual in the study, *peddy* was used to predict their genetic ancestry with a machine learning classifier trained on 2,504 individuals of known ancestry from the 1000 Genomes project^{18,19}. The classifier was trained on a set of approximately 25,000 bi-allelic sites in the 1000 Genomes project which were also present in the MCPS and UKB WES data, by first performing randomised PCA and then training an SVM on the first four principal components. Once trained, the SVM classifier was applied to the WES germline variant calls of each individual in our study after being projected onto the principal components calculated from the 1000 Genomes project samples. This generated the most likely ancestry (AFR, AMR, EAS, EUR or SAS) for each individual along with a probability defining the certainty of the prediction."

2. I suggest the use of "frequency" of CH over "prevalence" of CH as prevalence refers to the proportion of a defined population found to be affected by a condition at a specific time.

Indeed, as MCPS and UKB participants were recruited over a period of time. Specifically 1998-2004 for MCPS (PMID: 37821707; PMID: 16556648) and 2006-2010 for UKB (PMID: 34183854; PMID: 30305743). We have now replaced "prevalence" with "frequency" throughout the revised manuscript.

3. I appreciate the level of detail that went into ensuring somatic variants were called similarly between MCPS and UKBB. That said, the authors do not discuss potential

differences in sequencing chemistry or exome capture that could contribute to some (but likely not all) of the difference in CH frequency. I suggest some mention of this be added to the Discussion to provide a complete and balanced presentation.

We thank the reviewer for raising this issue. Both the MCPS and UKB studies used the same sequencing chemistry and exome capture kit, and in the revised manuscript, we are now discussing this and the potential limitations of a cross-study comparison in the Discussion (paragraph 2):

“A potential limitation of comparing CH frequency between MCPS and UKB is the technical differences in the sequencing and bioinformatics pipelines^{44,45}. This was mitigated as far as possible by whole-exome sequencing DNA libraries being prepared by the same exome capture kit (IDT xGen v1 capture kit), sequenced on the same platform (NovaSeq6000) with the same sequencing mode (75-bp paired-end mode) by the same sequencing provider (Regeneron Genetics Centre)^{23,46,47}, and subjected to the same germline (Illumina DRAGEN Bio-IT Platform Germline Pipeline v3.0.7) and somatic (MuTect2) variant calling pipeline^{20,21,48}. Collectively, this resulted in similar somatic variant allele frequencies and coverage profiles observed between MCPS and UKB samples. While technical variation between MCPS and UKB affecting CH detection cannot be entirely ruled out, we replicated the difference in CH frequency within the MCPS study itself, where we observed that increased CH risk was associated with a higher fraction of European compared to Indigenous American ancestry.”

4. The fact that none of the CH genes examined were more common in the MCPS population is interesting. I suggest further highlighting in the manuscript.

We agree that this is a notable finding that warrants highlighting in the manuscript. In addition to detailing this in the Results, we have now highlighted it in the Discussion (paragraph 1) of the revised manuscript as follows:

“...Notably, none of the CH genes were more common in MCPS...”

5. I found the paragraphs on telomere length and CH to be distinct enough from the current section that it should merit its own section detailing the relationship and highlighting important results/conclusions.

We agree that our analyses of telomere length and CH merits its own section, and thank the reviewer for the suggestion. We have now reported our findings on the relationship between telomere length and CH under "Association between telomere length, ancestry, and CH frequency" in the Results section of the revised manuscript. In line with this, three panels from Supplementary Figures have now been used to create Fig. 3:

Figure	Description
3a	Manhattan plot of telomere length GWAS among 9,598 MCPS participants with whole-genome sequencing data
3b	Telomere length PRS stratified by proportion of European genome
3c	Association between telomere length PRS and CH frequency

6. It was not clear to me the GWAS section was performed using genotyping arrays until I read through the methods. I suggest mentioning that in the GWAS results section as there is no prior mention of genotype data being available for the MCPS study.

We have now mentioned that germline variants for both MCPS and UKB were genotyped with SNP arrays in the Results (MCPS: "Genome-wide common variant associations with CH" section, paragraph 1; UKB: "Cross-ancestry meta-analysis of CH" section, paragraph 1) of the revised manuscript as follows:

"...MCPS germline variants included for GWAS were genotyped with SNP arrays and subsequently imputed with the TOPMed reference panel (see Methods)..."

“Cross-ancestry GWAS meta-analysis across UKB (imputed with the Haplotype Reference Consortium (HRC) and UK10K + 1000 Genomes panel³⁶; see Methods)...”

This clarifies that germline variants from both cohorts were derived from the same technology, i.e. SNP array, rather than whole-genome sequencing in either one or both cohorts.

7. Figure 2b would benefit by adding strata of age to the CH frequency by European genome percentage to better show this CH-European ancestry result is not being confounded by differences in age distribution.

We have now stratified the CH frequency by European genome percentage with age quartiles (Extended Data Fig. 6b-e) in the revised manuscript. The trend of increasing CH frequency with increasing European genome percentage was observed among individuals in the 2nd, 3rd, and 4th quartiles.

It is noteworthy that to account for age (in addition to sex and smoking status) as potential confounders in the association between European genome percentage and CH frequency, we have included age, sex, and smoking status as co-variables in the logistic regression model (Extended Data Fig. 6j and k). Reassuringly, the trend of increasing CH frequency with increasing European genome percentage was retained after adjusting for these co-variables. For example, the group of individuals with the highest percentage of European genome ($\geq 70\%$) had the highest risk of CH compared to the group of individuals with the lowest percentage of European genome ($< 10\%$, OR [95% CI] = 1.87 [1.46, 2.39], $P = 4.24 \times 10^{-7}$)

8. Figure 3a and 3c are very hard to see in the current circular Manhattan plots. I suggest something more similar to that used in Figure 4.

We have now replaced the circular plots (Previously Figures 3a and 3c) with regular Manhattan plots in the revised manuscript (Fig. 4a and c, Extended Data Fig. 9a-c) to make them clearer.

Reviewer #2:

Remarks to the Author:

Wen & Kuri-Morales et al present an interesting and well-written paper that explores the genetic basis of clonal hematopoiesis (CH) in large and well-characterized cohort. Mexico City Prospective Study (MCPS) provides a rich dataset for filling a major research gap – the under-representation of admixed populations in this research area. Most of the statistical analyses seem appropriate and clearly described, although a bit more elaboration on variant calling methods and cohort processing would be helpful. The major concern with this work relates to the case definition, specifically the 15 gene panel used to define CH.

We thank the Reviewer for the positive assessment of our study and for highlighting that our analyses of clonal haematopoiesis in an under-represented population fills a gap in the research. The comments to improve the manuscript, and the concerns raised are appreciated, and have been addressed in the revised manuscript as detailed below. In particular, our new analysis on an expanded 58-CH gene panel (PMID: 36652671), where the principal results of our study were replicated, increases the robustness of our findings. We believe the Reviewer's concerns have been addressed and look forward to any further comments they may have.

Specific comments:

1. What is the sequencing depth? Referenced papers say ~20X, but it would be good to include in this directly in manuscript as well. Extended Data Figure 3 partly addresses this, although there is significant difference in coverage for some genes.

We have now reported this metric in the Methods ("Whole-exome sequencing pre-processing and variant calling", paragraph 1) section of the revised manuscript as follows:

“...The average sequencing depth across UKB and MCPS samples was 57.8x and 57.2x, respectively. This metric was computed as the average alignment coverage over the consensus coding sequence (CCDS) located on the autosomes...”

It is noteworthy that the UKB Exome Sequencing Consortium previously reported 95.8% of targeted bases were covered at a depth of 20x or greater (PMID: 34662886). In our study, we similarly observed 97.1% bases across the 15-CH gene panel were covered at a depth of 20x or greater in UKB and MCPS samples (Extended Data Fig. 1g).

2. A panel of normals (PoN) was "created from 200 of the youngest UKB participants without a hematological malignancy", however CH variants, and particularly those specific to the UKB population, could still be present. Could MCPS participants have been included in the PoN? If not, would removal of variants derived only from UKB affect CH detection?

We thank the reviewer for raising an issue that could potentially have created a discrepancy between the detection of CH in the UKB and MCPS. We made the decision to use the same PoN (created from UKB) for both UKB and MCPS as it is designed to filter artifacts, and these should be similar between the two cohorts as they underwent the same sequencing protocol and were analysed with the same bioinformatics pipeline. Therefore, we anticipate minimal bias in CH detection when the UKB-derived PoN was used for CH detection among MCPS samples. Nevertheless, it is conceivable that using the UKB PoN for the detection of CH variants in MCPS samples may have led to a bias between MCPS and UKB in the reported frequency of CH.

To test whether there was any bias, we inspected variants filtered by the PoN in MCPS which would have otherwise been considered CH driver variants in our study. In the 15-gene panel, we observed only 3 CH qualifying variants filtered by Mutect2's PoN which would have otherwise been PASS. One of these variants was also filtered by the PoN in a UKB participant, while 2 were specific to MCPS as follows:

Chr	Position	Ref. allele	Alt. allele	Gene	Consequence	No. of individuals
2	25246633	CAGGCC CTTAGGG CCAGAAG GCTG	C	DNMT3A	Frameshift deletion	1
4	105237059	GTTAT	G	TET2	Frameshift deletion	1

Inclusion of the 2 individuals with PoN-flagged variants as CH samples yielded a CH frequency of 3.117% (4,251 / 136,401), which represents a very small increase from the original CH frequency of 3.115% (4,249 / 136,401) reported in our manuscript. Therefore, PoN filtering does not account for the lower frequency of CH observed in MCPS compared to UKB.

When we extended our analysis to the 58-gene panel (Vlasschaert *et al.*; PMID: 36652671), we observed 10 variants flagged by PoN, of which 7 were also filtered in UKB participants, while 3 were specific to MCPS as follows:

Chr	Position	Ref. allele	Alt. allele	Gene	Consequence	No. of individuals
2	25246633	CAGGCC CTTAGGG CCAGAAG GCTG	C	DNMT3A	Frameshift deletion	1
4	105237059	GTTAT	G	TET2	Frameshift deletion	1
16	3729209	T	TG	CREBBP	Frameshift insertion	1

Inclusion of the 3 individuals with PoN-flagged variants as CH samples yielded CH frequency of 3.621% (4,939 / 136,401), which represents only a marginal increase from the original CH frequency of 3.619% (4,936 / 136,401) reported in our manuscript.

It is also noteworthy that the rate of PoN-flagged variants in UKB was similar to MCPS. In the 15-gene panel, we observed only 5 CH qualifying variants filtered by Mutect2's PoN which would have otherwise been PASS. One of these variants was also filtered by the PoN in a MCPS participant, while 4 were specific to UKB as follows:

Chr	Position	Ref. allele	Alt. allele	Gene	Consequence	No. of individuals
4	105269705	T	A	TET2	Missense	1

20	32434638	AGG	A	ASXL1	Frameshift deletion	5
20	32434638	A	AT	ASXL1	Frameshift insertion	2
20	32434638	A	AGAGG	ASXL1	Frameshift insertion	1

Of the 9 individuals with PoN-flagged variants above, 2 individuals already had qualifying CH variants identified. Specifically, 1 individual (with 4-105269705-T-A variant) already had qualifying variants in *DNMT3A* and *TET2* identified, and another 1 individual (with 20-32434638-AGG-A variant) already had a qualifying variant in *TET2* identified. Therefore, 7 additional CH individuals would be identified with these PoN-flagged variants. Nevertheless, inclusion of these 7 individuals with PoN-flagged variants as CH samples yielded CH frequency of 4.925% (20,495 / 416,118), which represents only a marginal increase from the original CH frequency of 4.924% (20,488 / 416,118) reported in our manuscript.

When we extended our analysis to the 58-gene panel (Vlasschaert *et al.*; PMID: 36652671), we observed 20 variants flagged by PoN, of which 7 were also filtered in MCPS participants, while 13 were specific to UKB as follow:

Chr	Position	Ref. allele	Alt. allele	Gene	Consequence	No. of individuals
4	105269705	T	A	TET2	Missense	1
20	32434638	AGG	A	ASXL1	Frameshift deletion	5
20	32434638	A	AT	ASXL1	Frameshift insertion	2
20	32434638	A	AGAGG	ASXL1	Frameshift insertion	1
13	32758086	G	T	PDS5B	Splice site	1
12	11884510	C	T	ETV6	Nonsense	1
7	139409618	A	AAGAG	LUC7L2	Frameshift insertion	1
X	124061770	G	T	STAG2	Splice site	1
X	124061770	G	A	STAG2	Splice site	3
X	124066174	G	T	STAG2	Splice site	31
1	150961067	C	A	SETDB1	Nonsense	3
1	150961067	C	G	SETDB1	Nonsense	1
1	150961084	G	T	SETDB1	Nonsense	2

Of the 53 individuals with PoN-flagged variants above, 6 individuals already had qualifying CH variants identified. Specifically, 1 individual (with 7-139409618-A-AAGAG variant) already had qualifying variants in *DNMT3A* identified, another 3 individuals (with 20-32434638-AGG-A, X-124066174-G-T, or X-124066174-G-T variant) already had a qualifying variant in *TET2* identified, another 1 individual

(with 4-105269705-T-A variant) already had qualifying variants in *DNMT3A* and *TET2* identified, and another 1 individual (with X-124066174-G-T variant) already had qualifying variants in *BCORL1* identified. Therefore, 47 additional CH individuals would be identified with these PoN-flagged variants. Nevertheless, inclusion of these 47 individuals with PoN-flagged variants as CH samples yielded CH frequency of 5.673% (23,605 / 416,118), which represents only a marginal increase from the original CH frequency of 5.661% (23,558 / 416,118) reported in our manuscript.

3. There may be a major issue here arising from the fact that the 15 gene catalog used to define CH was defined in UKB. These genes were selected because they demonstrate increased prevalence of driver variants with age. So naturally, we'd expect UKB patients to harbor these variants at a high prevalence relative to other genes (and even more-so with age). It could be problematic to apply these same 15 genes to the MCPS population, particularly given the differences in ancestry. This calls into question one of the main results that CH was less common in MCPS than UKB (OR=0.56, $p < 10^{-200}$) and the observation that EUR ancestry is associated with higher risk of CH.

The 15-gene panel used in our study captures the majority of CH cases and includes the three most common CH drivers, namely *DNMT3A*, *TET2* and *ASXL1* that between them account for >75% of all cases in UKB (PMID: 36652671; PMID: 37726541). The canonical CH driver genes (*DNMT3A*, *TET2*, *ASXL1*), are known to be orders of magnitude more common than the least common driver genes (e.g., *NRAS*, *BRAF*, *MPL*), and we found this to be true also in MCPS. We believe that the lower frequency of CH driven by these canonical driver genes in MCPS to be an important result in itself, and that it is unlikely there are genes to be discovered which are similarly common.

Nevertheless, we do recognise that there could have been a bias in our selection of 15 genes and we have gone on to test the impact of extending our variant calling to include the 58 genes validated by Vlasschaert *et al* (PMID: 36652671) as CH drivers. The results for overall CH defined by these 58 genes are now reported alongside the 15-gene panel in the revised manuscript (15-gene panel: Fig. 1c; 15-

and 58-gene panel: Supplementary Table 8). Consistent with our finding for the 15-gene panel, the 58-gene panel showed CH was less common in MCPS compared to UKB after adjusting for age, sex, and smoking status:

CH (outcome)	OR [95% CI]	P value
15-gene panel	0.59 [0.57,0.61]	7.31×10^{-185}
58-gene panel (Vlasschaert et al.)	0.62 [0.60,0.64]	5.11×10^{-179}

Similarly, in our intra-population analysis of MCPS individuals, we observed a higher frequency of CH among individuals with a higher proportion of European genome compared to individuals with a higher proportion of non-European (American/African) genome after adjusting for age, sex, and smoking status, when overall CH was defined with the 15- or 58-gene panel (15-gene panel: Fig. 2d; 15- and 58-gene panel: Supplementary Table 9):

CH (outcome)	Beta [95% CI]	P value
15-gene panel	0.84 [0.66,1.03]	7.35×10^{-19}
58-gene panel (Vlasschaert et al.)	0.71 [0.54,0.88]	1.41×10^{-15}

- Perhaps there are a different set of genes with high prevalence of driver variants in MCPS? On average actually fewer somatic variants per individual in UKB than MCPS (1.08 vs. 1.10): 4,656 somatic variants in 4,234 individuals in MCPS vs. 22,518 somatic variants in 20,786 UKB individuals.

Indeed, in our revised manuscript, we observed fewer somatic variants per individuals in UKB vs. MCPS (1.082 vs. 1.101): 4,678 somatic variants in 4,249 CH individuals in MCPS vs. 22,161 somatic variants in 20,488 CH individuals in UKB. After extending our analysis to include all 58 CH genes (Vlasschaert *et al.*; PMID: 36652671), we similarly observed fewer somatic variants per individuals in UKB vs. MCPS (1.107 vs. 1.117): 5,514 somatic variants in 4,936 CH individuals in MCPS vs. 26,083 somatic variants in 23,558 CH individuals in UKB.

The higher number of somatic variants per individual in MCPS is likely attributed to the recruitment of individuals >70 years old into MCPS, whereas the age range of UKB participants were 40 to 70 years old, and by extension the mean age of CH participants in MCPS was older compared to UKB (67.0 vs. 56.8, respectively). We confirmed this by age-matching both cohorts, and observed a very similar number of somatic variants per CH individual in UKB vs MCPS (1.078 vs. 1.077): 1,827 somatic variants in 1,697 CH individuals in MCPS vs. 17,429 somatic variants in 16,170 CH individuals in UKB.

We nevertheless agree that it is conceivable that there are CH genes that may be more frequent in MCPS than UKB. After extending our analysis to include all 58 CH genes defined by (Vlasschaert *et al.*; PMID: 36652671), we observed only one CH gene to be more common in MCPS than UKB with frequency of 0.014% and 0.003%, respectively (*CBL*: OR [95% CI] = 5.88 [2.69, 12.9], $P = 5.16 \times 10^{-4}$). Therefore, even when using the 58-gene panel, the frequency of CH remained statistically more common in UKB relative to MCPS, and also among MCPS individuals with higher proportion of European genome (please also see response to comment 3).

5. It's not clear what the role of the 74-gene panel is. It seems this is the most comprehensive set of CH mutations, so how does the 15 gene subset relate to this? Why not use information from the rest of the panel? How does the CH case definition compare with the TOPMed study (Bick et al. 2020)

The 74-gene panel used in the TOPMed study (PMID: 33057201) was a curated list that also contained leukaemia-defining mutations such as *NPM1* and *FLT3*. However, CH is by definition a preleukaemic condition of HSC expansion in the absence of overt blood cancers such as acute myeloid leukaemia (AML). The same team of researchers subsequently refined their list of 74 CH genes in the TOPMed study to a panel of 58 genes that met criteria for (1) increasing somatic mutation frequency with increasing age or (2) myeloid preleukaemic and leukaemic clonal selection (PMID: 36652671).

The genes that constitute the 15-gene panel in our study were those from the 74 genes in the TOPMed study where we found an association between age and the frequency of putative drivers (Extended Data Fig. 4a and b, and Supplementary Table 7). However, as described in our response to comment 3, we have now replicated the population difference in CH frequency in an expanded CH gene set using the 58-gene panel (Vlasschaert *et al.*; PMID: 36652671).

One potential problem with the addition of another 43 genes that are relatively rare drivers of CH, is that the definition of individual variants as driver mutations, passenger mutations or errors, can be difficult to ascertain, due to the lack of recurrence. This can “contaminate” the list of CH carriers with individuals that do not have *bona fide* CH. Indeed, genetic association analyses in UKB demonstrated an attenuation of associations of common germline variants with overall CH for the 58-gene panel relative to the 15-gene panel. Specifically, GWAS of CH limited to the 15-gene panel identified 14 genome-wide significant loci ($P < 5 \times 10^{-8}$), whereas GWAS for 58-gene panel CH identified only 12 genome-wide significant loci. More importantly, of the 11 overlapping significant loci between the two panels, the effect sizes and P values were attenuated in the 58-gene panel relative to the 15-gene panel CH. This may be indicating that the 58-gene panel, while identifying more individuals with *bona fide* CH may also contain a non-negligible proportion of misidentified individuals.

UKB GWAS of CH (as defined with 15-gene panel)

UKB GWAS of CH (as defined with 58-gene panel)

Locus	15-Gene panel			58-Gene panel (Vlasschaert et al.)		
	Top SNP (rs ID)	OR [95% CI]	P value	Top SNP (rs ID)	OR [95% CI]	P value
PARP1	rs138994074	0.916 [0.89,0.944]	5.81E-09	rs1136410	0.926 [0.902,0.951]	1.87E-08
MYCN	rs12471506	1.06 [1.04,1.09]	3.66E-08	Did not reach genome-wide significance ($P \geq 5E-08$)		
LY75,CD302	rs1549387	1.06 [1.04,1.09]	3.19E-09	rs1549387	1.06 [1.04,1.08]	2.51E-08
TRIM59,IFT80	rs56658671	1.13 [1.1,1.15]	1.77E-30	rs56658671	1.11 [1.09,1.13]	4.63E-28
CXXC4,TET2	rs116597408	1.19 [1.13,1.25]	3.04E-10	rs116597408	1.17 [1.11,1.23]	9.21E-10
TERT	rs7705526	1.26 [1.24,1.29]	4.15E-103	rs7705526	1.24 [1.22,1.27]	4.30E-102
HCG26,MICB-DT,HLA-C	rs3131643	0.918 [0.892,0.944]	2.48E-09	Did not reach genome-wide significance ($P \geq 5E-08$)		
CD164	rs12193493	0.904 [0.885,0.924]	1.19E-19	rs2275652	0.916 [0.897,0.936]	2.44E-16
STN1	rs34763036	0.91 [0.88,0.94]	2.06E-08	Did not reach genome-wide significance ($P \geq 5E-08$)		
NPAT,ATM	rs228606	1.1 [1.08,1.13]	8.36E-21	rs7129527	1.09 [1.07,1.11]	1.52E-17
ITPR2	rs149752564	1.14 [1.1,1.18]	2.26E-11	rs149752564	1.13 [1.09,1.17]	1.77E-11
MSI2	rs118121072	0.867 [0.825,0.911]	8.83E-09	rs188761458	0.822 [0.767,0.882]	2.02E-08
SETBP1	rs1849209	0.895 [0.874,0.916]	2.13E-20	rs8088824	0.909 [0.889,0.929]	2.08E-17
RUNX1	rs2834707	1.06 [1.04,1.09]	4.29E-09	rs2834706	1.06 [1.04,1.08]	2.90E-08
MIR4708,FUT8	Did not reach genome-wide significance ($P \geq 5E-08$)			rs4899178	1.06 [1.04,1.08]	2.11E-08

Comparing MCPS CH GWAS based on the 15- and 58-gene panel demonstrated comparable effect sizes and P values for the two genome-wide significant loci (*TERT* and *TCL1A-TCL1B*). Therefore, the association results were not

significantly affected by extending the analysis from the 15-gene to the 58-gene panel.

Locus	15-Gene panel			58-Gene panel (Vlasschaert et al.)		
	Top SNP (rs ID)	OR [95% CI]	P value	Top SNP (rs ID)	OR [95% CI]	P value
TERT	rs2853677	0.766 [0.728,0.805]	1.62E-24	rs2853677	0.777 [0.742,0.814]	3.35E-25
TCL1B	rs968294563	1.78 [1.49,2.13]	2.88E-09	rs968294563	1.68 [1.42,1.99]	1.67E-08

Our findings suggest that neither the 15-gene nor the 58-gene panel calls represents an entirely accurate and comprehensive set of CH gene variants. Therefore, we have included analysis using putative driver variants from both the 15- and the 58-gene panels, and reported findings from both panels in the revised manuscript.

Other comments/suggestions:

- Please clarify whether the 9,649 MCPS individuals with WGS data used to derive the PRS for telomere length (TL) are distinct from the remaining 136,401 MCPS subjects. Assuming the training and testing subsets are non-overlapping, I suggest considering a genome-wide approach (eg: LDPre2, PRS-CS) rather than selecting suggestively associated ($p < 10^{-6}$) independent variants.

We thank the reviewer for highlighting the lack of clarity on whether the MCPS individuals with WGS data was a subset or distinct from the 136,401 MCPS subjects. In our revised manuscript, among the 136,401 MCPS subjects included in our study, 9,598 individuals with coverage-adjusted telomere length derived from WGS data (PMID: 37821707; PMID: 39192095) were included for our telomere length GWAS, with the purpose of building the PRS model for the remaining 126,803 subjects. Therefore, these 9,598 MCPS individuals with WGS data were a subset of the 136,401 MCPS subjects and subsequently excluded from downstream telomere length PRS-related association analysis in Fig. 3b and 3c and Extended Data Fig. 8c-d. This has now been clarified in the Methods ("Telomere length polygenic risk score", paragraph 1 and 2) as follows, and in the captions of Fig. 3 and Extended Data Fig. 8:

“Leukocyte telomere length (LTL) was inferred in 9,602 individuals with WGS data available²³ using coverage-normalised TelSeq measurements^{28,29}. Specifically, TelSeq was used to infer telomere length from WGS data and was further normalised with sample-specific sequencing coverage. Of the 9,602 individuals with WGS data, 9,598 individuals with LTL within ± 3 standard deviation were included for LTL GWAS. REGENIE⁶¹ was used for genetic association analysis with age, sex and the first ten genetic principal components included as covariates as described above. No evidence of genomic inflation was observed (inflation factor = 1.07).

Variants with MAF $\geq 1\%$ and imputation score ≥ 0.3 were retained and the GWAS summary statistics of retained variants were subsequently used to compute the LTL polygenic risk scores (PRS) in the remaining 126,803 MCPS participants...”

As the training subset (9,598 individuals with WGS data included for telomere length GWAS) and testing subset (126,803 subjects in which the telomere length PRS were computed) are non-overlapping, we have now applied a genome-wide approach (PRS-CS; PMID: 30992449; PMID: 33945532), in lieu of top SNPs from each suggestive locus ($P < 5 \times 10^{-6}$), for PRS calculation and downstream association analyses in our revised manuscript.

7. I would caution against over-interpreting associations between ancestry and genetically inferred TL: “MCPS individuals with higher proportions of European ancestry had longer genetically predicted telomere length”. This could simply reflect the better predictive performance of the TL PRS in individuals with a greater proportion of EUR ancestry, rather than a true genetic difference. Please consider validating each TL PRS in an independent dataset.

While we demonstrated that MCPS individuals with higher proportion of European genome had longer genetically predicted telomere length based on PRS derived from MCPS individuals with WGS-inferred telomere length (Fig. 3b), we agree that it is conceivable that this observation may be a reflection of better predictive performance of the PRS model in individuals with higher proportion of European

genome. Indeed, we similarly observed MCPS individuals with higher proportion of European genome had longer genetically predicted telomere length based on PRS derived from European-dominant UKB cohort (PMID: 34611362; Extended Data Fig. 8d).

To this end, we validated our MCPS-derived telomere length PRS in the varying ancestry groups among UKB participants. We reasoned that if our MCPS-derived telomere length PRS is specific to Admixed Americans, we anticipate the correlation between telomere length PRS versus WGS-inferred telomere length to be higher among Admixed Americans compared to Europeans or other ancestry groups in UKB (PMID: 31346163).

To test this hypothesis in the UKB, we fitted a linear regression model in each ancestry group with WGS-inferred telomere length as the outcome and MCPS-derived telomere length PRS as the predictor, adjusted for age, sex, smoking status, and first four genetic principal components. The effect size of MCPS-derived telomere length PRS was used as a metric for the correlation between MCPS-derived telomere length PRS and WGS-inferred telomere length. Reassuringly, we observed the highest effect size for MCPS-derived telomere length PRS in Admixed Americans in UKB (beta = 0.127), followed by East Asians (0.054), Africans (0.031), Europeans (0.026), and South Asians (0.024; Extended Data Fig. 8e).

We additionally fitted a second model where we excluded telomere length PRS as a co-variate but still included age, sex, smoking status, and first four genetic principal components. The % increase in R-squared value for the 1st model (with PRS as co-variate) relative to the 2nd model (without PRS as co-variate) was computed. A bigger % increase in R-squared value would indicate more WGS-inferred telomere length variance explained by telomere length PRS, and by extension, better performance of the telomere length PRS. Reassuringly, we observed the largest increase in R-squared value among Admixed Americans (17.23%) in UKB, followed by East Asians (4.32%), Europeans (1.23%), Africans (0.92%), and South Asians (0.69%; Extended Data Fig. 8f).

8. Supplementary Table 9 is informative for comparing with previously published work, but it seems that conditional analyses should also be performed in the MCPS to characterize independent associations at the *TERT* locus in this population.

We have now performed conditional analysis at the *TERT* locus to identify independent associations in MCPS. Our analysis revealed no independent associations at the *TERT* locus. Therefore, the top variant (rs2853677) at this locus remained the only independent variant associated with CH risk in MCPS. We have updated the **Results** (“Genome-wide common variant associations with CH” section, paragraph 1) with this finding, and also provide the reviewer with the association results after conditioning the 41 variants on the top variant (rs2853677) below:

locus	rs ID	OR [95% CI]	P value (Ranked from most-to-least significant)
TERT	rs402710	0.898 [0.856,0.943]	1.56E-05
TERT	rs27919	1.139 [1.074,1.208]	1.72E-05
TERT	rs7705526	1.153 [1.08,1.229]	1.86E-05
TERT	rs2736103	1.109 [1.055,1.165]	4.53E-05
TERT	rs27064	1.126 [1.062,1.194]	7.90E-05
TERT	rs7725218	1.134 [1.065,1.207]	8.47E-05
TERT	rs410805	1.125 [1.061,1.192]	9.24E-05
TERT	rs2736105	1.1 [1.048,1.155]	0.000119
TERT	rs27066	1.122 [1.059,1.19]	0.000124
TERT	rs7726159	1.135 [1.063,1.212]	0.000155
TERT	rs13174919	0.903 [0.857,0.953]	0.000192
TERT	rs13174814	0.904 [0.857,0.953]	0.00021
TERT	rs7734992	1.122 [1.056,1.193]	0.000242
TERT	rs4975612	0.908 [0.861,0.957]	0.000378
TERT	rs2735846	1.094 [1.041,1.149]	0.000424
TERT	rs428499	1.103 [1.042,1.168]	0.000806
TERT	rs6897196	1.106 [1.041,1.176]	0.00123
TERT	rs2735845	1.097 [1.035,1.163]	0.00191
TERT	rs246994	1.088 [1.031,1.148]	0.00214
TERT	rs10054203	1.097 [1.033,1.165]	0.00266
TERT	rs11278847	1.101 [1.033,1.173]	0.00299
TERT	rs2736108	1.104 [1.034,1.179]	0.0032
TERT	rs4975538	1.1 [1.032,1.172]	0.00334
TERT	rs246995	1.084 [1.027,1.144]	0.00339
TERT	rs2735940	1.125 [1.04,1.217]	0.00372
TERT	rs2853672	1.121 [1.037,1.213]	0.00462
TERT	rs538006730	0.927 [0.88,0.977]	0.00468
TERT	rs2853669	1.097 [1.027,1.172]	0.006
TERT	rs10462706	0.932 [0.884,0.982]	0.00822
TERT	rs2736100	1.101 [1.022,1.185]	0.0109
TERT	rs72713502	1.068 [1.015,1.123]	0.0116
TERT	rs13167280	1.106 [1.022,1.198]	0.013
TERT	rs27065	1.063 [1.011,1.119]	0.0179
TERT	rs62329683	1.063 [1.004,1.125]	0.0361
TERT	rs2075785	1.033 [0.973,1.095]	0.286
TERT	rs2736099	1.046 [0.954,1.147]	0.335
TERT	rs2736098	1.115 [1.044,1.191]	failed test
TERT	rs3215401	1.103 [1.032,1.18]	failed test
TERT	rs10548207	1.104 [1.032,1.18]	failed test
TERT	rs2736107	1.115 [1.043,1.191]	failed test
TERT	rs370610372	1.109 [1.056,1.164]	failed test

9. Cross-ancestry meta-analysis: did the authors consider meta-analyzing or replicating some their findings in TOPMed (Bick et al. 2020), as this would include at least some Hispanic/Latino individuals with Indigenous ancestry?

We have now included the multi-ancestry TOPMed cohort to assess the replicability of our cross-ancestry meta-analysis. Broadly, the TOPMed cohort consists of 41% Europeans, 31% Africans, 15% Hispanics or Latinos, 9% Asians, and 4% “other” ancestries (PMID: 36119389). CH GWAS summary statistics derived from the analysis of 65,404 participants (4,141 CH individuals and 61,263 controls) were retrieved from the database of Genotypes and Phenotypes (dbGaP; application ID: 135432-3; PMID: 33057201). Summary statistics were available for overall CH, i.e., individuals with at least one driver CH mutations, but not for gene-specific CH. Therefore, replication analysis was performed with TOPMed overall CH GWAS summary statistics.

Of the 16 loci identified as genome-wide significant in our cross-ancestry meta-analysis, 14 loci (leading SNPs) were captured by TOPMed and were included for analysis. Of the 14 leading SNPs, 7 (*PARP1*, *TRIM590-IFT80*, *TERT*, *CD164*, *ATM*, *SETBP1*, *RUNX1*) were associated with CH at $P < 0.05$, and 13 showed consistent directionality.

After inclusion of TOPMed into our cross-ancestry MCPS-UKB meta-analysis, all 14 associations remained genome-wide significant, of which 10 associations were strengthened as indicated by the improvement of P values (Figure below and Supplementary Table 20).

We have now reported these findings in the **Results (“Cross-ancestry meta-analysis of CH” section, paragraph 1)** of the revised manuscript as follows:

“...Summary statistics from the multi-ancestry TOPMed cohort consisting of 4,141 CH individuals and 61,263 controls were available for overall CH for replication analysis^{17,37}. Of the 14 leading variants identified in TOPMed, 7 (*PARP1*, *TRIM590-IFT80*, *TERT*, *CD164*, *ATM*, *SETBP1*, *RUNX1*) were significantly associated with CH at $P < 0.05$ and 13 demonstrated consistent directionality. Inclusion of TOPMed in our cross-ancestry meta-analysis strengthened the association with CH for 10 out of the 14 leading variants (**Supplementary Table 20**).”

Minor points:

10. Abstract: please include the number of CH cases, this is more informative for assessing the power of the study than the total sample size

We have now included the numbers of CH cases (and controls) in the **Abstract** of the revised manuscript as follow:

“...Here, we investigate this by studying CH in 136,401 (4,249 cases and 132,152 controls) admixed participants from the Mexico City Prospective Study (MCPS) and 416,118 (20,488 cases and 395,630 controls) European participants from the UK Biobank (UKB)...”

11. I couldn't find any variant-level heterogeneity statistics in the cross-ancestry meta-analysis. Please include this information

We have now included variant-level heterogeneity statistics as assessed with Cochran's Q test and implemented in the METAL software (PMID: 20616382). Of the five novel loci identified in our cross-ancestry meta-analysis (*MYCN*, *MEIS1*, *PARP11-CCND2*, *THRB*, *UBE2G1-SPNS3*; **Fig. 5**), none had $P_{\text{heterogeneity}} < 0.05$ when the inverse variance-weighted average (IVW) or *P*-value based meta-analysis method was implemented (**Supplementary Table 19**).

12. I suggest toning down the conclusions in the abstract “... demonstrates the profound impact of ancestry on CH development”

We have now toned down our findings on the association between ancestry and CH in the **Abstract** of the revised manuscript as below:

“... reveals ancestry as a risk factor for CH development...”.

Reviewer #3:

Remarks to the Author:

In this paper the investigators used whole exome sequencing (WES) to detect clonal hematopoiesis (CH) in a sample of people with admixed European and indigenous American ancestry from the Mexico City Prospective Study (MCPS). They compare CH in MCPS to CH detected previously in European ancestry individuals from the UK Biobank (UKB). Their most notable finding is that CH is very significantly more common in Europeans from the UKB than it is in MCPS participants. They carry out a GWAS in the MCPS and find ancestry-specific germline variants at *TCL1B*, close to but with separate effects from previously known variants near the neighbouring *TCL1A* gene. They detect several other novel GWAS variants, either through the MCPS alone or in a meta-analysis of MCPS and UKB.

Overall the paper is very nicely written, well illustrated and makes for an interesting read. The primary finding of a big difference in CH rates between European and indigenous American ancestries is, if true, an important contribution to our understanding of CH. Moreover, the finding of variants at *TCL1B* that demonstrate similar properties to those at *TCL1A* but nevertheless map to a distinct location raises interesting questions about which gene (or both) is implicated in the genesis of CH. Accordingly, I think the paper would appeal to a broad general audience.

However, there are areas of concern, some quite major. The reported difference in CH prevalence between MCPS and UKB is very dramatic (OR = 0.56, Pval = 1.60E-206). In my sorry experience, results of such magnitude are almost invariably traceable to a technical artefact. The onus is on the investigators to convince the reader of the biological veracity of their results.

We thank the reviewer for the very thorough review of our manuscript and for stating its appeal to a large audience. We also recognise the concerns raised and believe that the way we have addressed them in the revised manuscript, as detailed below, significantly improves the quality of our work. In particular, to further validate our finding that CH is less common in individuals of Admixed American versus European

ancestry we have clarified the methodological details and performed substantial additional analyses: firstly, to confirm the similarity of the MCPS and UKB sequencing and bioinformatics pipelines; secondly, to replicate our principal findings in an expanded 58-CH gene panel; thirdly, to demonstrate that the difference in CH frequency is not due to technical differences. Collectively, we hope that this addresses the Reviewer's concerns and provides confidence as to the veracity of our primary finding, one which has precedent in oncology, in for example prostate cancer, where incidence is 75% higher in individuals of African ancestry compared to those of European ancestry (PMID: 33398198). We thank the Reviewer again for their constructive feedback and look forward to their further consideration of our work for publication in *Nature Genetics*.

1. According to Vlasschaert et al. (Ref 39), small changes in sequence processing can have large effects on the accuracy of CH calling. The primary whole exome sequencing was done outside of the authors' group, at Regeneron. It appears that FASTQ files were transferred to the authors' group at AstraZeneca and analysed further there. The authors need to provide a detailed account of how the sequence data for both MCPS and UKB were produced and processed. Such an account should be self-contained within this manuscript so that the reader does not have to trawl through the references to assemble an account of exactly what was done for each cohort. Any differences in sequence production and processing pipelines between MCPS and UKB should be highlighted and an explanation given as to why this would not affect the conclusions. A more detailed examination of the resulting mutation spectra in the two cohorts might be helpful here. I note also in this context that the description of CH mutation detection given in Dhindsa et al. (Ref 21) is a bit too sketchy for the current purposes.

We thank the reviewer for raising this issue and believe that the clarification on the similarity of the sequencing and bioinformatics pipelines we have now added to the Methods section strengthens our results. Additionally, a section on the potential limitations of a cross-study analysis is now included in the discussion section.

Specifically, the sequencing libraries for both MCPS and UKB were generated and analysed with the same approach, and this is now made explicit in the **Methods**

("Whole-exome sequencing pre-processing and variant calling", paragraph 1 and 2) of the revised manuscript as follows:

"Whole-exome sequencing data for genomic DNA from MCPS and UKB was generated by Regeneron Genetics Centre as previously described^{23,46,47}. Both MCPS and UKB samples were prepared with the same production pipeline. Specifically, sequencing libraries were generated using IDT xGen v1 capture kit, and subsequently sequenced on the NovaSeq6000 platform in 75-bp paired-end mode. The average sequencing depth across UKB and MCPS samples was 57.8x and 57.2x, respectively. This metric was computed as the average alignment coverage over the consensus coding sequence (CCDS) located on the autosomes. Both cohorts also had similar mapping quality with 89.21% and 89.28% of reads with MAPQ (mapping quality score) >40%, respectively.

The FASTQ files were subsequently processed at AstraZeneca as previously described⁴⁸. Both MCPS and UKB samples were subjected to the same processing pipeline. Specifically, sequencing reads were demultiplexed using the 10-bp index barcodes with `bcl2fastq v2.19.0` to obtain the sequencing reads for each sample in FASTQ format. Next, sequencing read alignment to the GRCh38 genome reference and germline variant detection, for exome-wide association analysis and gene-level collapsing analysis, was performed using the Illumina DRAGEN Bio-IT Platform Germline Pipeline v3.0.7. Somatic variant calling was performed using GATK MuTect2^{20,21}. A panel of normals was created from 200 of the youngest UKB participants without a haematologic malignancy diagnosis to remove potential recurrent artifacts with GATK *FilterMutectCalls*. The `--orientation-bias-artifact-priors` option was also specified to remove read orientation artifacts based on priors generated with *LearnReadOrientationModel*."

The section on the potential limitations of a cross-cohort study, and why we do not believe they have affected the overall conclusions of our study, is included in the Discussion (paragraph 2) of the revised manuscript as follows:

"A potential limitation of comparing CH frequency between MCPS and UKB is the technical differences in the sequencing and bioinformatics pipelines^{44,45}. This was

mitigated as far as possible by whole-exome sequencing DNA libraries being prepared by the same exome capture kit (IDT xGen v1 capture kit), sequenced on the same platform (NovaSeq6000) with the same sequencing mode (75-bp paired-end mode) by the same sequencing provider (Regeneron Genetics Centre)^{23,46,47}, and subjected to the same germline (Illumina DRAGEN Bio-IT Platform Germline Pipeline v3.0.7) and somatic (MuTect2) variant calling pipeline^{20,21,48}. Collectively, this resulted in similar somatic variant allele frequencies and coverage profiles observed between MCPS and UKB samples. While technical variation between MCPS and UKB affecting CH detection cannot be entirely ruled out, we replicated the difference in CH frequency within the MCPS study itself, where we observed that increased CH risk was associated with a higher fraction of European compared to Indigenous American ancestry.”

Lastly, we agree on the lack of an in-depth description of our methods for CH mutation detection. We have now provided more complete description on CH mutation detection in the **Methods (“CH detection”)** of the revised manuscript as below:

“To identify CH driver variants, we first retrieved Mutect2 PASS somatic variants occurring in a previously defined list of 74 genes with leukemogenic driver mutations¹⁷. They were annotated with the transcript ID, exon number, cDNA change, amino acid change, and protein consequence with Ensembl Variant Effect Predictor software^{56,57}, and filtered according to the previously defined criteria for putative CH drivers based on variant consequence¹⁷ (**Supplementary Table 4**). Only variants supported by at least 3 alternate allele reads and with a variant allele frequency (VAF) of at least 3% but not more than 40% were retained. All subsequent analysis was initially restricted to 15 pre-leukemic driver genes which demonstrated association between the presence of a putative driver and age, namely *DNMT3A*, *TET2*, *ASXL1*, *PPM1D*, *TP53*, *SF3B1*, *SRSF2*, *GNB1*, *IDH2*, *JAK2*, *PRPF8*, *KRAS*, *NRAS*, *BRAF*, and *MPL* (**Extended Data Fig. 4a and b, and Supplementary Table 7**)¹⁵. Based on this CH variant classification, the gene-panel consisted of genes with any protein-truncating variants (frameshift, nonsense, and splice-site) along the gene body and recurrent hotspot variants (*DNMT3A*, *TET2*,

and *TP53*), genes with protein-truncating variants at specific exons (*ASXL1* and *PPM1D*), and genes with only recurrent hotspot variants (*SF3B1*, *SRSF2*, *GNB1*, *IDH2*, *JAK2*, *PRPF8*, *KRAS*, *NRAS*, *BRAF*, and *MPL*). Additionally, using identical variant filtering criteria, we extended CH detection to the 58-gene panel (Vlasschaert *et al.*)²², and reported and compared downstream association results for both the 15- and 58-gene panels.”

2. The authors are clearly aware of this issue and they adopt the rather elegant solution of partitioning the MCPS subjects into those with predominantly European and predominantly indigenous American ancestry. They find that the difference in CH prevalence is maintained, albeit with reduced effect due to the admixture. On the face of it, this is pretty compelling evidence, but:
 - a. Their definition of CH requires the detection of one or more mutations that are on shortlist of 15 candidate driver genes with specified mutations. This shortlist was compiled from a much longer list of 74 candidate driver genes/mutations. The criteria for selecting the shortlist are not adequately described. The genes and mutations that constitute the shortlist are not identified in the manuscript. It appears that the shortlist was constructed using the predominantly European UKB sample. This could introduce a search bias in favour of CH mutations most commonly found in Europeans and in particular in UKB participants. Such a bias would be carried over into the ancestry partition of the MCPS. The result would be a higher estimated CH prevalence in the European ancestry partition of the MCPS and in the entire UKB sample.

The 74-gene panel used in the TOPMed study (PMID: 33057201) was a curated list that also contained leukaemia-defining mutations such as *NPM1* and *FLT3*. However, CH is by definition a preleukaemic condition, of HSC expansion in the absence of overt blood cancers such as acute myeloid leukaemia (AML). The same authors refined their list of CH genes to a panel of 58 genes that met criteria for (1) increasing somatic mutation frequency with increasing age and (2) myeloid preleukaemic and leukaemic clonal selection (PMID: 36652671).

The genes that constitute the 15-gene panel in our study were those from the proposed 74 CH genes in the TOPMed study (PMID: 33057201), which we confirmed to be associated with increasing age (Extended Data Fig. 4a and b, and Supplementary Table 7). This has been clarified in the Methods section “CH detection”. Additionally, we have now also included analysis using the much larger 58-gene panel (Vlasschaert *et al.*; PMID: 36652671) to ensure the robustness of our conclusions.

Consistent with our finding for the 15-gene panel, the 58-gene panel showed CH was less common in MCPS compared to UKB after adjusting for age, sex, and smoking status (Fig. 1c and Supplementary Table 8):

Inter-population comparison:

CH (outcome)	OR [95% CI]	P value
15-gene panel	0.59 [0.57,0.61]	7.31×10^{-185}
58-gene panel (Vlasschaert et al.)	0.62 [0.60,0.64]	5.11×10^{-179}

Similarly, in our intra-population analysis of MCPS individuals, we observed a higher frequency of CH among individuals with a higher proportion of European genome compared to individuals with a higher proportion of non-European (American/African) genome after adjusting for age, sex, and smoking status, when overall CH was defined with the 15- or 58-gene panel (15-gene panel: Fig. 2d; 15- and 58-gene panel: Supplementary Table 9):

Intra-population comparison:

CH (outcome)	Beta [95% CI]	P value
15-gene panel	0.84 [0.66,1.03]	7.35×10^{-19}
58-gene panel (Vlasschaert et al.)	0.71 [0.54,0.88]	1.41×10^{-15}

One potential problem with the addition of another 43 genes that are relatively rare drivers of CH, is that the definition of individual variants as driver mutations, passenger mutations or errors, can be difficult to ascertain, due to the lack of recurrence. This can “contaminate” the list of CH carriers with individuals that

do not have *bona fide* CH. Indeed, genetic association analyses in UKB demonstrated an attenuation of associations of common germline variants with overall CH for the 58-gene panel relative to the 15-gene panel. Specifically, GWAS for variants limited to the 15-gene panel identified 14 genome-wide significant loci ($P < 5 \times 10^{-8}$), whereas GWAS for 58-gene panel identified 12 genome-wide significant loci. More importantly, of the 11 overlapping significant loci between the two panels, the effect sizes and P values were attenuated in 58-gene panel relative to the 15-gene panel. This may be indicating that the 58-gene panel, while identifying more individuals with *bona fide* CH may also contain a non-negligible proportion of misidentified individuals.

UKB GWAS of CH (as defined with 15-gene panel)

UKB GWAS of CH (as defined with 58-gene panel)

Locus	15-Gene panel			58-Gene panel (Vlasschaert et al.)		
	Top SNP (rs ID)	OR [95% CI]	P value	Top SNP (rs ID)	OR [95% CI]	P value
PARP1	rs138994074	0.916 [0.89,0.944]	5.81E-09	rs1136410	0.926 [0.902,0.951]	1.87E-08
MYCN	rs12471506	1.06 [1.04,1.09]	3.66E-08	Did not reach genome-wide significance ($P \geq 5E-08$)		
LY75,CD302	rs1549387	1.06 [1.04,1.09]	3.19E-09	rs1549387	1.06 [1.04,1.08]	2.51E-08
TRIM59,IFT80	rs56658671	1.13 [1.1,1.15]	1.77E-30	rs56658671	1.11 [1.09,1.13]	4.63E-28
CXXC4,TET2	rs116597408	1.19 [1.13,1.25]	3.04E-10	rs116597408	1.17 [1.11,1.23]	9.21E-10
TERT	rs7705526	1.26 [1.24,1.29]	4.15E-103	rs7705526	1.24 [1.22,1.27]	4.30E-102
HCG26,MICB-DT,HLA-C	rs3131643	0.918 [0.892,0.944]	2.48E-09	Did not reach genome-wide significance ($P \geq 5E-08$)		
CD164	rs12193493	0.904 [0.885,0.924]	1.19E-19	rs2275652	0.916 [0.897,0.936]	2.44E-16
STN1	rs34763036	0.91 [0.88,0.94]	2.06E-08	Did not reach genome-wide significance ($P \geq 5E-08$)		
NPAT,ATM	rs228606	1.1 [1.08,1.13]	8.36E-21	rs7129527	1.09 [1.07,1.11]	1.52E-17
ITPR2	rs149752564	1.14 [1.1,1.18]	2.26E-11	rs149752564	1.13 [1.09,1.17]	1.77E-11
MSI2	rs118121072	0.867 [0.825,0.911]	8.83E-09	rs188761458	0.822 [0.767,0.882]	2.02E-08
SETBP1	rs1849209	0.895 [0.874,0.916]	2.13E-20	rs8088824	0.909 [0.889,0.929]	2.08E-17
RUNX1	rs2834707	1.06 [1.04,1.09]	4.29E-09	rs2834706	1.06 [1.04,1.08]	2.90E-08
MIR4708,FUT8	Did not reach genome-wide significance ($P \geq 5E-08$)			rs4899178	1.06 [1.04,1.08]	2.11E-08

Comparing MCPS CH GWAS based on the 15- and 58-gene panel demonstrated comparable effect sizes and *P* values for the two genome-wide significant loci (*TERT* and *TCL1A-TCL1B*). Therefore, the association results were not significantly affected by extending the analysis from the 15-gene to the 58-gene panel.

Locus	15-Gene panel			58-Gene panel (Vlasschaert et al.)		
	Top SNP (rs ID)	OR [95% CI]	P value	Top SNP (rs ID)	OR [95% CI]	P value
TERT	rs2853677	0.766 [0.728,0.805]	1.62E-24	rs2853677	0.777 [0.742,0.814]	3.35E-25
TCL1B	rs968294563	1.78 [1.49,2.13]	2.88E-09	rs968294563	1.68 [1.42,1.99]	1.67E-08

Our findings suggest that neither the 15-gene nor the 58-gene panel calls represents an entirely accurate and comprehensive set of CH gene variants. Therefore, we have included our analysis using variants from both the 15- and the 58-gene panels reporting findings from both panels in the revised manuscript. Reassuringly, the result that CH is less frequent in Admixed Americans vs. Europeans is consistent across the two gene panels.

- b. MuTect2 uses a panel of normals to estimate per-site beta distribution parameters for use in refining somatic likelihood assignment (Kessler et al., Ref 16). In the present MCPS study, the investigators used a panel of normals comprised of 200 young UKB participants (i.e. Europeans) and applied it to both UKB and MCPS samples. This could create a somatic mutation calling bias in favour of Europeans and such a bias would again be carried over to the

ancestry partition of the MCPS. Other investigators (e.g. Kessler et al.) generate separate panels of normals for each cohort.

We thank the reviewer for raising an issue that could potentially have created a discrepancy between the detection of CH in UKB and MCPS. We made the decision to use the same PoN (created from UKB) in both UKB and MCPS as it is designed to filter artifacts, and both cohorts underwent the same sequencing protocol and were analysed with the same bioinformatics pipeline. Mutect2 performs the filtering by flagging any variant which was detected in at least two samples used in the PoN, but does not use the beta distribution (<https://gatk.broadinstitute.org/hc/en-us/community/posts/360077617092-CreateSomaticPanelOfNormals-BETA-tag>). Therefore, we anticipate minimal bias in CH detection when the UKB-derived PoN was used for CH detection among MCPS samples.

Nevertheless, it is conceivable that using the UKB PoN for the detection of CH variants in MCPS samples may have led to a bias in detection. To investigate, we inspected variants filtered by the PoN in MCPS which would have otherwise been considered CH driver variants in our study. In the 15-gene panel, we observed only 3 CH qualifying variants filtered by Mutect2's PoN which would have otherwise been PASS. One of these variants was also filtered by the PoN in a UKB participant, while 2 were specific to MCPS as follows:

Chr	Position	Ref. allele	Alt. allele	Gene	Consequence	No. of individuals
2	25246633	CAGGCCCTTAGGGCCAG AAGGCTG	C	DNMT3A	Frameshift deletion	1
4	105237059	GTTAT	G	TET2	Frameshift deletion	1

Inclusion of the 2 individuals with PoN-flagged variants as CH samples yielded CH frequency of 3.117% (4,251 / 136,401), which represents a very small increase from the original CH frequency of 3.115% (4,249 / 136,401) reported in our manuscript. Therefore, PoN filtering does not account for the lower frequency of CH observed in MCPS compared to UKB.

When we extended our analysis to the 58-gene panel (Vlasschaert *et al.*; PMID: 36652671), we observed 10 variants flagged by PoN, of which 7 were also filtered in UKB participants, while 3 were specific to MCPS as follows:

Chr	Position	Ref. allele	Alt. allele	Gene	Consequence	No. of individuals
2	25246633	CAGGCCCTTAGGGCCAG AAGGCTG	C	DNMT3A	Frameshift deletion	1
4	105237059	GTTAT	G	TET2	Frameshift deletion	1
16	3729209	T	TG	CREBBP	Frameshift insertion	1

Inclusion of the 3 individuals with PoN-flagged variants as CH samples yielded CH frequency of 3.621% (4,939 / 136,401), which represents only a marginal increase from the original CH frequency of 3.619% (4,936 / 136,401) reported in our manuscript.

It is also noteworthy that the rate of PoN-flagged variants in UKB was similar to MCPS. In the 15-gene panel, we observed only 5 CH qualifying variants filtered by Mutect2's PoN which would have otherwise been PASS. One of these variants was also filtered by the PoN in a MCPS participant, while 4 were specific to UKB as follows:

Chr	Position	Ref. allele	Alt. allele	Gene	Consequence	No. of individuals
4	105269705	T	A	TET2	Missense	1
20	32434638	AGG	A	ASXL1	Frameshift deletion	5
20	32434638	A	AT	ASXL1	Frameshift insertion	2
20	32434638	A	AGAGG	ASXL1	Frameshift insertion	1

Of the 9 individuals with PoN-flagged variants above, 2 individuals already had qualifying CH variants identified. Specifically, 1 individual (with 4-105269705-T-A variant) already had qualifying variants in *DNMT3A* and *TET2* identified, and another 1 individual (with 20-32434638-AGG-A variant) already had a qualifying variant in *TET2* identified. Therefore, 7 additional CH individuals would be identified with these PoN-flagged variants. Nevertheless, inclusion of these 7

individuals with PoN-flagged variants as CH samples yielded CH frequency of 4.925% (20,495 / 416,118), which represents only a marginal increase from the original CH frequency of 4.924% (20,488 / 416,118) reported in our manuscript.

When we extended our analysis to the 58-gene panel proposed by (Vlasschaert *et al.*; PMID: 36652671) we observed 20 variants flagged by PoN, of which 7 were also filtered in MCPS participants, while 13 were specific to UKB as follows:

Chr	Position	Ref. allele	Alt. allele	Gene	Consequence	No. of individuals
4	105269705	T	A	TET2	Missense	1
20	32434638	AGG	A	ASXL1	Frameshift deletion	5
20	32434638	A	AT	ASXL1	Frameshift insertion	2
20	32434638	A	AGAGG	ASXL1	Frameshift insertion	1
13	32758086	G	T	PDS5B	Splice site	1
12	11884510	C	T	ETV6	Nonsense	1
7	139409618	A	AAGAG	LUC7L2	Frameshift insertion	1
X	124061770	G	T	STAG2	Splice site	1
X	124061770	G	A	STAG2	Splice site	3
X	124066174	G	T	STAG2	Splice site	31
1	150961067	C	A	SETDB1	Nonsense	3
1	150961067	C	G	SETDB1	Nonsense	1
1	150961084	G	T	SETDB1	Nonsense	2

Of the 53 individuals with PoN-flagged variants above, 6 individuals already had qualifying CH variants identified. Specifically, 1 individual (with 7-139409618-A-AAGAG variant) already had qualifying variants in *DNMT3A* identified, another 3 individuals (with 20-32434638-AGG-A, X-124066174-G-T, or X-124066174-G-T variant) already had a qualifying variant in *TET2* identified, another 1 individual (with 4-105269705-T-A variant) already had qualifying variants in *DNMT3A* and *TET2* identified, and another 1 individual (with X-124066174-G-T variant) already had qualifying variants in *BCORL1* identified. Therefore, 47 additional CH individuals would be identified with these PoN-flagged variants. Nevertheless, inclusion of these 47 individuals with PoN-flagged variants as CH samples yielded CH frequency of 5.673% (23,605 / 416,118), which represents only a marginal increase from the original CH frequency of 5.661% (23,558 / 416,118) reported in our manuscript.

- c. Sequence read alignment used GRCh38 as a reference, which is essentially a European genome. MuTect2 uses mapping quality measures to generate confidence in somatic mutation identification. This could create a mapping quality bias in favour of Europeans which again would be carried over to somatic mutation calls in the ancestry partition of the MCPS.

Indeed, current human reference genome is predominantly European-derived. We would expect a mapping quality bias to be reflected in the downstream coverage distribution and consequently variant calls. To assess for differences in coverage distribution between MCPS and UKB samples, we plotted the coverage profile along the coding region of each CH gene. Reassuringly, we observed no prominent differences in the coverage profile between MCPS and UKB samples (Extended Data Fig. 2a-o).

Furthermore, at the exome-level, we showed both MCPS and UKB had similar coverage and mapping quality in the Methods (“Whole-exome sequencing pre-processing and variant calling”, paragraph 1) as follows:

“...The average sequencing depth across UKB and MCPS samples was 57.8x and 57.2x, respectively. This metric was computed as the average alignment coverage over the consensus coding sequence (CCDS) located on the autosomes. Both cohorts also had similar mapping quality with 89.21% and 89.28% of reads with MAPQ (mapping quality score) >40%, respectively.”

3. Smoking is an important determinant of CH risk (Stacey et al., Ref 8). How smoking was taken into account in the analyses is not sufficiently explained in the manuscript. Moreover, smoking shows a strong association with proportion of European ancestry amongst Hispanic/Latino population samples, with a higher prevalence of smoking behaviour correlated with greater European ancestry (Choquet et al. 2021, PMID: 33633108). If not properly managed, this could cause an increase in apparent CH prevalence in Europeans due to residual confounding.

We thank the reviewer for pointing us to this study. We have clarified in our manuscript how smoking was taken into account, and performed additional analysis to ensure it has not confounded our conclusion that CH frequency is associated with ancestry in Results (“Association between ancestry and CH frequency” section, paragraph 2 and 3):

“Smoking frequency has been reported to be higher in individuals of European descent compared to those of American descent among self-reported Hispanics/Latinos²⁵. In MCPS, a logistic regression model including age and sex as co-variables, showed that the proportion of European ancestry was indeed associated with smoking frequency (ever-smoker: beta [95% CI] = 1.64 [1.56, 1.71], $P < 2.2 \times 10^{-16}$; previous smoker: 1.05 [0.96, 1.14], $P < 2.2 \times 10^{-16}$; current smoker: 2.01 [1.91, 2.10], $P < 2.2 \times 10^{-16}$). As an example, the frequency of ever-smokers among individuals with $\geq 70\%$ European genome was 66.6%, but only 39.7% among individuals with $< 10\%$ European genome (Fig. 2c). Smoking, in turn, was associated with increased risk of CH as previously reported¹⁵⁻¹⁷ (ever-smoker: OR [95% CI] = 1.13 [1.05, 1.21], $P = 8.21 \times 10^{-4}$; previous smoker: 1.10 [1.02, 1.20], $P = 0.017$; current smoker: 1.17 [1.07, 1.28], $P = 6.13 \times 10^{-4}$). Stratified by smoking status, the trend of increasing CH frequency with increasing European genome percentage was observed among individuals across all smoking strata (Extended Data Fig. 6f-i).

Finally, in a logistic regression model with CH as the outcome and proportion of European genome (in bins of 10%) as the main predictor adjusted for age, sex, and smoking status, we observed increasing CH risk with increasing proportion of European genome (Extended Data Fig. 6j and k). Notably, the group of individuals with the highest percentage of European genome ($\geq 70\%$) had the highest risk of CH relative to the group of individuals with the lowest percentage of European genome ($< 10\%$, OR = 1.87 [1.46, 2.39], $P = 4.24 \times 10^{-7}$). Consistent with this, we observed an increased risk of overall CH in an intra-population logistic regression with CH as the outcome and proportion of European genome (continuous variable) as the main predictor adjusted for age, sex, and smoking (beta = 0.84 [0.66, 1.03], $P = 7.35 \times 10^{-19}$), and also with increased risk of DNMT3A-CH (1.15 [0.89, 1.41], P

= 4.52×10^{-18}), *ASXL1*-CH (1.55 [0.97, 2.14], $P = 1.76 \times 10^{-7}$), and *SRSF2*-CH (2.17 [0.88,3.45], $P = 9.64 \times 10^{-4}$; Fig. 2d and Supplementary Table 9).”

4. It is not clear precisely how the odds ratios comparing the CH risk in the UKB compared with the MCPS were arrived at. This is of particular concern because of the very different age structures between the two cohorts. The stated odds ratios do not seem to tally with the differences in prevalence as shown in Fig 1b. The authors need to show precisely what model they used and provide particular detail on how age and smoking covariates were handled. The same applies to the odds ratios derived from the ancestry partition of the MCPS.

To account for the differences between the cohorts we performed two sets of analyses: firstly, where age, sex and smoking were included as covariates; and secondly in an age and sex matched analysis. This has is now clarified in the Results section of the text, while Figure 1b shows the raw CH frequency in UKB and MCPS cohorts (4.92% and 3.12% respectively). Likewise, in the ancestry partitioned analysis of MCPS, age, sex and smoking were included as covariates, and a smoking status stratified analysis was performed.

Specifically, in an inter-population logistic regression with CH as the outcome and population (UKB versus MCPS) as the main predictor adjusted for age, sex and smoking as covariates, the OR of CH in UKB relative to MCPS was 1.69. Conversely, the inverse OR of CH of 0.59, i.e. MCPS relative to UKB, is reported in the Abstract. We have now clarified that age, sex, and smoking were included as covariates that were adjusted for in the inter-population logistic regression model in the “Results: Frequency of CH” section (paragraph 3) of the revised manuscript as below:

“Consistent with the increased frequency of CH in UKB, we observed an increased risk of overall CH in UKB relative to MCPS participants in an inter-population logistic regression with CH as the outcome and population as the main predictor adjusted for age, sex, and smoking (OR [95% C] = 1.69 [1.63,1.75], $P = 7.31 \times 10^{-185}$; Fig. 1c).”

Similarly, we have now clarified that age, sex, and smoking were adjusted for in the intra-population logistic regression model for the resulting OR derived from the ancestry partition of the MCPS in the Results (“Association between ancestry and CH frequency”, paragraph 3) of the revised manuscript as follow:

“Finally, in a logistic regression model with CH as the outcome and proportion of European genome (in bins of 10%) as the main predictor adjusted for age, sex, and smoking status, we observed increasing CH risk with increasing proportion of European genome (Extended Data Fig. 6j and k). Notably, the group of individuals with the highest percentage of European genome ($\geq 70\%$) had the highest risk of CH relative to the group of individuals with the lowest percentage of European genome ($< 10\%$, OR = 1.87 [1.46, 2.39], $P = 4.24 \times 10^{-7}$). Consistent with this, we observed an increased risk of overall CH in an intra-population logistic regression with CH as the outcome and proportion of European genome (continuous variable) as the main predictor adjusted for age, sex, and smoking (beta = 0.84 [0.66, 1.03], $P = 7.35 \times 10^{-19}$), and also with increased risk of *DNMT3A*-CH (1.15 [0.89, 1.41], $P = 4.52 \times 10^{-18}$), *ASXL1*-CH (1.55 [0.97, 2.14], $P = 1.76 \times 10^{-7}$), and *SRSF2*-CH (2.17 [0.88, 3.45], $P = 9.64 \times 10^{-4}$; Fig. 2d and Supplementary Table 9).”

5. A component with Mexican/Central American ancestry can be identified within the UKB cohort. Do the authors’ findings replicate in an ancestry partition of the UKB sample?

In the UKB, we identified 704 genetically-defined Admixed Americans (*peddy*-inferred probability $\geq 95\%$) passing QC, without diagnosis of haematological neoplasm prior to blood sample collection, and consent not withdrawn as of April 2024. CH frequency in UKB was lower in Admixed Americans compared to Europeans (OR = 0.83 [95% CI = 0.53, 1.27]) after adjusting for age, sex, and smoking status, albeit not statistically significant ($P = 0.39$). This is likely due to the relatively small numbers of Admixed Americans in UKB. This highlights the value of studying large-scale non-European cohorts, such as MCPS, for epidemiological studies.

6. The Methods describes only the exclusion of samples for technical reasons. Were individuals with prior neoplasia excluded from either of the cohorts? What ICD10 codes were used for exclusion? What other exclusion criteria were employed? Were they the same for both cohorts?

We have now excluded samples with diagnosis of haematological neoplasms prior to blood sample collection in UKB as previously described (PMID: 35835912; PMID: 36450978). The list of codes used to identify these individuals are now provided in Supplementary Table 3. However, records of samples with prior diagnosis of haematological neoplasms were not captured in MCPS. We do not expect this difference to have resulted in our observation of lower frequency of CH in MCPS compared to UKB. Specifically, in the UKB, higher frequency of CH was observed among individuals with prior diagnosis of haematological cancers, i.e. 374 of 2,971 (12.59%). Conceivably, the frequency of CH would increase if we were to include these individuals, i.e., 20,862 of 419,089 (4.98%). Conversely, frequency of CH was lower when these individuals were excluded (as reported in our revised manuscript), i.e., 20,488 of 416,118 (4.92%). Therefore, without excluding individuals with prior haematological neoplasm from MCPS, our current estimates of CH among MCPS participants are at the upper bound.

7. Page 8: It is unclear why the investigators choose to compare the LD of rs187319135 with rs10131341 rather than the more reproducibly associated *TCL1A* variant rs2887399. Are the LD relationships similar? It is not clear in the text whether the D-prime and r-squared values are derived from the MCPS sample.

The *TCL1B* upstream variant (rs187319135) was identified as genome-wide significant in MCPS *TET2*-CH GWAS while in a recent CH epidemiological study (PMID: 35835912), the *TCL1A* upstream variant (rs10131341) was identified as the most strongly associated variant at the *TCL1A-TCL1B* locus in the *TET2*-CH GWAS. We therefore compared the LD of rs187319135 with rs10131341 (Extended Data Fig. 10a). As the *TCL1A* promoter variant (rs2887399) was identified as the lead variant at the *TCL1A-TCL1B* locus in *DNMT3A*- (PMID: 33057201; PMID: 36450978) and *TET2*-CH (PMID: 33057201) GWAS in other CH

epidemiological studies, and also in a recent GWAS of CH expansion rate (PMID: 37046083), we have extended our analysis to include this variant.

In our new analysis for rs2887399, we similarly observed the minor allele of rs187319135 to be in high LD with the major of rs2887399 ($D' = 0.97$, $r^2 = 0.0016$; Extended Data Fig. 11a), but that the association of rs187319135 with CH was independent of rs2887399 (Extended Data Fig. 11b-c). We have added this to the text and now clarified that the LD association analyses was performed in MCPS in the Results (“Genome-wide common variant associations with CH” section, paragraph 3) of the revised manuscript as follows:

“... In MCPS, the minor allele of rs187319135 was observed to be frequently co-inherited alongside the major alleles of rs10131341 ($D' = 0.95$) and rs2887399 ($D' = 0.97$, Extended Data Fig. 10a and 11a), however the overall allele correlation was low ($r^2 = 0.0024$ and $r^2 = 0.0016$, respectively) due to rs187319135 being rarer relative to rs10131341 and rs2887399. Correspondingly, rs187319135 remained associated with overall CH, and *TET2*- and *SF3B1+SRSF2*-CH at genome-wide significance threshold even after conditioning on rs10131341 and rs2887399 in MCPS (Extended Data Fig. 10b-c and 11b-c)...”

8. It is fine for the authors to document their observations for the *CSGALNACT1* and *DIAPH3* loci. However, these associations arise from what is essentially a subgroup analysis (*ASXL1*-CH and *TP53*-CH subgroups). The levels of significance would not withstand a full Bonferroni adjustment. The associations do not appear to replicate well in the UKB, even though the variants are common enough there. The authors might want to add a “health warning” concerning the robustness of the observations for *CSGALNACT1* and *DIAPH3*. The statement that the *DIAPH3* variant “conferred an increased risk exclusively of *TP53*-CH” isn’t really true, as Extended Data Figure 9 shows there is some risk also of *PPM1D*-CH in MCPS.

In our revised manuscript, with the final 15-CH gene panel variant call set (PMID: 33057201; see Appendix of this document), the *DIAPH3* locus initially identified in *TP53*-CH is no longer genome-wide significant (before revision: $P = 3.58 \times 10^{-8}$;

after revision: $P = 7.20 \times 10^{-8}$), but the *CSGALNACT1* locus initially identified in *ASXL1*-CH remained genome-wide significant (before revision: $P = 3.63 \times 10^{-8}$; after revision: $P = 2.37 \times 10^{-8}$; Table 1 and Extended Data Fig. 9c). In the Results (“Genome-wide common variant associations with CH”, paragraph 4), we have now cautioned the readers that the *CSGALNACT1* locus was marginal genome-wide significant and was identified in the subgroup analysis, and therefore may not withstand full Bonferroni adjustment as follows:

“...It is noteworthy that the *CSGALNACT1* locus was marginally genome-wide significant and was identified in a subgroup (gene-specific CH) analysis, and therefore may not withstand full Bonferroni adjustment.”

9. The Discussion consists primarily of a restatement of the study findings. The authors claim that they have made “novel insights into CH pathogenesis” and revealed “fundamental biological insights”. What are they? How, for example, do the authors think that the difference in prevalence in CH between ancestries might come about? Are there differences in mutation rate, or clonal selection pressure, or what? Do the authors think that some germline variants promote CH through their effects on *TCL1B*, or *TCL1A*, or both? While I understand that the authors may not wish to be drawn into unbridled speculation, without some level of interpretation their findings seem to be reduced to the level of mere phenomenology.

Encouraged by the reviewer to provide further interpretation of our findings and to make reasonable speculations, we have now extensively rewritten the Discussion section of the article. The Discussion now includes paragraph 3:

“Differences in the prevalence of diseases between populations are the result of variation in both environmental exposures and genetics⁴⁹. While known risk factors (age, sex and smoking) were accounted for in our analysis, additional environmental factors may play a role. For example, metformin has recently been shown to reduce the clonal fitness of *DNMT3A*-mutant HSPCs⁵⁰. It is noteworthy that for a subset of driver genes (*TET2*, *PPM1D*, *TP53*, *JAK2*, and *MPL*), while our

inter-population analysis showed increased UKB frequency, our intra-population analysis among MCPS participants did not demonstrate higher frequencies among individuals with a higher proportion of European genome. This may suggest that environmental risk factors, which would be expected to vary more between MCPS and UKB, compared to within MCPS, play a larger role in determining the frequency of clones driven by mutations in these genes^{51,52}. As novel CH risk factors are revealed over the coming years, it will be important to assess if they explain population differences in CH frequency..”

And we have now discussed whether *TCL1B*, in addition to *TCL1A*, is involved in promoting CH in (paragraph 4):

“...Both *TCL1B* and *TCL1A* are observed to be aberrantly expressed in T cell leukaemia driven by t(14;14)(q11;q32,1), a translocation event that juxtaposes *TCL1B/A* to the α/δ T cell receptor locus⁵³. Therefore, while it’s tempting to implicate *TCL1B* as a potential driver of clonal expansion, and by extension, CH development, it is worth noting that *TCL1B* is epigenetically silenced and therefore transcriptionally repressed in HSPCs³³. Nevertheless, it is plausible that the *TCL1B* risk variants may lead to ectopic expression of the gene. One method of assessing the potential functional impact of GWAS/ExWAS variants is through their linkage to expression quantitative trait loci (eQTL). However, eQTL analysis of the *TCL1B* promoter risk variant in MCPS was hampered by the lack of such non-European-specific variants in publicly available European-majority eQTL datasets. While there are emerging non-European eQTL resources, these datasets are relatively small⁵⁴. This highlights the urgency of establishing large-scale non-European population-based resources, including eQTL databases, to allow equitable research in diverse ancestries/communities.”

Minor Points:

10. Figure 1b and others: the 95% confidence interval shadings are not visible. This may be simply due to how the graphic objects are rendered in MS-Word. Nevertheless, the reviewers need to be able to see them.

We apologise for the missing interval shadings. We have now realised the missing confidence interval shadings were typically observed on specific devices, primarily Windows. This was due to the figure being saved as PDF, rather than PNG. We have amended this, and the confidence interval shadings should now be visible on all devices.

11. Extended Data Figure 5b: It is not very clear how the OR on the Y axis were determined here. A better explanation or an alternative presentation is required.

We apologise for mis-labelled y-axis values. While OR was indicated on the y-axis, the values represented the beta. We have now rectified this mistake, and the values on the y-axis now represent OR as originally intended. The updated figure is now in Extended Data Fig. 6j and k.

12. Extended Data Figures 8d and 13a: Both red and blue bars are defined as “before conditioning” in the legends.

We apologise for the mis-labelled legends. This is now amended in Extended Data Fig. 10d and 16a of the revised manuscript.

13. Pg 10, last sentence: “Notably, CHEK2 c.1100del constituted...”. Don’t you mean c.1100delC here?

We thank the reviewer for spotting this. Indeed, the authors meant c.1100delC. This is now fixed in the Results (“Cross-ancestry meta-analysis of CH” section, paragraph 3) of the revised manuscript as follows:

“...Notably, *CHEK2* c.1100delC constituted 72% of all *CHEK2* qualifying variants in UKB individuals...”

14. Pg 12, first sentence: “promoter risk variant (rs774615666) were less common in Indigenous American...” Don’t you mean “more common” here?

We thank the reviewer for spotting this. Indeed, the authors meant that the *TCL1B* upstream (rs187319135) and *TCL1B* promoter risk variant (rs774615666) were more common in Indigenous American compared to European ancestry. In the interest of space and conciseness, and to avoid re-iterating statements already made in the Results (“Exome-wide rare variant associations with CH” section, paragraph 1) as follows, we have now removed this paragraph from the Discussion.

“...Within the MCPS cohort, rs774615666 was unique to Indigenous American (Mexican) ancestry (MCPS Variant Browser <https://rgc-mcps.regeneron.com/home/> (2023): MAF = 0.50%, versus MAF = 0% for European ancestry)...”

15. Pg 19, Data availability: “Summary statistics for GWAS, ExWAS, gene burden association analysis, ... are provided in the Supplementary Tables” The full summary data do not appear to be in the Supplementary Tables. Are the authors planning to release them?

The full summary statistics for GWAS, ExWAS and gene-collapsing association analysis have now been deposited onto the NHGRI-EBI GWAS Catalog (PMID: 36350656) and Zenodo (but remain under embargo until acceptance/publication of manuscript). The GWAS Catalog accession numbers and Zenodo links indicated in Supplementary Table 23. This is now reflected in Data availability section of the revised manuscript as follows:

“Full summary statistics for GWAS are available on NHGRI-EBI GWAS Catalog⁷¹ while the full summary statistics for ExWAS and gene-collapsing analysis are available on Zenodo. The GWAS Catalog accession numbers and Zendo links are indicated in Supplementary Table 23...”

Appendix

In our revised manuscript, we have ensured that variants for our 15-CH gene panel (and also 58-CH gene panel) call set rigorously match the criteria set out by TOPMed (PMID: 25426837; PMID: 33057201; Supplementary Table 4, and further detailed in the Methods (“CH detection” section)). In particular, the variant call set were harmonised with the specific isoform specified for each CH gene by TOPMed. Reassuringly, the differences in variant call set for our 15-CH gene panel from our original versus revised manuscript are minor.

For MCPS, the variant call set for 8 CH genes is identical in the original versus revised manuscript, namely *BRAF*, *GNB1*, *IDH2*, *KRAS*, *NRAS*, *PPM1D*, *PRPF8*, and *SF3B1*. Minor differences were observed for the remaining CH genes as follows. A notable exception is *TP53* where we identified 23 additional individuals with qualifying variants mapping to isoform NM_001126112.

DNMT3A

TET2

ASXL1

Variant type	N
Splice site (acceptor)	1

TP53

Variant type	N
Stop gained	10
Frameshift deletion	8
Frameshift insertion	2
Splice site (acceptor)	2
Splice site (donor)	1

JAK2

Variant type	N
In-frame deletion (p.R541_E543delinsK)	1

SRSF2

Variant type	N
In-frame insertion (p.R94dup)	2
In-frame deletion (p.P95_S101del)	1

MPL

Variant type	N
Missense (p.Q516K)	1

For UKB, the variant call set for 8 CH genes is identical in the original versus revised manuscript, namely *BRAF*, *GNB1*, *IDH2*, *KRAS*, *MPL*, *NRAS*, *PRPF8*, and *SF3B1*. Minor differences were observed for the remaining CH genes as follows. A notable exception is *TP53* where we identified 70 additional individuals with qualifying variants mapping to isoform NM_001126112.

DNMT3A

Reason for exclusion in current	N
intronic variants	6

Variant type	N
In-frame deletion (p.F732del)	4
Splice-site/in-frame	3
Splice-site (donor)	2

TET2

Reason for exclusion in current	N
Intronic variants on NM_001127208 isoform	11
Het binomial P >= 0.001	1

Variant type	N
Frameshift/splice-site	1

ASXL1

Variant type	N
Splice-site (c.1086-2A>G, c.1086-1G>A, c.1086-1G>C, c.1720-2A>G, c.1720-2A>T, c.1720-1G>T, c.1720-1G>C)	21

PPM1D

Reason for exclusion in current	N
Splice site	1

TP53

Variant type	N
Stop gained	32
Splice-site (acceptor)	14
Frameshift deletion	9
Splice-site (donor)	8
Frameshift insertion	6
Splice-site + frameshift	1

JAK2

Variant type	N
Inframe deletion (p.542-544del)	1

SRSF2

Variant type	N
In-frame insertion (p.R95dup)	1

Reviewer #1:

I commend the authors on thoughtfully addressing my comments with a substantially improved manuscript. I appreciate the point raised by Reviewer 3 with respect to confounding of the ancestry-CH relationship by smoking. Smoking is difficult to capture in population-based investigations as measures of ever-never or current/former/never do not capture important features such as duration, intensity or type of tobacco used. To provide further support of the robustness of the findings, I suggest a sensitivity analysis performed only among never smokers to demonstrate a similar ancestry effect in this "clean" subset. Likewise, I suggest adding a sentence to the Discussion section to indicate the possibility of residual confounding by unmeasured components of tobacco smoking such as duration of smoking, smoking intensity, etc.

We thank the Reviewer for appreciating the substantial revisions to the manuscript, and the improvements made possible by the reviewers' comments. We also thank the Reviewer for highlighting the point raised by Reviewer #3 on ancestry-CH-smoking relationship. We agree that the "never smokers" group represents a "clean subset" among the different self-reported smoking categories.

Previously, we stratified by smoking status (including a never smoker strata) when investigating the relationship between CH frequency and the fraction of European ancestry in the within MCPS analysis (Extended Data Figure 6f-i). Additionally, we have now assessed the frequency of CH between populations (UKB versus MCPS) after stratifying by smoking status. Reassuringly, and consistent with our previous results, we observe a higher frequency of CH in UKB relative to MCPS among never smokers (Extended Data Fig. 5e). For completeness, we also performed the association analyses among previous, current, and ever smokers, and similarly observed higher CH frequency in UKB relative to MCPS across all three smoking groups (Extended Data Fig. 5f-h).

As there is inherent uncertainty in capturing self-reported smoking intensity/duration, we have now cautioned the readers as to the possibility of residual confounding by unmeasured components of smoking in the Discussion (paragraph 3) as follows:

“...While we have accounted for known risk factors in our analysis (age, sex and smoking), residual confounding may nevertheless still be present, for example, due to unmeasured components of smoking including the duration of smoking and the number of cigarettes smoked daily.”

Reviewer #2 (Remarks to the Author):

I appreciate the extensive additional analyses, particularly the analyses based on the 58 gene panel. The authors have done an excellent job addressing my concerns. I have no further comments.

We are thankful for the positive feedback and kind words, and appreciate the Reviewer’s thoughtful comments, which have significantly contributed to the improvement of our manuscript.

Reviewer #3 (Remarks to the Author):

No further comments.

We thank the Reviewer again for reviewing our manuscript and are pleased that our revisions addressed their comments.